# Adaptive-Hybrid Redundancy with Error Injection

**Nicolas Hamilton** [1] , **Scott Graham** [1,*] , **Timothy Carbino** [1] , **James Petrosky** [2] and **Addison Betances** [1]

1   Department of Electrical and Computer Engineering, Air Force Institute of Technology, Wright-Patterson AFB, OH 45322, USA; nicolas.hamilton@afit.edu (N.H.); timothy.carbino@us.af.mil (T.C.); joan.betancesjorge@afit.edu (A.B.)
2   Department of Engineering Physics, Air Force Institute of Technology, Wright-Patterson AFB, OH 45322, USA; james.petrosky@us.af.mil
*   Correspondence: Scott.Graham@afit.edu; Tel.: +1-937-255-6565 (ext. 4220)

**Abstract:** Adaptive-Hybrid Redundancy (AHR) shows promise as a method to allow flexibility when selecting between processing speed and energy efficiency while maintaining a level of error mitigation in space radiation environments. Whereas previous work demonstrated AHR's feasibility in an error free environment, this work analyzes AHR performance in the presence of errors. Errors are deliberately injected into AHR at specific times in the processing chain to demonstrate best and worst case performance impacts. This analysis demonstrates that AHR provides flexibility in processing speed and energy efficiency in the presence of errors.

**Keywords:** adaptive-hybrid redundancy; radiation effects; single event upset; triple modular redundancy; temporal software redundancy; radiation mitigation; field programmable gate array

## 1. Introduction

Adaptive-Hybrid Redundancy (AHR) was developed to enable flexibility in radiation hardening redundancy methods for space vehicles. AHR incorporates Triple Modular Redundancy (TMR) and Temporal Software Redundancy (TSR) such that AHR can switch between TMR and TSR modes as needed [1]. This previous work demonstrated that AHR functions as designed, switches from TMR to TSR, and uses less energy to complete programs than TMR while completing those programs in less time than TSR in an error free environment [1]. The objective of this paper is to further illustrate the advantages and flexibility of AHR when compared to TMR or TSR alone both in error free and error prone simulated environments. This paper does not seek to prescribe how much time AHR should spend in TMR or TSR operating modes, but rather to provide a new redundancy framework to space vehicle designers, mission planners, and operators so they can decide how much time AHR should spend in each mode based upon radiation environment, processing speed requirements, energy consumption requirements, and mission requirements.

The remainder of this section discusses previous redundancy mitigation research and the architecture of AHR. Section 2 discusses the methods used to evaluate the performance of AHR. Section 3 discusses the results of the AHR performance evaluation. Section 4 discusses the impact of the results and concludes the paper.

### 1.1. Background

Previous Single-Event Upset (SEU) mitigation research has focused on hardware, software, hybrid, or adaptive redundancy techniques. This section will briefly review some of the research leading up to AHR.

### 1.1.1. Hardware Redundancy

Dual-Modular Redundancy (DMR) describes an SEU mitigation method that operates two processors in parallel by providing those processors simultaneous identical inputs and operating those processors on a common clock. A hardware comparator module compares the outputs of the two processors to ensure that they are identical. If the comparator detects a difference, an error has occurred, and both processors' internal state is restored to a previous known state that was stored to radiation hardened/immune memory. In DMR, the processors are periodically interrupted to save their internal state to memory; this is called a Save/Restore Point [2–6]. To create the Save/Restore Point, the comparator issues commands to both processors to write each register to memory. Upon receiving register values from the two processors, they are compared to ensure they are equal before writing the result to memory. If an error is detected at this stage, the comparator enters error recovery mode and returns to the last Save/Restore Point. After all registers are written to memory, the Program Counter value is written to memory as well. During error recovery, the comparator resets both processors, then issues a series of load commands to both processors to load each register value in the Save/Restore Point into registers. The comparator finishes the error recovery process by issuing a branch command to return the processors to the Program Counter value stored in the Save/Restore Point. The comparator performs Save/Restore Point creation and error recovery by traversing a series of states in its internal finite state machine which is implemented in hardware.

The timing of Save/Restore Point creation is application specific and depends on factors such as radiation environment and operational needs. The Save/Restore Point time period is specified as a set number of instructions to complete between Save/Restore Points. Save/Restore Point creation takes time away from processors performing intended tasks and slows execution time so there is some pressure to maximize the amount of time between Save/Restore Point creation events. However, in the event an error occurs, the amount of time between Save/Restore Points dictates the amount of time that must be spent to recover from an error to the point at which the error initially occurred. This represents a pressure to reduce the amount of time between Save/Restore Point creation.

Triple-Modular Redundancy (TMR) describes an SEU mitigation method that operates three processors in a similar manner to DMR with some notable differences. First, the hardware comparator module is replaced with a hardware majority voter. Secondly, when the majority voter detects that two of three processors agree while one disagrees, the disagreeing processor is reset and the internal state of the two agreeing processors is copied to the disagreeing processor so that all three processors are in agreement. This greatly reduces the recovery time when compared to DMR. In the unlikely event that two or more processors encounter an error such that all three processors disagree, TMR restores all three processors to a previous Save/Restore Point in the same way as DMR: the voter contains a hardware implemented finite state machine to create the Save/Restore Point and recover from errors [7–13]. The voter is typically assumed to be immune to errors or is hardened in some way. This research assumes that the voter is immune to errors as well. While the voter is this research is not implemented in such a way to be error immune or radiation hardened, radiation hardening could be achieved through radiation shielding, hardware redundancy, or hardening by design.

N-Modular Redundancy (NMR) is a majority voting redundancy method that is similar to TMR, but uses N processors instead of three processors. It is considered more robust than TMR because a permanent single processor failure will result in N-1-Modular Redundancy. So long as N-1 is greater than or equal to three, the majority voting redundancy method still functions. However, there is an energy penalty to be paid because NMR uses at least N times as much energy to complete a program as a single processor with no redundancy [14–16].

### 1.1.2. Software Redundancy

The first software redundancy method this paper discusses, Error Detection by Duplicated Instructions (EDDI), is very similar to DMR, but is implemented in software instead of hardware. This software redundancy method runs on a single processor. Dual redundancy is implemented by

duplicating all instructions that do not store data to memory. Each duplicated instruction stores its results to a different, physically separated register from the original instruction to achieve spatial separation of the original and duplicate results. This greatly reduces the likelihood that a single, or multiple, radiation event(s) will cause the exact same error in both the original and duplicated results. The DMR comparison function is implemented in software by adding a comparison instruction immediately prior to any store instruction. If the original and duplicate register are identical, the original is stored to memory. If the original and duplicate are not identical, an error has occurred and program execution jumps to code that performs error recovery. This error recovery code restores the state of the processor to a previously saved state called a Save/Restore Point in a similar manner to DMR, but use software instead of a hardware finite state machine. Similarly to DMR, EDDI periodically interrupts normal program execution to create Save/Restore points by executing code designed to create the Save/Restore points [17–19]. EDDI was proposed by Oh et al. [17,18] while Tokponnon et al. discuss a very similar software redundancy method [19]. Table 1 shows an example of what EDDI code looks like in the "redundant set" when compared to a non-redundant instruction set called the "original set". In this example, LUI is the MIPS load upper immediate instruction, ADD is the MIPS addition instruction, SW is the MIPS store word instruction, and BNE is the MIPS branch if not equal instruction. ERR is the value of the distance which the BNE should jump in code execution if R3 and R17 are not equal. OFFSET is the value that should be added to the memory location specified by R0 where R3 should be stored. Any value indicated by R# is a register number.

**Table 1.** Simple software redundancy example.

| Instruction Number | Original Set | Redundant Set |
|:---:|:---:|:---:|
| 1 | LUI R1 1 | LUI R1 1 |
| 2 | LUI R2 2 | LUI R15 1 |
| 3 | ADD R3 R1 R2 | LUI R2 2 |
| 4 | SW R3 R0 OFFSET | LUI R16 2 |
| 5 | | ADD R3 R1 R2 |
| 6 | | ADD R17 R15 R16 |
| 7 | | BNE R3 R17 ERR |
| 8 | | SW R3 R0 OFFSET |

Oh et al. [20] and Reis et al. [21,22] improved EDDI by adding signature detection in order to determine when Program Counter errors have occurred as a result of a missed branch or illegal branch. Signature detection methods break a program into segments and computes segment signatures at compile time and inserts a segment signature computation and a segment comparison instruction into each segment. The signatures are unique and are dependent upon the preceding segment and the current segment. At runtime, the signature is recomputed and compared to the compile time signature. Any discrepancy between the two is interpreted as a Program Counter error as a result of a missed branch or illegal branch between segments [20–22].

1.1.3. Hybrid Redundancy

Hybrid redundancy can take many forms and can consist of any number of combinations of hardware, software and error correcting codes.

The first hybrid redundancy method this paper discusses is only applicable to Field Programmable Gate Arrays (FPGAs) which have a configuration memory that is vulnerable to Single Event Upsets (SEUs). Many FPGAs configuration memories are comprised of Static Random Access Memory (SRAM) cells which are highly susceptible to SEUs. These configuration memories specify constants, logic functions, and signal routing on the FPGA. Any of these can have a catastrophic effect on the intended function of the hardware designed into the FPGA. Those who wish to implement a processor on an FPGA typically combine a TMR-like method with a method of correcting configuration memory called internal scrubbing. The primary concern with FPGAs is the configuration memory rather than the

registers used in the processor. Internal scrubbing detects and corrects errors in the configuration memory, but can only do so at the memory refresh rate. TMR is used as a stop-gap measure to ensure outputs are correct in spite of configuration errors until internal scrubbing can correct those errors. The TMR method only does majority voting and does not correct faulty processors [23–28].

Another method of hybrid redundancy duplicates instructions in software, similar to EDDI, but uses a hardware comparator similar to the DMR comparator [29]. A method that juxtaposes the software and hardware portions has also been implemented that uses hardware for redundancy and software for comparisons [30].

A few methods of hybrid redundancy combine hardware or software redundancy with error correcting codes to protect processor registers and/or memory [31–33].

### 1.1.4. Adaptive Redundancy

Only two adaptive redundancy approaches were found in the literature survey. The first is a very simple approach that uses a radiation sensor to detect the ambient radiation environment and determines when to implement TMR and when to operate using a single processor without any mitigation [34]. This is also the only example discovered in literature that applies adaptive redundancy to a processor.

The second uses three different software redundancy methods to protect memory. Each method differs in the degree of error protection, memory access speed, and energy consumption. When there are very few SEUs occurring, the method that provides the least error protection, operates the fastest, and uses the least amount of energy is utilized. As the SEU rate increases, the method that provides an intermediate level of error protection, intermediate memory access speed, and intermediate energy consumption is used. As the SEU rate becomes too great for the intermediate level of protection to handle, the method that provides the greatest level of error protection is used at the expense of the slowest memory access and greatest energy consumption [35].

### 1.2. Adaptive-Hybrid Redundancy

AHR, as implemented in this work and the previous work [1], consists of TMR and TSR. The TMR implementation is just as described in Section 1.1.1 and the TSR implementation is the EDDI method described in Section 1.1.2. For this research, the time between creation of Save/Restore Points was arbitrarily selected to be 10,000 instructions for TMR and AHR operating in TMR mode and 250 main program loops for TSR and AHR Operating in TSR mode (many real-world programs typically have a main loop which is repeated numerous times until program completion). These values were chosen to ensure that every program would create at least one Save/Restore Point during its execution. These are tunable parameters that a space vehicle designer, mission planner, or operator could change as needed based upon the program running on the processor, performance requirements, radiation environment, and mission needs.

A simple illustration of the TMR architecture is shown in Figure 1. The previous work demonstrated that a program running in an error free environment in TMR takes 65% longer to run than a program running on a single processor with no redundancy because the voter adds delay to all processor inputs and outputs. The TMR architecture also uses three times the instantaneous power of a single processor because there are three processors instead of one. A program running TMR takes approximately 430% more energy to complete due to the number of processors and the added time taken to run the program [1].

A simple illustration of the TSR architecture is shown in Figure 2. The EDDI TSR method uses a special compiler to take a normal program and make it into a TSR program. The previous work demonstrated that EDDI TSR programs take 113% longer to complete than non-redundant programs and the TSR architecture uses the same amount of instantaneous power as a single processor because the TSR architecture only uses a single processor. However, TSR uses 113% more energy to complete a program than a non-redundant program because it takes 113% longer to run that program [1].

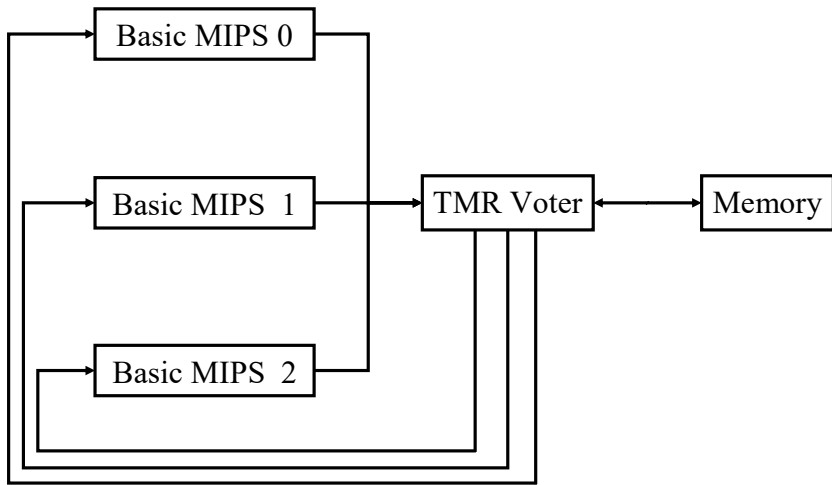

**Figure 1.** Triple Modular Redundancy (TMR) MIPS simplified block diagram.

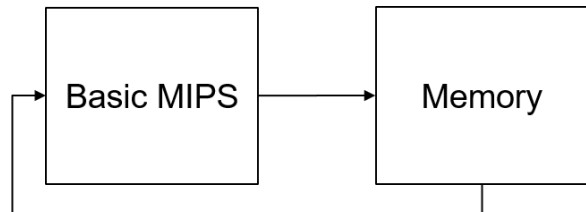

**Figure 2.** Temporal Software Redundancy (TSR) MIPS simplified block diagram.

The AHR architecture adds a module called the AHR Controller to the TMR architecture as shown in Figure 3. The AHR Controller is assumed to be immune from errors for this research just as the TMR voter is assumed to be immune from errors. When AHR operates in TMR mode, the TMR Voter and three processors operate normally and the signals between the TMR Voter and memory and from the TMR Voter to the three processors are passed through combinational logic in the AHR Controller with minimal delay. Figure 4 shows how signals flow when AHR is operating in TMR mode by illustrating connected signals and modules that are operational in black and those that are not in red. When operating in TSR mode, the AHR Controller turns off the TMR Voter and two of the three processors. The remaining single processor communicates directly with memory by passing signals through combinational logic in the AHR Controller with minimal delay. Figure 5 shows how signals flow when AHR is operating in TSR mode by illustrating connected signals and modules that are operational in black and those that are not in red.

AHR begins in TMR mode and switches to TSR mode after a predetermined number of TMR instructions are completed without encountering an error. AHR remains in TSR mode so long as two consecutive errors do not occur. Two errors are considered consecutive if a second error occurs after TSR recovers from a first error and the second error occurs before TSR can create a new save/restore point. If consecutive errors occur, AHR transitions to TMR when the second error is detected. If TSR creates a new save/restore point before encountering a second error, the second error is not consecutive and AHR continues in TSR mode. This approach gives TSR mode an opportunity to recover from errors so long as the error rate is sufficiently low.

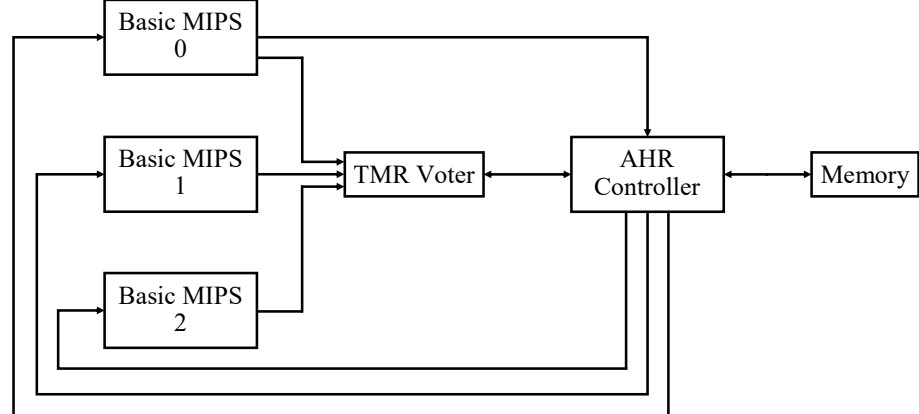

**Figure 3.** AHR MIPS simplified block diagram.

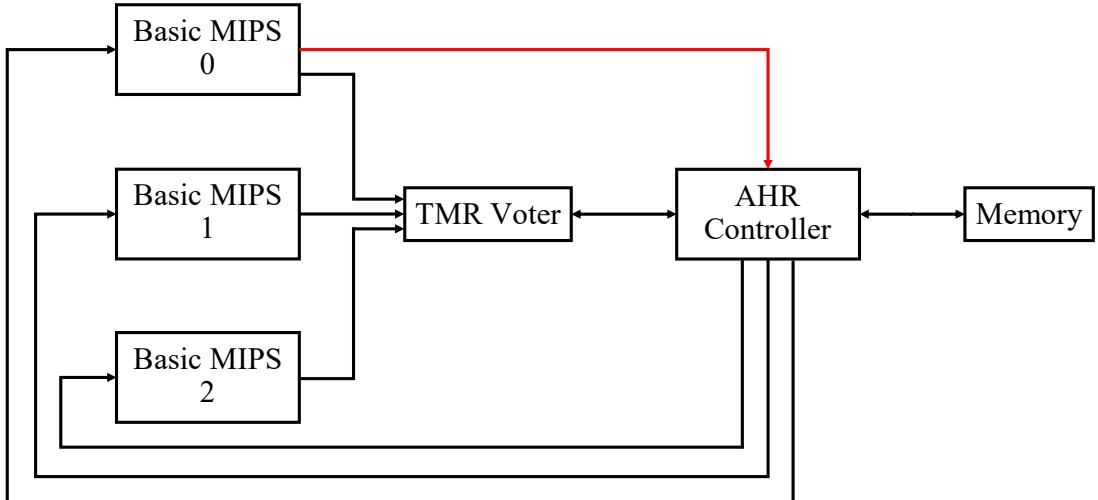

**Figure 4.** AHR MIPS in TMR mode simplified block diagram with disabled portions in red.

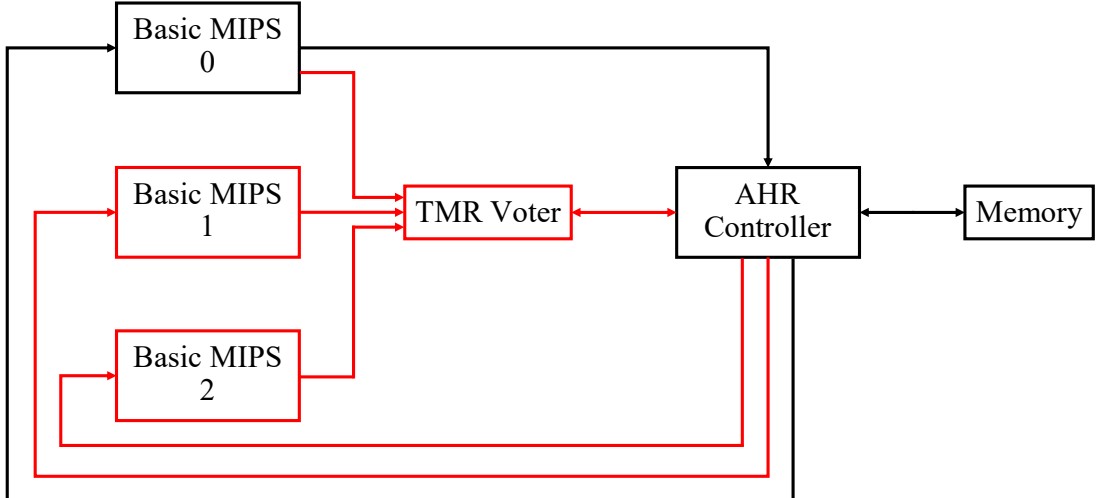

**Figure 5.** AHR MIPS in TSR mode simplified block diagram with disabled portions in red.

The processor upon which AHR, TMR, and TSR were based on past work and are based in this work is the Basic MIPS processor which is a simplified MIPS32$^{\text{TM}}$ processor that only supports 33

of the 168 MIPS32$^{\text{TM}}$ instructions [36]. The details of the Basic MIPS, TMR MIPS, and AHR MIPS architectures are available in Air Force Institute of Technology technical reports [36–38].

AHR was shown to bridge the gap between these two methods by switching between TMR and TSR so that it runs faster than TSR and uses less energy than TMR at the expense of running slower than TMR and using more energy than TMR [1]. In past work, AHR started in TMR mode and switched to TSR mode after TMR successfully completed 15,000 instructions without an error in an error free simulation. This work will examine how AHR performs when the TMR to TSR switch point is varied as well as how it performs when errors are injected into the simulation.

An appropriate error rate to be used for analysis was determined to be approximately one Single Event Upset (SEU) per hour by conducting radiation experiments on an Intel Cyclone V Field Programmable Gate Array (FPGA) at Sandia National Laboratories and performing post experimental analysis. This is the expected average rate for a space vehicle using the Cyclone V FPGA over the life of the mission. However, the SEU rate for real missions fluctuates as a result of orbital position with reference to earth's magnetic field lines (i.e., SEU rate increases over the South Atlantic Anomaly) and as a result of solar weather (i.e., changes in solar activity impact the SEU rate). This FPGA was chosen as a representative inexpensive commercial-off-the-shelf technology for past research and this research. Based on the experiment and the knowledge that nearly every program used in the previous work and the current work has a runtime of approximately 50ms or less, it is reasonable to expect no more than one error per program run.

## 2. Materials and Methods

The only materials required for this work were a computer with a network connection and MATLAB installed. Previous work implemented Basic, TMR, TSR, and AHR MIPS architectures in Very High Speed Integrated Circuit (VHSIC) Hardware Description Language (VHDL). Previous work also made use of Mentor Graphics Questa Sim to simulate the VHDL architectures in order to determine important timing parameters for use in MATLAB analyses to compute program runtimes on the various architectures. These timing parameters include the time to complete individual instructions, program start, program close, save/restore point creation, error recovery, and TMR to TSR transition. The results of those simulations are used in the simulations and analyses in the current work. Previous work also made use of the Intel PowerPlay Early Power Estimator for Cyclone IV and Cyclone V tool to estimate the instantaneous power used by Basic, TMR, TSR, and AHR MIPS [1].

Section 2.1 will discuss the mechanism used to inject errors into Basic MIPS. Section 2.2 will discuss when and where errors will be injected and provide the analysis tools necessary to calculate program runtime when an error is injected.

### 2.1. Error Injection Mechanism

In previous works, fault injection was performed using software manipulation or direct electrical injection for Hardware-in-the-Loop (HITL) simulations [16,17,23,24,27,28,35]. This research modifies the Basic MIPS Datapath to directly inject errors into a specific general purpose register at a specific point during program execution. This is software error injection because Basic MIPS is simulated in software for this research. The General Purpose Registers (GPRs) were selected as the injection points because general purpose registers account for 992 bits that are susceptible to SEUs in the Basic MIPS architecture whereas the remaining susceptible registers account for only 68 bits. These remaining registers are the program counter, instruction register, Finite State Machine (FSM) register, and additional Datapath registers. If an SEU were to occur, it has a 94% chance of occurring in a GPR as opposed to a 6% chance of occurring elsewhere. Additionally, EDDI is unable to detect and correct all errors occurring in these other registers, so it was determined that it would be better to inject errors into GPRs so that TSR and AHR operating in TSR could detect and correct the injected errors for a more accurate performance comparison between TMR, TSR, and AHR. A detailed schematic showing how the Basic MIPS Datapath was modified to incorporate GPR error injection is provided in Appendix A.

If an error occurred in the program counter of TSR or AHR operating in TSR mode, the error would cause program execution to jump backwards and repeat instructions or forwards and skip instructions. If a backwards jump were performed, the impact would be increased runtime and energy used to complete the program. Additionally, an incorrect result may be written to memory. If a forward jump were performed, instructions would be skipped. In some instances, this might be detected if one of two redundant instructions were skipped. In other instances, the illegal branch might go undetected and several results that should have been written to memory might not be written to memory. Additionally, program runtime and energy to complete the program would be reduced.

If an error occurred in the instruction register of TSR or AHR operating in TMR mode, the error would simply be corrected by Basic MIPS internal mechanism if the error resulted in an invalid instruction. If the error caused the instruction to change to another valid instruction, it might be detected if the error affected the result of a duplicated instruction. This same error might result in changing a store word instruction to something else and a result would fail to be written to memory resulting in an undetected error. A store word instruction error might also cause a result to be written to a wrong location in memory and this error would also go undetected. An instruction register error might also create a branch instruction where none existed or change the address of a branch instruction.

A FSM register error could cause TSR or AHR operating in TSR mode to incorrectly jump to another state which would most likely cause the processor to remain trapped in the incorrect state. Processing would cease and the program would never complete. At present TSR and AHR have no protection against such an error.

The other Datapath registers are used in determining whether to execute a program jump or not when evaluating a branch instruction. An error in one of these registers would cause the incorrect branch path to be taken. This could result in a longer program execution time with greater energy usage or a shorter program execution time with reduced energy usage.

*2.2. Error Injection Timing*

Errors are injected into GPRs that are going to be stored to memory so that the TMR Voter and the TSR comparison instruction will detect them and initiate error recovery operations. This is done immediately prior to the TMR instruction to store a GPR to memory and immediately prior to a TSR comparison instruction before storing a GPR to memory. Injecting an error for every store word instruction for all 1000 programs used in the current work is not feasible. Instead, errors are selectively injected to probe the minimum and maximum time and energy performance of the TMR, TSR, and AHR architectures. In all three architectures, the best-case errors will minimize the amount of time and energy expended on error detection and recovery and the worst-case errors will maximize the amount of time and energy expended on error detection and recovery.

Errors intended to maximize the amount of time and energy needed to recover from an error are injected immediately before Save/Restore Point creation. A TMR processor or AHR processor in TMR mode has to perform error recovery, then repeat 10,000 instructions to return to the point at which the error initially occurred. A TSR processor or AHR processor in TSR mode has to perform error recovery, then repeat 250 main program loops to return to the point at which the error initially occurred. Errors intended to minimize the amount of time and energy needed to recover from an error are injected immediately after Save/Restore Point creation. For all processors, this minimizes the number of instructions that must be repeated after error recovery to return to the point at which the error initially occurred. For a more detailed discussion concerning when errors are injected and the calculations performed to determine the program runtime for each type of error, please refer to Appendix B.

The 1000 programs used in this work were randomly generated and exercise the full range of Basic MIPS instructions and GPRs. Individual programs may not exercise the full range of Basic MIPS instructions or utilize all GPRs; however, some programs utilize most, if not all of the Basic MIPS instructions and some programs utilize all the GPRs.

*2.3. Energy Used When Errors Are Injected*

The energy computations are much simpler than the timing computations. The energy to complete a TMR or TSR MIPS program is the time to complete the TMR or TSR MIPS programs multiplied by the TMR or TSR MIPS instantaneous power respectively. The time to complete AHR programs is the time spent in TMR mode multiplied by the TMR MIPS instantaneous power plus the time spent in TSR mode multiplied by the TSR MIPS instantaneous power. For a more detailed discussion concerning the calculations performed to determine the energy used to complete a program for each type of error, please refer to Appendix C.

## 3. Results

This section examines the results of software simulations and computational analysis when errors are injected as described in Sections 2.2 and 2.3.

Figure 6 shows the average time and energy to complete 1000 programs for each error type in each architecture (including no error injection) This figure illustrates how AHR MIPS bridges the gap between TMR MIPS and TSR MIPS performance. The AHR MIPS TMR Type A and Type B-Best errors appear to fall on a line between the TSR MIPS Best-case error and TMR MIPS Type A and Type B-Best errors. A similar pattern appears for AHR MIPS TMR Type B-Worst and TSR Worst-case errors which appear to nearly fall on a line between the TMR Type B-Worst and TSR Worst-case errors. However, this figure does not tell the entire story as the best-case, worst-case, early, and later errors define maximum and minimum bounds for AHR MIPS performance in the presence of errors.

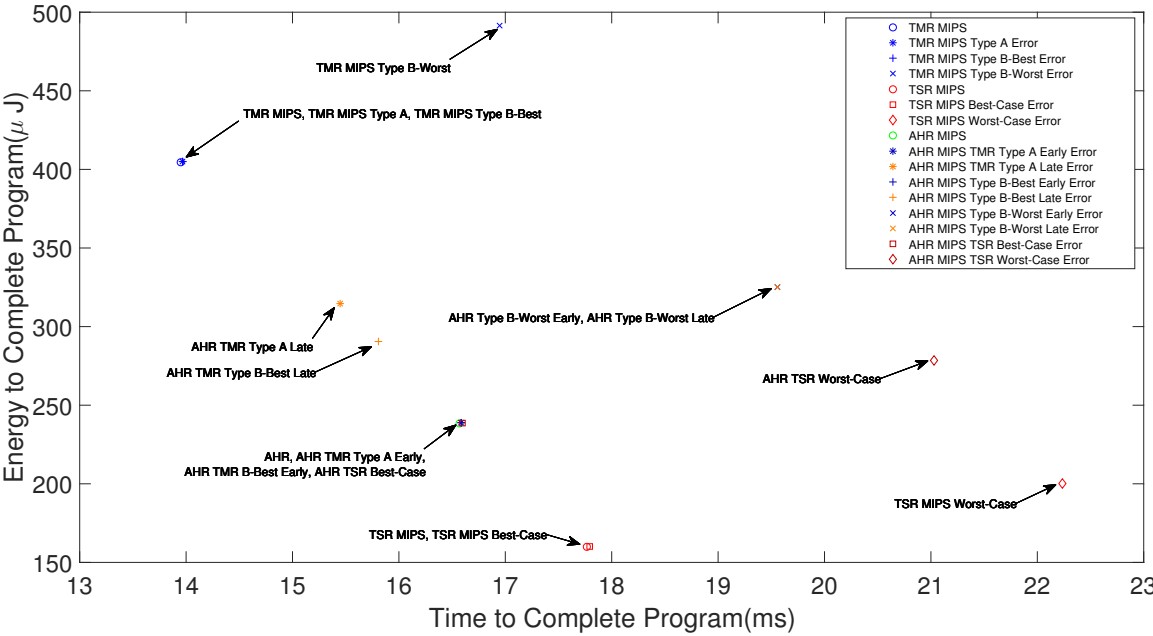

**Figure 6.** Averaged results of software simulation of all errors: energy vs. time to complete.

Figure 7 shows the performance bounds as bounding boxes when the TMR to TSR transition point occurs at 15,000 instructions. Note that the points plotted in this figure are the same as those plotted in Figure 6 and represent the average program completion times; however, the TMR Type B-Best, TSR Best-Case, AHR TMR Type A Early, AHR TMR Type B-Best Early, and AHR TSR Best-Case errors have been omitted from this plot. The TMR Type B-Best case error result was nearly identical to the TMR MIPS with no error result. The TSR MIPS Best-Case error result was nearly identical to the TSR MIPS with no error result. The AHR TMR Type A Early, AHR TMR Type B-Best Early, and AHR TSR Best-Case error results were nearly identical to the AHR MIPS with no error result. The bounding

boxes indicate that the average program completion time should fall somewhere within the bounding box when errors are present.

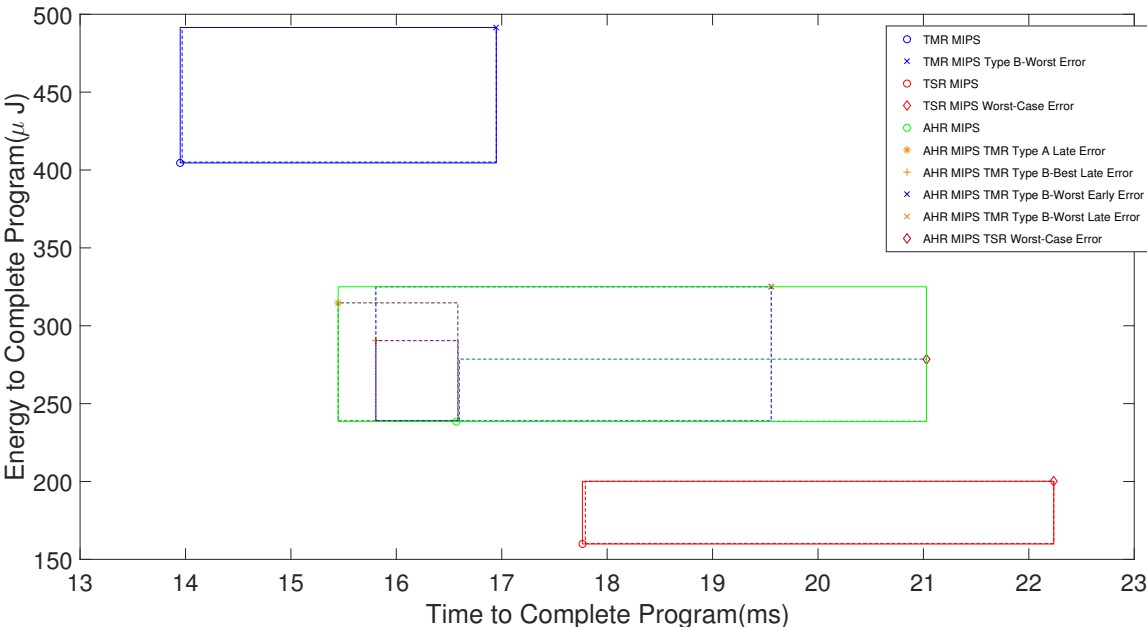

**Figure 7.** Average performance bounds for AHR MIPS with a TMR to TSR point at 15,000 instructions.

The corners of the bounding box encompassing the TMR MIPS Type B-Best and Type B-Worst errors is shown as a dotted blue line. This box indicates that average program completion time and energy usage for a TMR MIPS program encountering a Type B error will end up within this box and it nearly overlaps the second box indicated by a solid blue line. The second box is used to outline the average performance of TMR MIPS when errors may or may not be present. It includes the no error, TMR Type A, and TMR Type B errors.

The corners of the bounding box encompassing the TSR MIPS Best- and Worst-case errors is shown as a red dashed line with the red square and the red diamond at opposing corners to indicate the average Best- and Worst-case error program runtime and energy usage. A second box with a red solid line is used to outline the average performance of TSR MIPS when errors may or may not be present. It includes the no error, TSR Best-case, and TSR Worst-case errors. Once again, these boxes almost overlap one another.

The corners of the bounding box encompassing the AHR MIPS TMR Type A Early and Late errors is shown as a gray dashed line to indicate how a program will perform in terms of time and energy usage on average when a TMR Type A error will occur. Similarly, the purple dashed line indicates the average performance of a program experiencing a TMR Type B-Best case error. The orange dashed line indicates the average performance of a program experiencing a TMR Type B-Worst error, however, no such box is visible in this figure as the TMR Type B-Worst Early and Late errors are identical in this figure. A dark blue dashed line bounding box extends from the left most plus sign (+) to the right most "X" and from the AHR MIPS no error (green circle) to the top most "X" to show the average bounds of a AHR MIPS program that encounters any TMR Type B error. The dashed teal line indicates the bounds of a AHR MIPS TSR error. Finally, the solid green line indicates the average bounds for a AHR MIPS program experiencing any TMR error, TSR error, or no error.

Note that the portion of the bounding box extending to the left of the average AHR MIPS no error runtime and energy usage does not necessarily indicate that an error could occur such that the runtime would decrease without a change in energy usage. It should be expected that a decrease in runtime would correspond to a greater number of instructions being performed in TMR mode and

a resulting energy increase; however, there is insufficient analysis at this time to determine a more precise boundary region and the creation of such a region is left for future work.

Now, because the bounding boxes indicate that the average time and energy used to complete a program in the presence of errors should fall within the boxes, they should not be treated as program specific bounding boxes. It would be trivial to create program specific bounding boxes, but these are not shown here for brevity. However, the next figures will begin to show the versatility of the AHR MIPS approach as the TMR to TSR transition point is varied.

Figures 8–15 show what happens to the bounding boxes as the TMR to TSR transition point increases from 11,000, to 20,000, 30,000, 40,000, 50,000, 60,000, 70,000, and 80,000 instructions. These figures are shown to illustrate the flexibility of AHR MIPS as the TMR to TSR transition point is varied. A space vehicle designer, mission planner, or operator can change the TMR to TSR transition point depending on the program being run on the processor, radiation environment, processing speed requirements, power requirements, and other mission needs. For example, it might be desirable to operate in TMR in a low radiation environment for the purpose of maximizing processing speed when energy usage is not a concern. This is achieved by setting the TMR to TSR transition point to such a large value that the TMR to TSR transition never occurs. Alternatively, it might be desirable to operate in a low power mode regardless of the external radiation environment and its potential impact on registers other than GPRs. This is achieved by setting the TMR to TSR transition point to zero so that AHR switches from TMR back to TSR as quickly as possible after AHR suffers multiple errors in TSR mode.

As the transition point increases, The overall bounding box begins to grow, then shrinks down to match the TMR MIPS bounding box. Note that in some of these figures, the AHR MIPS TMR Type B-Worst Early and Late errors do not always coincide. Note also how the size and shapes of the smaller bounding boxes change. As the TMR to TSR transition point increases, the AHR MIPS TMR Type B-Best bounding box increases in size, then decreases in size until it becomes nonexistent. The AHR MIPS TMR Type B-Worst bounding box increases in size from nonexistence, then decreases in size until becoming nonexistent again. The AHR MIPS TMR Type A box also increases then decreases in size until becoming nearly nonexistent. The AHR MIPS TSR box decreases in size until becoming nearly nonexistent.

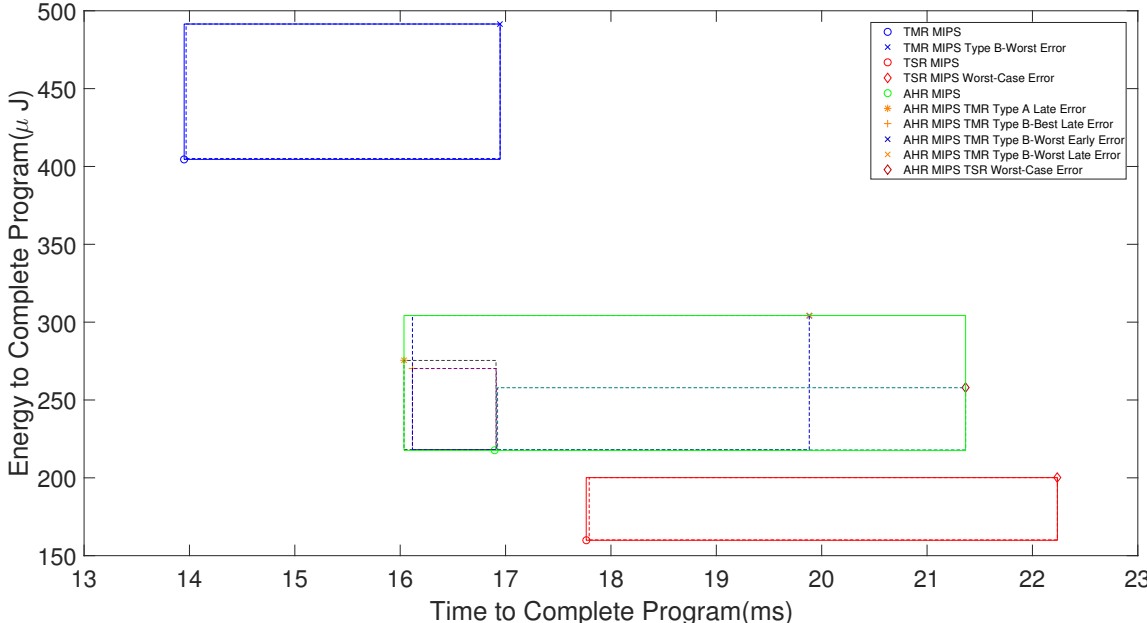

**Figure 8.** Average performance bounds for AHR MIPS with a TMR to TSR point at 11,000 instructions.

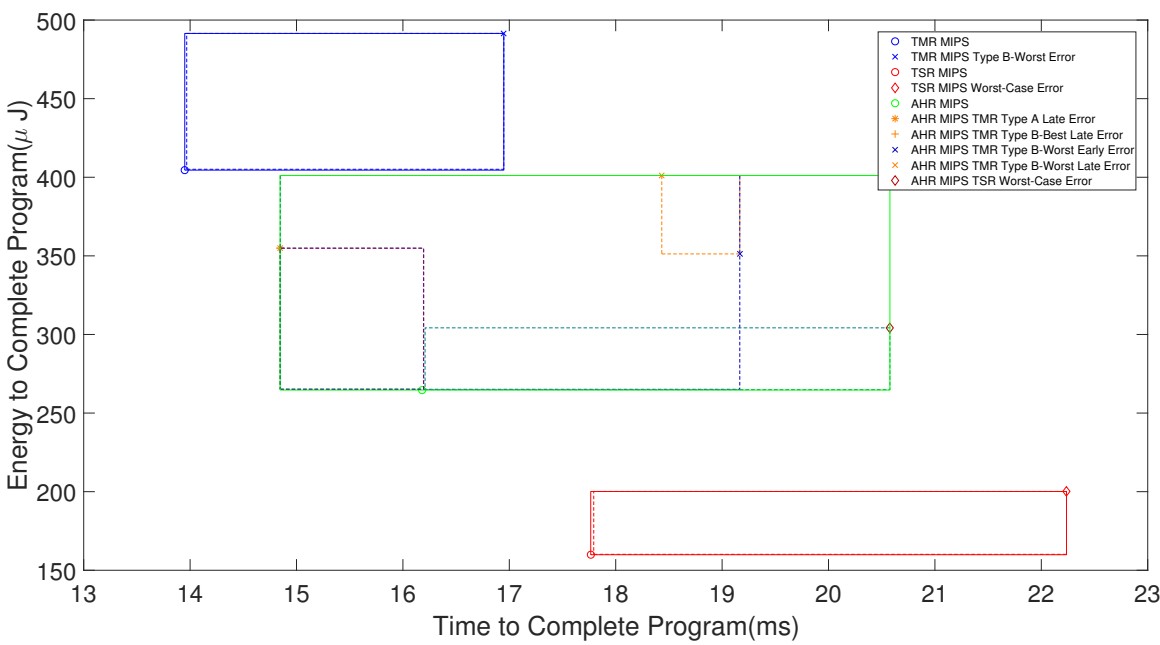

**Figure 9.** Average performance bounds for AHR MIPS with a TMR to TSR point at 20,000 instructions.

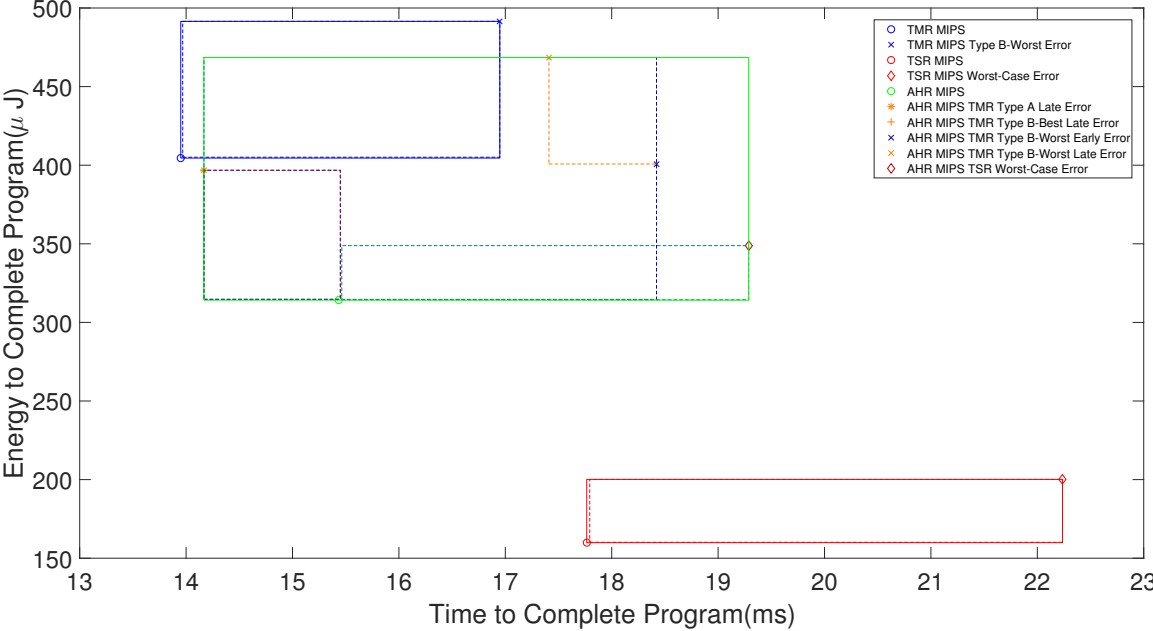

**Figure 10.** Average performance bounds for AHR MIPS with a TMR to TSR point at 30,000 instructions.

While these figures represent the average performance for 1000 programs, they have greater utility when created for a specific program to show how the expected program runtime and energy usage change as the TMR to TSR transition point is changed. A satellite designer, mission planner, or operator could use these to determine the best transition point based on the needs of the system. For example, the TMR to TSR transition point could be selected in order to meet certain performance criteria such as staying under maximum runtime or energy constraints.

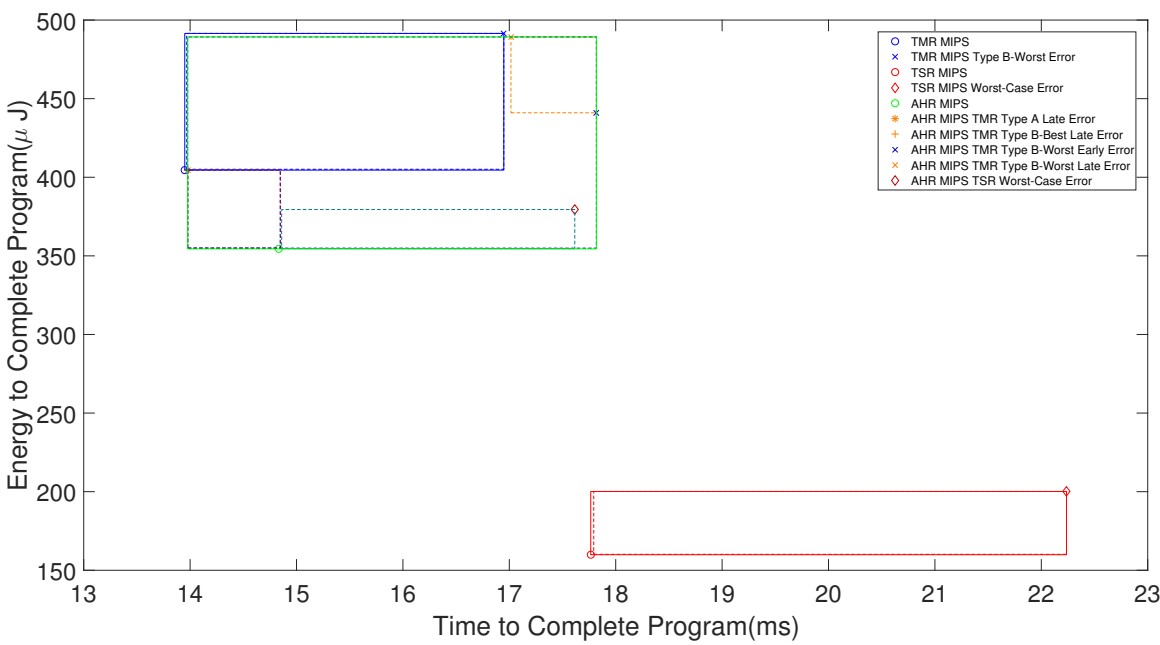

**Figure 11.** Average performance bounds for AHR MIPS with a TMR to TSR point at 40,000 instructions.

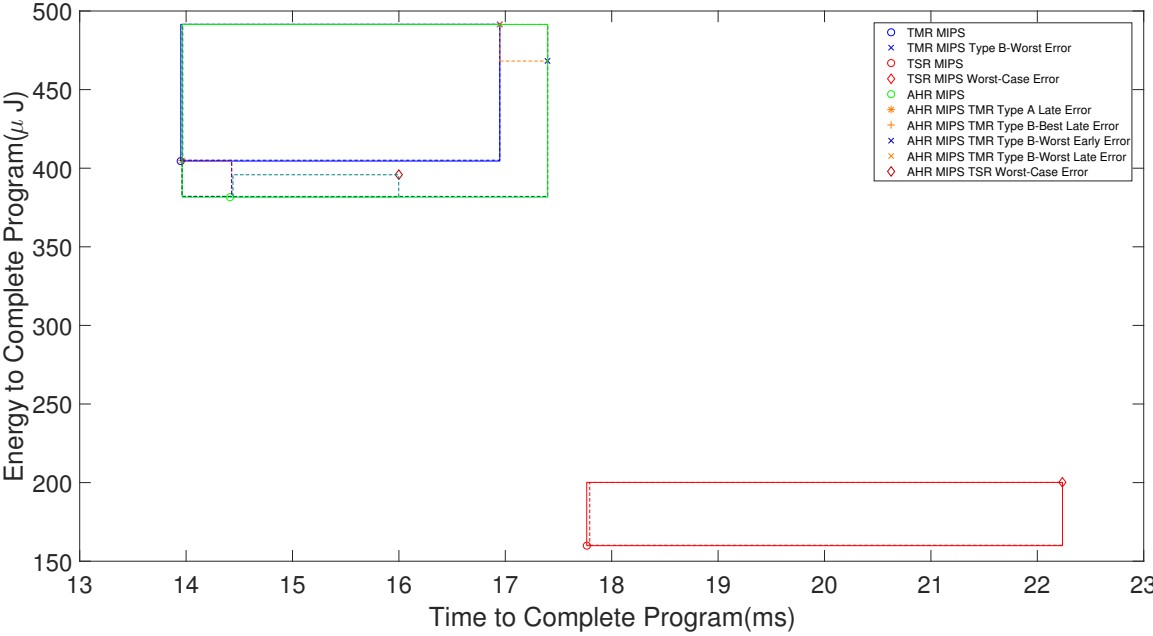

**Figure 12.** Average performance bounds for AHR MIPS with a TMR to TSR point at 50,000 instructions.

Figure 16 shows the same things as Figure 6, but allows the TMR to TSR transition point to vary from 11,000 to 80,000 instructions in increments of 1000. This figure also provides a slightly different view to the bounding boxes in the previous figures. It is most useful in visualizing how the average program runtime and energy usage for each error scenario changes as the TMR to TSR transition point changes. Curves for all AHR MIPS error scenarios, and the no error scenario, become evident. When there are only 11,000 instructions completed in TMR before the TMR to TSR transition point, AHR MIPS behaves much more closely to TSR MIPS. As the transition point moves towards 80,000 instructions, the AHR MIPS results begin moving up and to the left until they coincide with the TMR MIPS results. Note that the AHR MIPS TSR Best- and Worst-case scenarios collapse to the no error solution for TMR MIPS when very little, if any time is spent in TSR MIPS because the TMR to TSR

transition point is no longer reached during the duration of most programs. Similarly, the AHR MIPS TMR Type B-Worst scenarios converge to the TMR Type B-Worst error scenario.

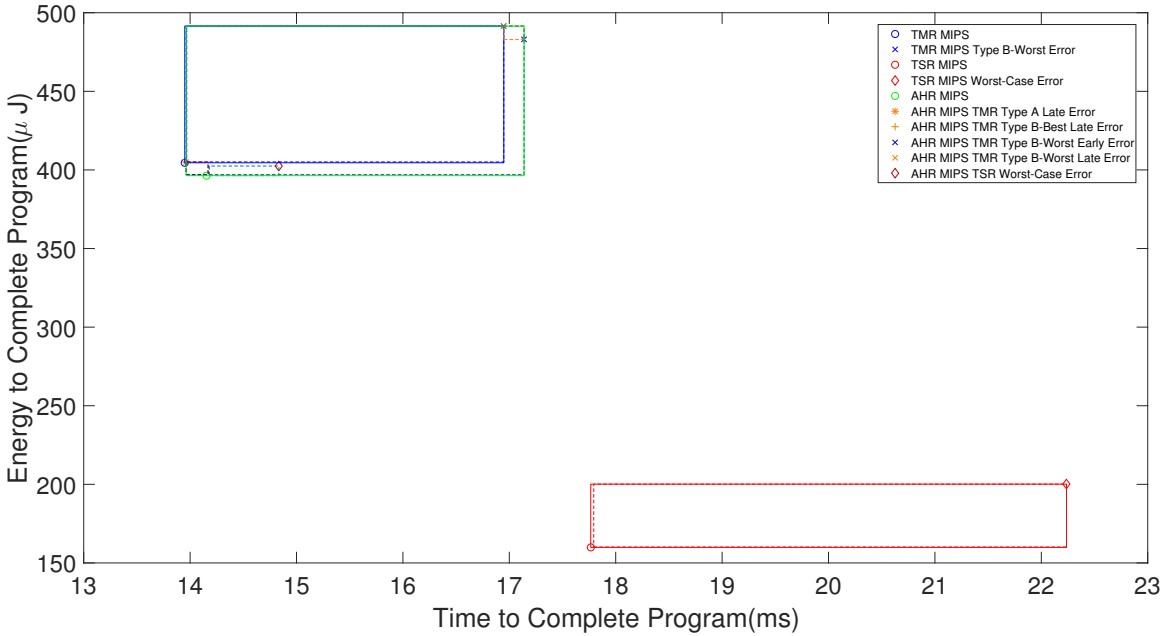

**Figure 13.** Average performance bounds for AHR MIPS with a TMR to TSR point at 60,000 instructions.

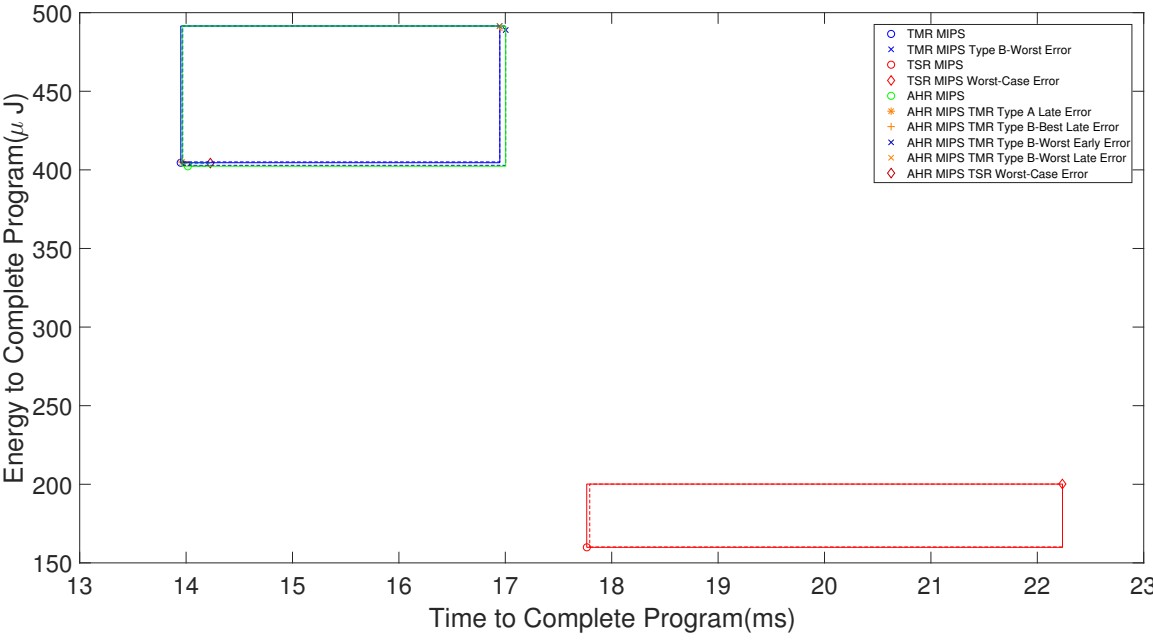

**Figure 14.** Average performance bounds for AHR MIPS with a TMR to TSR point at 70,000 instructions.

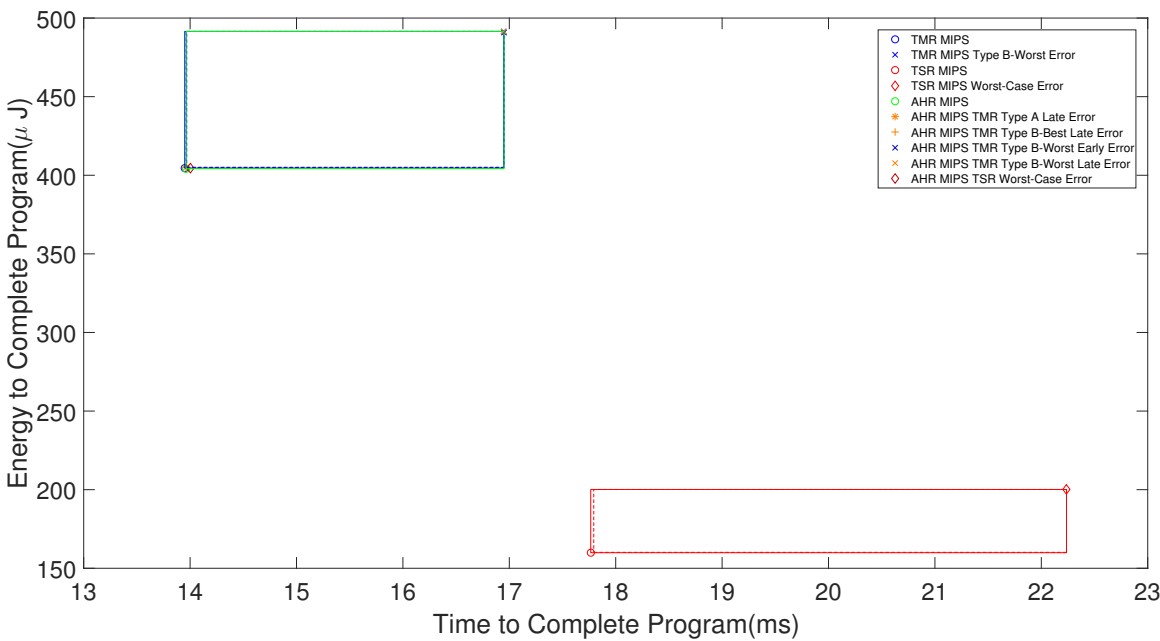

**Figure 15.** Average performance bounds for AHR MIPS with a TMR to TSR point at 80,000 instructions.

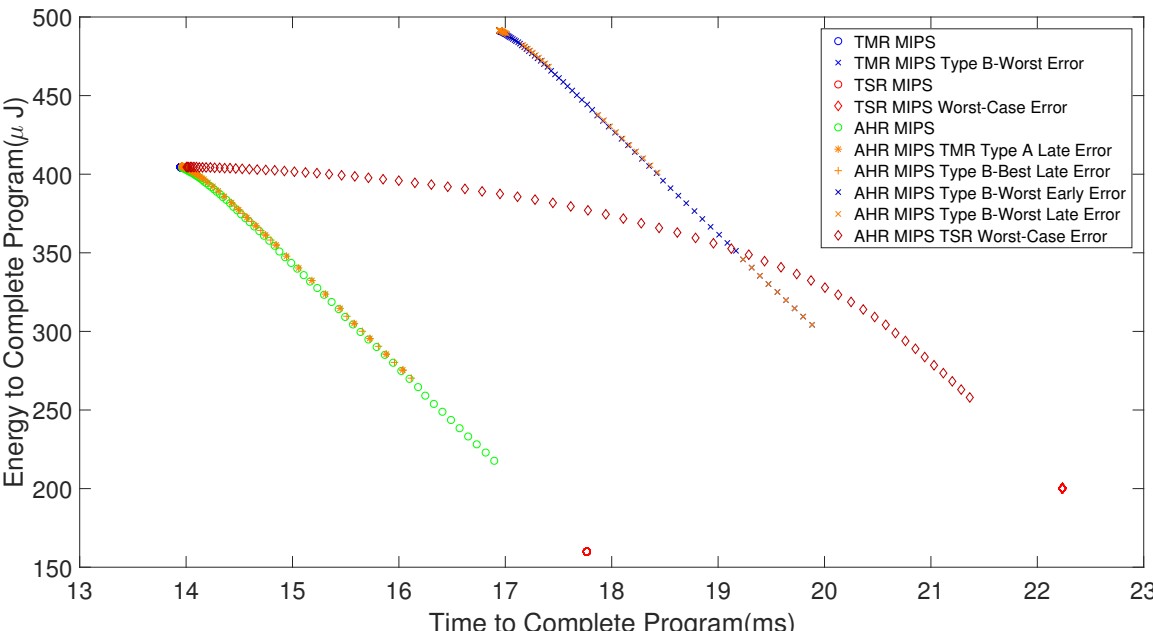

**Figure 16.** TMR to TSR transition varying from 11,000 to 80,000 instructions: energy vs. time to complete.

Another interesting comparison is to look at the average percent difference in runtime and energy usage for each error scenario and no error scenario when compared to Basic MIPS with no errors. The average percent difference for the no error scenarios were given in the previous work [1]. The average percent difference for the programs experiencing errors are given in Equations (1)–(26).

$$
\begin{aligned}
PD_{Time\ TMR\ ErrA\ v\ Basic} &= \cdots \\
\frac{\sum_{n=1}^{N_{programs}} \left[ \dfrac{T_{TMR\ ErrA}(n) - T_{Basic\ MIPS}(n)}{T_{Basic\ MIPS}(n)} \times 100\% \right]}{N_{programs}}
\end{aligned} \tag{1}
$$

$$PD_{Time\ TMR\ ErrB\ Best\ v\ Basic} = \cdots$$
$$\frac{\sum_{n=1}^{N_{programs}} \left[ \frac{T_{TMR\ ErrB\ Best}(n) - T_{Basic\ MIPS}(n)}{T_{Basic\ MIPS}(n)} \times 100\% \right]}{N_{programs}} \tag{2}$$

$$PD_{Time\ TMR\ ErrB\ Worst\ v\ Basic} = \cdots$$
$$\frac{\sum_{n=1}^{N_{programs}} \left[ \frac{T_{TMR\ ErrB\ Worst}(n) - T_{Basic\ MIPS}(n)}{T_{Basic\ MIPS}(n)} \times 100\% \right]}{N_{programs}} \tag{3}$$

$$PD_{Time\ TSR\ Best\ v\ Basic} = \cdots$$
$$\frac{\sum_{n=1}^{N_{programs}} \left[ \frac{T_{TSR\ Best}(n) - T_{Basic\ MIPS}(n)}{T_{Basic\ MIPS}(n)} \times 100\% \right]}{N_{programs}} \tag{4}$$

$$PD_{Time\ TSR\ Worst\ v\ Basic} = \cdots$$
$$\frac{\sum_{n=1}^{N_{programs}} \left[ \frac{T_{TSR\ Worst}(n) - T_{Basic\ MIPS}(n)}{T_{Basic\ MIPS}(n)} \times 100\% \right]}{N_{programs}} \tag{5}$$

$$PD_{Time\ CTMR\ A\ Early\ v\ Basic} = \cdots$$
$$\frac{\sum_{n=1}^{N_{programs}} \left[ \frac{T_{CTMR\ A\ Early}(n) - T_{Basic\ MIPS}(n)}{T_{Basic\ MIPS}(n)} \times 100\% \right]}{N_{programs}} \tag{6}$$

$$PD_{Time\ CTMR\ A\ Late\ v\ Basic} = \cdots$$
$$\frac{\sum_{n=1}^{N_{programs}} \left[ \frac{T_{CTMR\ A\ Late}(n) - T_{Basic\ MIPS}(n)}{T_{Basic\ MIPS}(n)} \times 100\% \right]}{N_{programs}} \tag{7}$$

$$PD_{Time\ CTMR\ B\ Best\ Early\ v\ Basic} = \cdots$$
$$\frac{\sum_{n=1}^{N_{programs}} \left[ \frac{T_{CTMR\ B\ Best\ Early}(n) - T_{Basic\ MIPS}(n)}{T_{Basic\ MIPS}(n)} \times 100\% \right]}{N_{programs}} \tag{8}$$

$$PD_{Time\ CTMR\ B\ Best\ Late\ v\ Basic} = \cdots$$
$$\frac{\sum_{n=1}^{N_{programs}} \left[ \frac{T_{CTMR\ B\ Best\ Late}(n) - T_{Basic\ MIPS}(n)}{T_{Basic\ MIPS}(n)} \times 100\% \right]}{N_{programs}} \tag{9}$$

$$PD_{Time\ CTMR\ B\ Worst\ Early\ v\ Basic} = \cdots$$
$$\frac{\sum_{n=1}^{N_{programs}} \left[ \frac{T_{CTMR\ B\ Worst\ Early}(n) - T_{Basic\ MIPS}(n)}{T_{Basic\ MIPS}(n)} \times 100\% \right]}{N_{programs}} \tag{10}$$

$$PD_{Time\ CTMR\ B\ Worst\ Late\ v\ Basic} = \cdots$$
$$\frac{\sum_{n=1}^{N_{programs}} \left[ \frac{T_{CTMR\ B\ Worst\ Late}(n) - T_{Basic\ MIPS}(n)}{T_{Basic\ MIPS}(n)} \times 100\% \right]}{N_{programs}} \tag{11}$$

$$PD_{Time\ CTSR\ Best\ v\ Basic} = \cdots$$
$$\frac{\sum_{n=1}^{N_{programs}} \left[ \frac{T_{CTSR\ Best}(n) - T_{Basic\ MIPS}(n)}{T_{Basic\ MIPS}(n)} \times 100\% \right]}{N_{programs}} \tag{12}$$

$$PD_{Time\ CTSR\ Worst\ v\ Basic} = \cdots$$
$$\frac{\sum_{n=1}^{N_{programs}} \left[ \dfrac{T_{CTMR\ Worst}(n) - T_{Basic\ MIPS}(n)}{T_{Basic\ MIPS}(n)} \times 100\% \right]}{N_{programs}} \tag{13}$$

$$PD_{Energy\ TMR\ ErrA\ v\ Basic} = \cdots$$
$$\frac{\sum_{n=1}^{N_{programs}} \left[ \dfrac{E_{TMR\ ErrA}(n) - E_{Basic\ MIPS}(n)}{E_{Basic\ MIPS}(n)} \times 100\% \right]}{N_{programs}} \tag{14}$$

$$PD_{Energy\ TMR\ ErrB\ Best\ v\ Basic} = \cdots$$
$$\frac{\sum_{n=1}^{N_{programs}} \left[ \dfrac{E_{TMR\ ErrB\ Best}(n) - E_{Basic\ MIPS}(n)}{E_{Basic\ MIPS}(n)} \times 100\% \right]}{N_{programs}} \tag{15}$$

$$PD_{Energy\ TMR\ ErrB\ Worst\ v\ Basic} = \cdots$$
$$\frac{\sum_{n=1}^{N_{programs}} \left[ \dfrac{E_{TMR\ ErrB\ Worst}(n) - E_{Basic\ MIPS}(n)}{E_{Basic\ MIPS}(n)} \times 100\% \right]}{N_{programs}} \tag{16}$$

$$PD_{Energy\ TSR\ Best\ v\ Basic} = \cdots$$
$$\frac{\sum_{n=1}^{N_{programs}} \left[ \dfrac{E_{TSR\ Best}(n) - E_{Basic\ MIPS}(n)}{E_{Basic\ MIPS}(n)} \times 100\% \right]}{N_{programs}} \tag{17}$$

$$PD_{Energy\ TSR\ Worst\ v\ Basic} = \cdots$$
$$\frac{\sum_{n=1}^{N_{programs}} \left[ \dfrac{E_{TSR\ Worst}(n) - E_{Basic\ MIPS}(n)}{E_{Basic\ MIPS}(n)} \times 100\% \right]}{N_{programs}} \tag{18}$$

$$PD_{Energy\ CTMR\ A\ Early\ v\ Basic} = \cdots$$
$$\frac{\sum_{n=1}^{N_{programs}} \left[ \dfrac{E_{CTMR\ A\ Early}(n) - E_{Basic\ MIPS}(n)}{E_{Basic\ MIPS}(n)} \times 100\% \right]}{N_{programs}} \tag{19}$$

$$PD_{Energy\ CTMR\ A\ Late\ v\ Basic} = \cdots$$
$$\frac{\sum_{n=1}^{N_{programs}} \left[ \dfrac{E_{CTMR\ A\ Late}(n) - E_{Basic\ MIPS}(n)}{E_{Basic\ MIPS}(n)} \times 100\% \right]}{N_{programs}} \tag{20}$$

$$PD_{Energy\ CTMR\ B\ Best\ Early\ v\ Basic} = \cdots$$
$$\frac{\sum_{n=1}^{N_{programs}} \left[ \dfrac{E_{CTMR\ B\ Best\ Early}(n) - E_{Basic\ MIPS}(n)}{E_{Basic\ MIPS}(n)} \times 100\% \right]}{N_{programs}} \tag{21}$$

$$PD_{Energy\ CTMR\ B\ Best\ Late\ v\ Basic} = \cdots$$
$$\frac{\sum_{n=1}^{N_{programs}} \left[ \dfrac{E_{CTMR\ B\ Best\ Late}(n) - E_{Basic\ MIPS}(n)}{E_{Basic\ MIPS}(n)} \times 100\% \right]}{N_{programs}} \tag{22}$$

$$PD_{Energy\ CTMR\ B\ Worst\ Early\ v\ Basic} = \cdots$$
$$\frac{\sum_{n=1}^{N_{programs}} \left[ \dfrac{E_{CTMR\ B\ Worst\ Early}(n) - E_{Basic\ MIPS}(n)}{E_{Basic\ MIPS}(n)} \times 100\% \right]}{N_{programs}} \tag{23}$$

$$PD_{Energy\ CTMR\ B\ Worst\ Late\ v\ Basic} = \cdots$$
$$\frac{\sum_{n=1}^{N_{programs}} \left[ \frac{E_{CTMR\ B\ Worst\ Late}(n) - E_{Basic\ MIPS}(n)}{E_{Basic\ MIPS}(n)} \times 100\% \right]}{N_{programs}} \tag{24}$$

$$PD_{Energy\ CTSR\ Best\ v\ Basic} = \cdots$$
$$\frac{\sum_{n=1}^{N_{programs}} \left[ \frac{E_{CTSR\ Best}(n) - E_{Basic\ MIPS}(n)}{E_{Basic\ MIPS}(n)} \times 100\% \right]}{N_{programs}} \tag{25}$$

$$PD_{Energy\ CTSR\ Worst\ v\ Basic} = \cdots$$
$$\frac{\sum_{n=1}^{N_{programs}} \left[ \frac{E_{CTSR\ Worst}(n) - E_{Basic\ MIPS}(n)}{E_{Basic\ MIPS}(n)} \times 100\% \right]}{N_{programs}} \tag{26}$$

The average percent difference equations were used to calculate the average percent difference for all programs experiencing errors and no errors when the TMR to TSR transition point from 11,000 to 80,000 in increments of 1000. The results for the runtime calculations are shown in Figure 17 and the results for energy usage calculations are shown in Figure 18. These figures really highlight how AHR MIPS runtime and energy performance changes when compared to Basic MIPS as the TMR to TSR transition point changes. As in some of the previous figures, the TMR Type A and TMR Type B-Best error results are omitted because they are nearly identical to the TMR no error results. The same is true for the TSR Best error results because they are identical to the TSR no error results. Additionally, the AHR TMR Type A Early, AHR TMR Type B-Best Early, and AHR TSR Best error results have been omitted because they are nearly identical to the AHR no error results. The first thing to note from these figures is how the AHR MIPS no error, AHR TMR Type A, AHR TMR Type B-Best, AHR TSR Best, and AHR TSR Worst average percent differences approach the TMR average percent difference as the number of instructions before the TMR to TSR transition increases. This is consistent with prior results because AHR MIPS performance is nearly identical to TMR MIPS performance as the number of instructions that AHR MIPS processes in TSR mode approaches zero and nearly all instructions are processed in TMR mode. Additionally, the AHR MIPS TMR Type B-Worst average percent differences approach the TMR Type B-Worst average percent difference, which is also expected for the same reasons just given.

There are a few other things to note from the percent difference figures. The first is that programs experiencing an AHR TMR Type A Late error complete faster than programs experiencing an AHR TMR Type B-Best Late error. Both of these complete faster than AHR MIPS programs experiencing no error, AHR TMR Type A Early, AHR TMR Type B-Best Early, and AHR TSR Best-case errors. The no error, AHR TMR Type A Early, AHR TMR Type B-Best Early, and AHR TSR Best-case error scenarios all take less time to complete than programs experiencing AHR TMR Type B-Worst Early, AHR TMR Type B Worst Late, and AHR TSR Worst-case errors. Programs experiencing AHR TMR Type B-Worst Late errors always complete faster than those experiencing AHR TMR Type B-Worst Early errors. AHR MIPS programs experiencing TSR Worst-case errors have the worst runtime when the TMR to TSR transition point is under about 30,000, but runs faster than programs experiencing TMR Type B-Worst Early errors when the transition point is greater than 31,000 instructions and faster than programs experiencing TMR Type B-Worst Late errors when the transition point is greater than about 37,000 instructions.

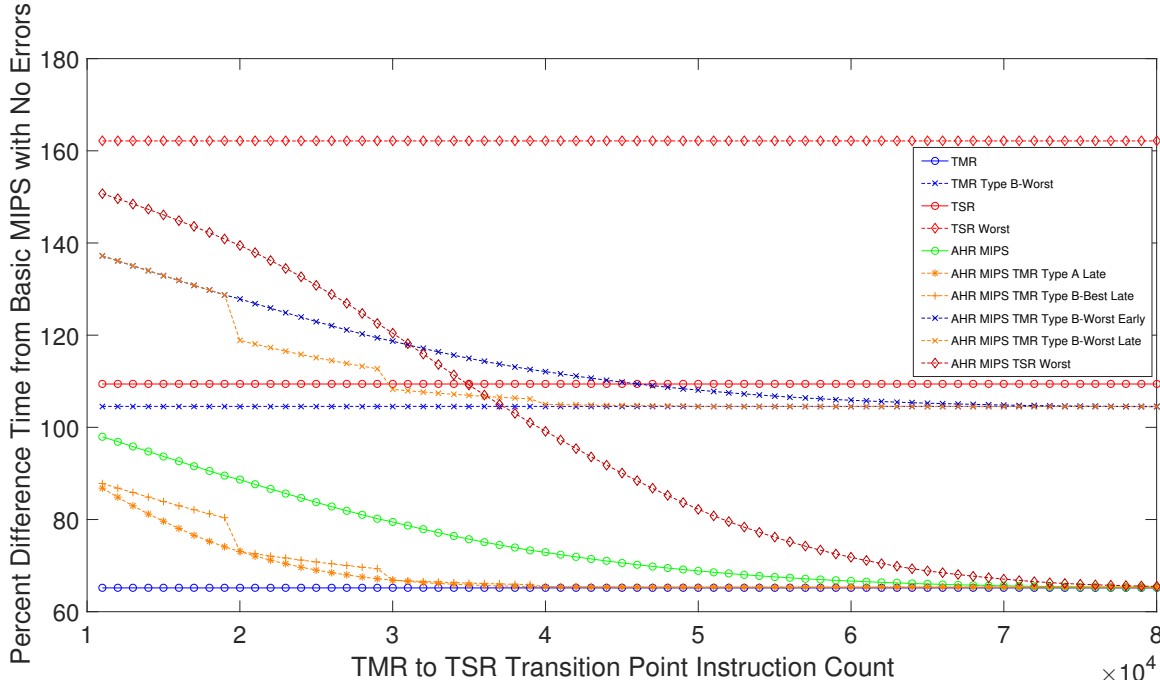

**Figure 17.** AHR MIPS TMR to TSR transition varying from 11,000 to 80,000 instructions: energy vs. time to complete.

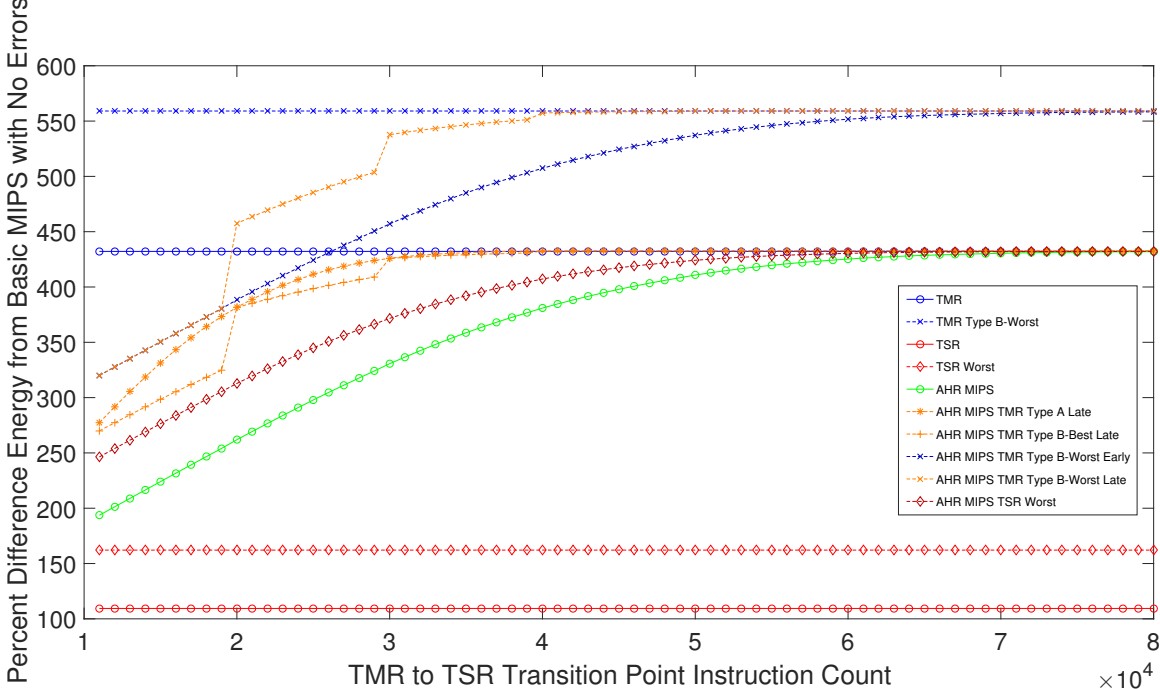

**Figure 18.** AHR MIPS TMR to TSR transition varying from 11,000 to 80,000 instructions: energy vs. time to complete.

AHR MIPS programs experiencing no error, TMR Type A Early, TMR Type B-Best Early, and TSR Best-case all take about the same amount of energy to complete and use less energy than an AHR MIPS program experiencing any other type of error. AHR MIPS programs experiencing TMR Type B-Worst Late errors use the most energy followed by programs experiencing TMR Type B-Worst Early errors, then TMR Type A Late errors, then TSR Worst-case errors.

One final thing to note are the jump discontinuities in the AHR MIPS TMR Type B-Best Late and AHR MIPS TMR Type B-Worst Late timing and energy percent differences. These are a direct result of the TMR to TSR transition point moving passed one of the TMR save/restore point creation times which occur every 10,000 instructions. Note that these discontinuities occur as the TMR to TSR transition point passes 20,000, 30,000, and 40,000 instructions. This is because these late errors go from having a minimal impact when the TMR to TSR transition point occurs immediately before a save/restore point creation to a maximum impact when the TMR to TSR transition point occurs immediately after a save/restore point creation.

Figures 19 and 20 are essentially derivative plots of Figures 17 and 18 except that they use the average time and energy results rather than the percent differences. Each point on these graphs represent the difference in average time and average energy to complete 1000 different programs with the given error type (or no error at all) from one AHR transition point value to the previous AHR transition point value where these transition points started at 11,000 instructions, ended at 80,000 instructions, and had step sizes of 1000 instructions. Plots like these may help a mission planner determine the most optimal point, in terms of processing speed and energy usage, at which to transition AHR from TMR mode to TSR mode.

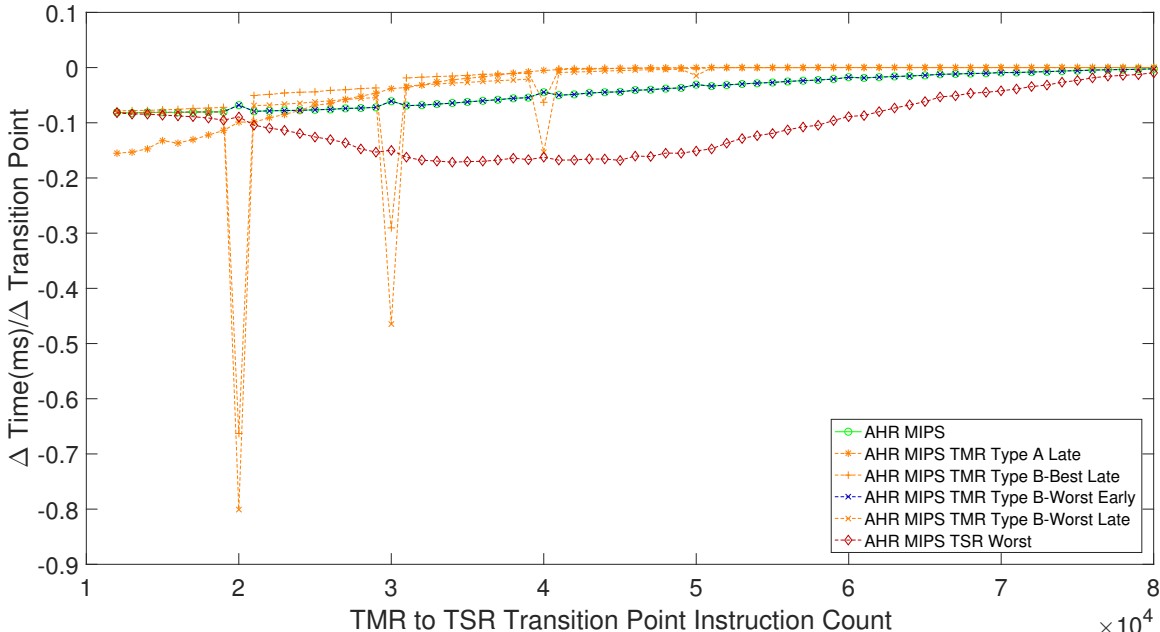

**Figure 19.** Time Difference Between Successive Steps of TMR to TSR Transition Point When Varying from 11,000 to 80,000 in Steps of 1000.

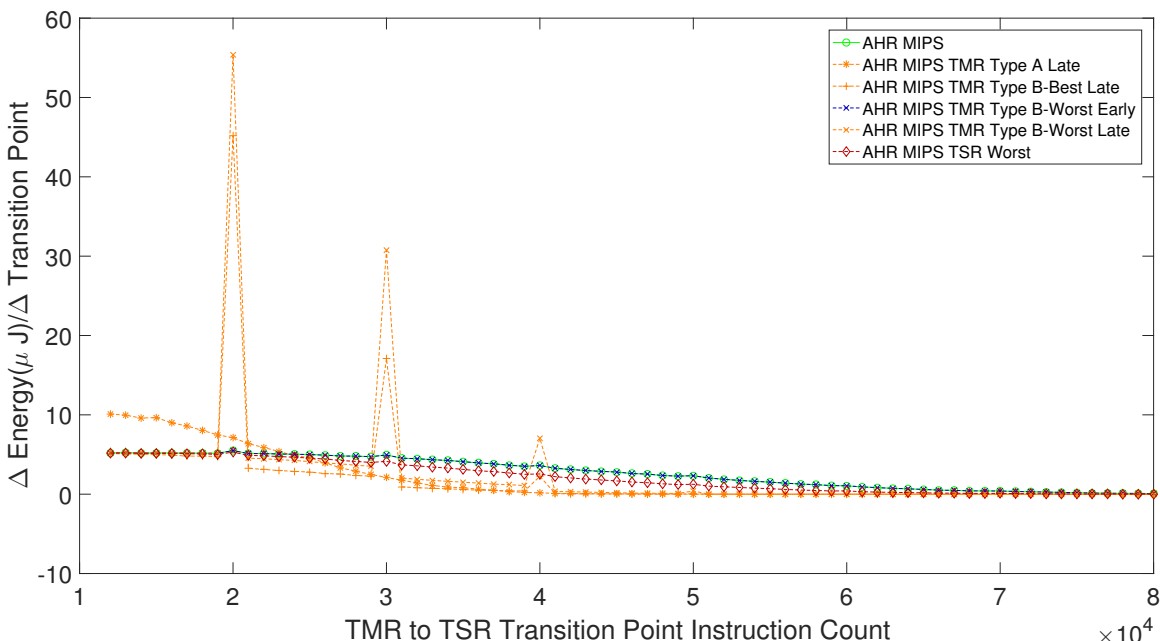

**Figure 20.** Energy difference between successive steps of TMR to TSR transition point when varying from 11,000 to 80,000 in steps of 1000.

## 4. Discussion

AHR uses less energy than TMR and takes less time than TSR to complete programs when errors are injected. Additionally, changing the TMR to TSR transition point allows space vehicle designers, mission planners, and operators the flexibility to select operating points that meet mission processing speed and energy usage requirements not only under optimal error free conditions, but also in the worst-case error scenarios. This was demonstrated through simulation results shown in Section 3 where the time to complete programs varied between the TMR time to complete a program and the TSR time to complete a program as the TMR to TSR transition point was varied. This was also shown as the energy used to complete programs varied between the TMR energy used to complete a program and the TSR energy used to complete a program. The figures illustrated how a space vehicle designer, mission planner, or operator could choose a TMR to TSR transition point that meets the specific needs of their mission. As previously noted, if a mission needed to maximize processing speed at the expense of increased energy usage regardless of the external radiation environment, the TMR to TSR transition point could be set to such an arbitrarily large value that AHR always remains in TMR mode. In contrast, if a mission needs to minimize energy usage at the expense of slower processing speeds regardless of the radiation environment, the TMR to TSR transition point could be set to zero to ensure that AHR remains in TSR mode. For mission needs in between these two extremes, the TMR to TSR transition point could be set to a value that meets certain timing and energy performance criteria while accounting for the radiation environment and its impact on processing speed and energy usage. Additionally, the transition point can be program specific for a processor entrusted with running many different programs so that the transition point is optimized for each program. Furthermore, the transition point can be varied at any time over the course of the mission. It could even be changed during a single orbit to ensure an optimal value at all times when radiation levels and mission needs are taken into account.

Future work will implement TMR, TSR, and AHR on a Cyclone V FPGA to determine how they perform under error free and error injection conditions in terms of time and energy performance. This will be done in an effort to verify that this method works in application and not just in the realm of simulation and analysis.

Another area for future work is expanding AHR to include more redundancy methods such as dual modular redundancy, N-modular redundancy, and advanced TSR methods that can detect and correct program counter errors, which EDDI is unable to detect.

**Author Contributions:** Conceptualization, N.H.; methodology, N.H.; software, N.H.; validation, N.H., S.G., T.C. and A.B.; formal analysis, N.H.; investigation, N.H.; resources, N.H.; data curation, N.H. and J.P.; writing, original draft preparation, N.H.; writing, review and editing, N.H. and S.G.; visualization, N.H.; supervision, S.G.; project administration, S.G.; funding acquisition, T.C.

**Funding:** No sponsor funding provided for this research. Authors are United States Government employees and compensated by the government.

**Conflicts of Interest:** The authors declare no conflict of interest. The funders had no role in the design of the study; in the collection, analyses, or interpretation of data; in the writing of the manuscript; nor in the decision to publish the results.

**Disclaimer:** The views expressed in this paper are those of the authors, and do not reflect the official policy or position of the United States Air Force, Department of Defense, or the U.S. Government. This document has been approved for public release; distribution unlimited, Case #88ABW-2019-4400.

## Abbreviations

The following abbreviations are used in this manuscript:

| | |
|---|---|
| AHR | Adaptive-Hybrid Redundancy |
| EDDI | Error Detection by Duplicated Instructions |
| FPGA | Field Programmable Gate Array |
| FSM | Finite State Machine |
| GPR | General Purpose Register |
| TMR | Triple Modular Redundancy |
| TSR | Temporal Software Redundancy |
| SET | Single Event Transient |
| SEU | Single Event Upset |

## Appendix A. Basic MIPS Datapath Error Injection Schematics

Errors are injected into Basic MIPS GPRs by adding hardware to the Basic MIPS Datapath. The Basic MIPS Datapath is shown in Figure A1. The modified Basic MIPS Datapath that enables error injection is shown in Figure A2 and changes are denoted in red. The Error_Inject module overrides the inputs to the GPR_Bank at a predetermined program counter value and loop count value when Basic MIPS is in state zero. The Error_Inject module injects an error at this predetermined time by inverting the value of a single predetermined bit of a predetermined register in the GPR_Bank.

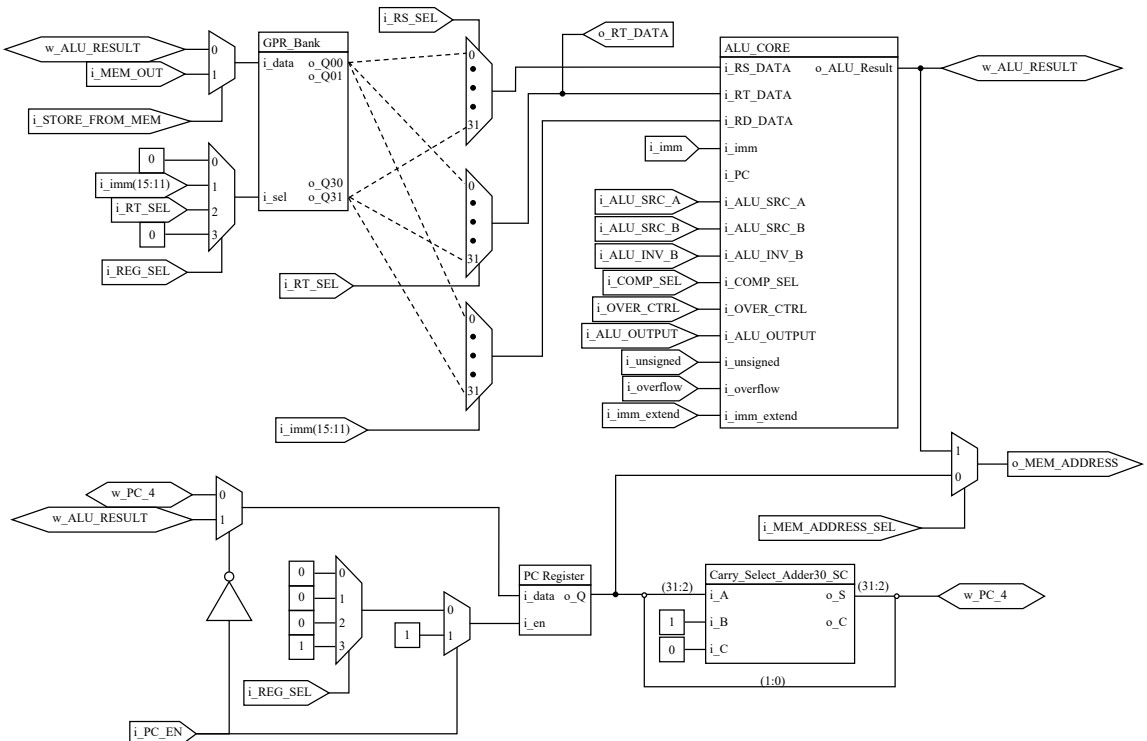

**Figure A1.** Basic MIPS Datapath Schematic.

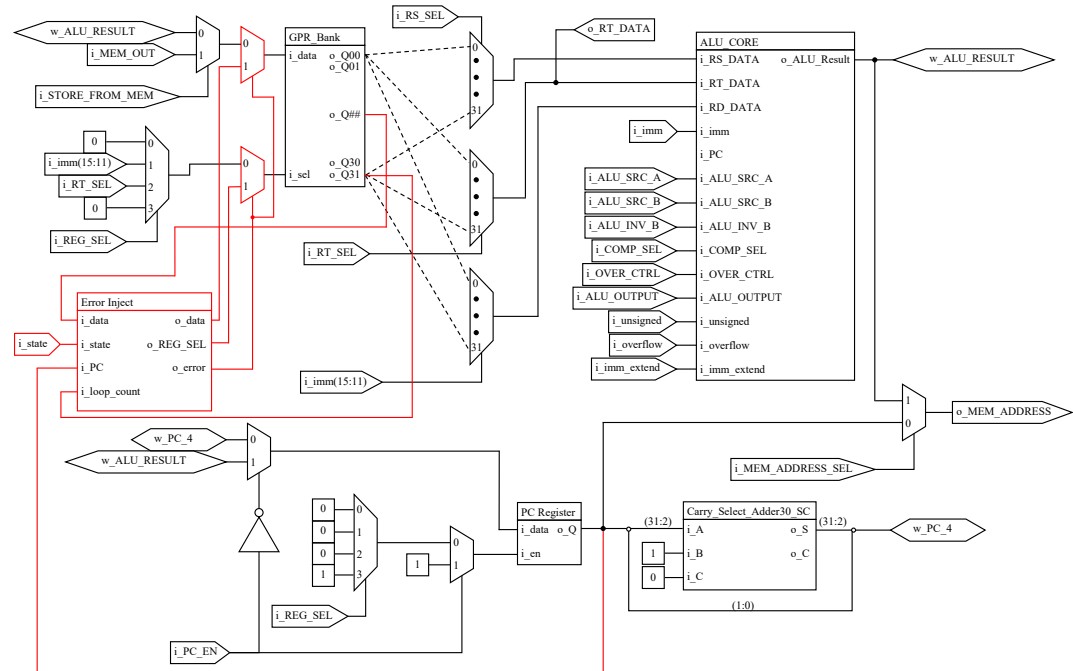

**Figure A2.** Basic MIPS Datapath With Error Injection Schematic.

## Appendix B. Error Injection Timing and Calculations

TMR errors are divided into single and multiple processor errors. A single processor error, called a TMR Type A error, recovers to the same instruction at which the initial error occurred as shown in Figure A3. A multiple processor error, called a TMR Type B error, occurs when all three processors disagree and all three processors are reset and restored to a previously saved state called a save/restore

point. This is shown in Figure A4. In this figure, the acronym SRP denotes points in the TMR program execution at which a save/restore point is created. The TMR Type A error is one example of a best-case TMR error. A TMR Type B error that occurs immediately after creation of a save/restore point is another example of a best-case TMR error because it minimizes the number of instructions that must be recomputed to fully recover from the error. This is called a TMR Type B-Best error. The worst-case TMR error occurs when a Type B error occurs during save/restore point creation and maximizes the number of instructions to be recomputed to fully recover from the error. This is called a TMR Type B-Worst error. The Type B-Best and Type B-Worst errors are shown in Figure A5.

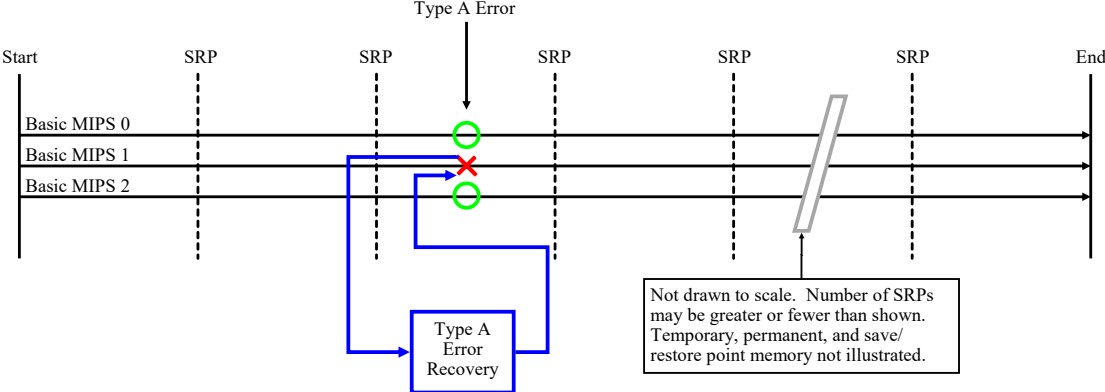

**Figure A3.** TMR MIPS Type A error timing diagram.

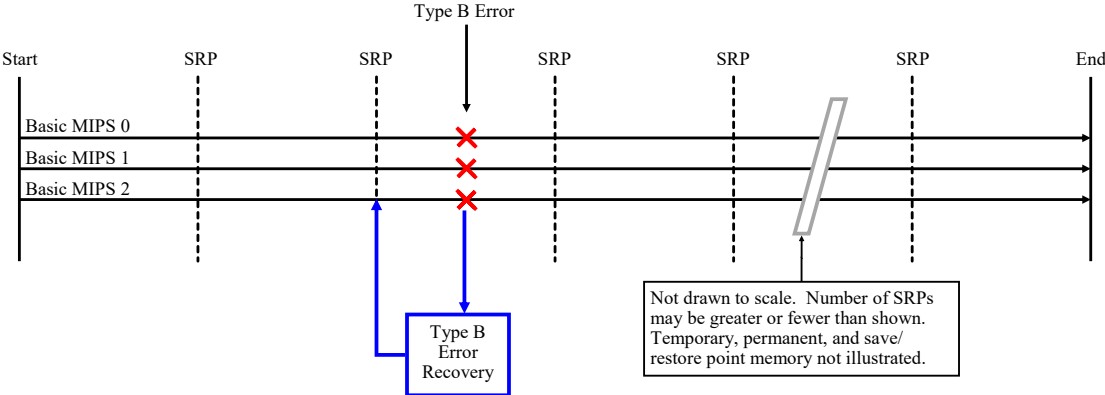

**Figure A4.** TMR MIPS Type B error timing diagram.

The runtime for a TMR MIPS Type A error is given in Equation (A1) where $T_{TMR\ MIPS}$ is the time for TMR MIPS to complete a program in the absence of an error from the previous work [1], $T_{TMR\ ttdA}$ is error detection time, $T_{TMR\ recA}$ is the Type A recovery time, $T_{TMR\ retA}$ is the time to return to the instruction at which the error occurred, and $T_{TMR\ repA}$ is the time required to repeat the instruction at which the error occurred. The last four of these values are determined from simulation.

$$T_{TMR\ ErrA} = T_{TMR\ MIPS} + T_{TMR\ ttdA} + T_{TMR\ recA} + T_{TMR\ retA} + T_{TMR\ repA} \tag{A1}$$

The runtime for TMR MIPS Type B Best-case error is given in Equation (A2) where $T_{TMR\ ttdB}$ is the error detection time, $T_{TMR\ recB}$ is the Type B recovery time, $T_{TMR\ retB\ Best}$ is the time to re-accomplish the instructions between the last completed save/restore point and the instruction at which the error occurred. The time to detect the error and recover from the error are determined from simulation, but the time to return from the error to the point at which the error occurred is determined by analysis.

$$T_{TMR\ ErrB\ Best} = T_{TMR\ MIPS} + T_{TMR\ ttdB} + T_{TMR\ recB} + T_{TMR\ retB\ Best} \tag{A2}$$

While there are many locations in a program where a TMR Type B-Best error may occur, the absolute Best-case error is the one that minimizes the number of instructions between the return from save/restore point creation and the store word instruction following it. In order to determine which pairing of save/restore point creation and store word instructions has the shortest distance between them, the loop count and instruction index of every save/restore point creation and store word instruction must be determined. The store word instruction indices are simply located by examining the program. Equation (A3) shows how to calculate where save/restore point creation occurs where $SI_{TMR}$ is a vector containing the instruction index in the TMR program where save/restore points are created, $SL_{TMR}$ is a vector containing the program loop count values where the save/restore points are created, and $ST_{TMR}$ is a vector containing the amount of time from the beginning of the program to the points at which save/restore points are created. $ST_{TMR}$ is not used now in calculating the TMR Type B-Best program completion time, but will be used shortly.

$$
\begin{aligned}
&for\ m = 0\ to\ n_{SRP} - 1 \\
&\quad if\ m = 0 \\
&\qquad SI_{TMR}(m+1) = 1 \\
&\qquad SL_{TMR}(m+1) = 0 \\
&\qquad ST_{TMR}(m+1) = 0 \\
&\quad else \\
&\qquad SI_{TMR}(m+1) = mod(m \cdot n_{save} - n_{TMR\ init}, N_{TMR}) + n_{TMR\_init} + 1 \\
&\qquad SL_{TMR}(m+1) = \left\lfloor \frac{m \cdot n_{save} - n_{TMR\ init}}{n_{TMR}} \right\rfloor \\
&\qquad T_{add} = \sum_{n=n_{TMR\ init}+1}^{SI_{TMR}(m+1)-1} t_{I_{TMR\ n}} \\
&\qquad ST_{TMR}(m+1) = T_{TMR\ init} + T_{TMR\ loop} \cdot SL_{TMR}(m+1) + T_{add} + \cdots \\
&\qquad (m-1) \cdot T_{TMR\ SRP} \\
&\quad end \\
&end
\end{aligned}
\tag{A3}
$$

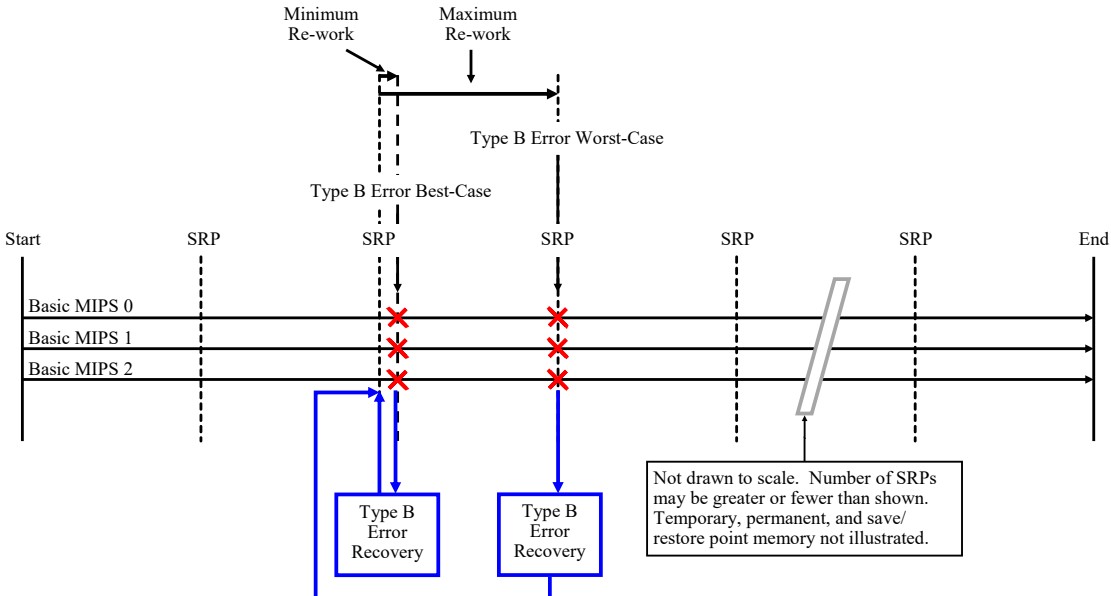

**Figure A5.** TMR MIPS Type B Best- and Worst-Case error timing diagram.

The next step is to compute all possible differences ($SD_1$) between save/restore point indices and store word indices as shown in Equation (A4) where $SI_{TMR}^T$ is the transpose of $SI_{TMR}$. This formula states $SD_1$ is a matrix of row vectors such that the $n$th row subtracts each value of $SI_{TMR}$ from the $n$th

$SW_{TMR}$ value. Note that $SW_{TMR}$ is a vector containing the indices of every store word instruction in a program.

$$
\begin{aligned}
&for\ n = 1\ to\ length(SW_{TMR}) \\
&\qquad SD_1(n,:) = SW_{TMR}(n) - SI_{TMR}^T \\
&end
\end{aligned}
\tag{A4}
$$

Next, because some of the values in $SD_1$ may be negative because a save/restore point may occur at the end of one loop and the next store word may occur at the beginning of the next loop, $SD_1$ is modified so that all values are positive as shown in Equation (A5). In this equation, the "$<$" and "$>$" operators are logical operators that populate a matrix with ones or zeros depending on whether the individual matrix entries are less than or greater than the argument to the right of the operator. The term $SD_1 \cdot (SD_1 > 0)$ creates a matrix with all the positive values of $SD_1$ and where all the negative values of $SD_1$ are set to zero (the "$\cdot$" operator denotes element-wise multiplication). The term $SD_1 \cdot (SD_1 < 0)$ creates a matrix with all the negative values of $SD_1$ and where all the positive values of $SD_1$ are set to zero. The term $N_{TMR} \cdot (SD_1 < 0)$ creates a matrix where all the negative values of $SD_1$ are replaced by $N_{TMR}$ and all the positive values of $SD_1$ are set to zero. The term $SD_1 \cdot (SD_1 < 0) + N_{TMR} \cdot (SD_1 < 0)$ creates a matrix where all the negative values of $SD_1$ are replaced by the positive number of instructions from the save/restore point at the end of a loop to the store word instruction at the beginning of the next loop and accounts for the fact that code execution jumped from the end of the loop back to the start of the loop. Finally, $SD_2$ contains all the positive instruction distances between save/restore points and the store words following them.

$$
SD_2 = SD_1 \cdot (SD_1 > 0) + (SD_1 \cdot (SD_1 < 0) + N_{TMR} \cdot (SD_1 < 0))
\tag{A5}
$$

The next step is to determine the minimum distance between a save/restore point creation and a store word instruction. Equation (A6) is used to calculate the minimum distance where *min* is a function that returns the minimum value of each column vector of a matrix in the row vector $a_1$ and returns the index of each minimum value in each column vector in the row vector $b_1$. For a vector, *min* returns the minimum value in $c_1$ and the index of the minimum value in $d_1$. The value $d_1$ is as an index into the columns of $SD_2$ and tells which column contains the minimum distance between a save/restore point and a store word. The value $b_1(d_1)$ is an index into the rows of $SD_2$ and tells which row contains the minimum distance between a save/restore point and a store word. Because the columns of $SD_2$ correspond to the save/restore point indices $SI_{TMR}$ and the rows correspond to the store word indices $SW_{TMR}$, $SI_{TMR}(d_1)$ is the address of the instruction at which the save/restore point is created closest to the store word instruction at the address specified to $SW_{TMR}(b_1(d_1))$. In other words, this is the absolute shortest distance between the creation of a save/restore point and when an error could occur at a store word and constitutes the best-case multiple processor error for TMR MIPS.

$$
\begin{aligned}
[a_1, b_1] &= min(SD_2) \\
[c_1, d_1] &= min(a_1)
\end{aligned}
\tag{A6}
$$

The formula for determining $T_{TMR\ retB\ Best}$ is now presented in Equation (A7). The reason for the if-else statement is because the program is in a loop and $SW_{TMR}(b_1(d_1))$ could be less than $SI_{TMR}(d_1)$.

$$
\begin{aligned}
&if\ SW_{TMR}(b_1(d_1)) \geq SI_{TMR}(d_1) \\
&\qquad T_{TMR\ retB\ Best} = \sum_{n=SI_{TMR}(d_1)}^{SW_{TMR}(b_1(d_1))} t_{I_{TMR}\ n} \\
&else \\
&\qquad T_{TMR\ retB\ Best} = \sum_{n=SI_{TMR}(d_1)}^{n_{TMR\_init}+N_{TMR}} t_{I_{TMR}\ n} + \sum_{n=n_{TMR\_init}+1}^{SW_{TMR}(b_1(d_1))} t_{I_{TMR}\ n} \\
&end
\end{aligned}
\tag{A7}
$$

The definition of the *min* function in Equation (A6) presents an interesting situation when $SW_{TMR}$ or $SI_{TMR}$ is a scalar rather than a vector. In this situation, $SD_2$ will be a vector instead of a matrix and performing the operations in Equation (A6) will not provide usable results for proper indexing into $SW$ and $SI$ in Equation (A7). If $SW_{TMR}$ is a scalar, $b_1$ is used as the indexing variable into $SI_{TMR}$ and no index variable is used for $SW_{TMR}$ because it is a scalar. These adjustments are made to Equation (A7) as shown in Equation (A8). If $SI_{TMR}$ is a scalar, $b_1$ is used as the indexing variable into $SW_{TMR}$ and no index variable is used for $SI_{TMR}$ because it is a scalar. These adjustments are made to Equation (A7) as shown in Equation (A9).

$$
\begin{aligned}
&if\, SW_{TMR} \geq SI_{TMR}(b_1) \\
&\quad T_{TMR\ retB\ Best2} = \sum_{n=SI_{TMR}(b_1)}^{SW_{TMR}} t_{I_{TMR}\ n} \\
\\
&else \\
&\quad T_{TMR\ retB\ Best2} = \sum_{n=SI_{TMR}(b_1)}^{n_{TMR\_init}+N_{TMR}} t_{I_{TMR}\ n} + \sum_{n=n_{TMR\_init}+1}^{SW_{TMR}} t_{I_{TMR}\ n} \\
&end
\end{aligned}
\tag{A8}
$$

$$
\begin{aligned}
&if\, SW_{TMR}(b_1) \geq SI_{TMR} \\
&\quad T_{TMR\ retB\ Best} = \sum_{n=SI_{TMR}}^{SW_{TMR}(b_1)} t_{I_{TMR}\ n} \\
\\
&else \\
&\quad T_{TMR\ retB\ Best} = \sum_{n=SI_{TMR}}^{n_{TMR\_init}+N_{TMR}} t_{I_{TMR}\ n} + \sum_{n=n_{TMR\_init}+1}^{SW_{TMR}(b_1)} t_{I_{TMR}\ n} \\
&end
\end{aligned}
\tag{A9}
$$

The Type B-Worst error occurs at the end of creating a save/restore point so that the error is detected before successful save/restore point creation. The multiple bit error is injected when attempting to write the loop counter when creating the save/restore point such that a multiple processor error is detected and triggers recovery operations. In this scenario, 10,000 instructions and save/restore point creation must be repeated to return to the point in the program at which the error occurred. The Type B Worst-case error is shown in Figure A5.

The runtime for TMR MIPS Type B Worst-case error is given in Equation (A10) where $T_{TMR\ SRP\ Err}$ is the time it takes TMR MIPS to encounter an error during creation of a save/restore point when the error occurs in the loop counter of multiple processors when attempting to save the loop counter to memory, $T_{TMR\ recB}$ is the time to recover from a multiple processor error, $T_{TMR\ retB\ Worst}$ is the time to return to the instruction at which the error occurred. The time $T_{TMR\ SRP\ Err}$ is determined from the simulation, but $T_{TMR\ retB\ Worst}$ is determined by analysis.

$$
\begin{aligned}
T_{TMR\ ErrB\ Worst} &= T_{TMR\ MIPS} + T_{TMR\ SRP\ Err} + \cdots \\
T_{TMR\ recB} &+ T_{TMR\ retB\ Worst}
\end{aligned}
\tag{A10}
$$

To compute the worst-case scenario time to return to the instruction at which the error occurred, the worst-case time between save points must first be determined according to Equation (A11) where $ST_{TMR}$ was previously defined in Equation (A3), $SL_{TMR}(m)$ is the number of full loops completed by the time the $m$th save/restore point creation is reached, $T_{add}$ is the time from the start of the loop in which the save/restore point is created to the instruction in that loop at which the save/restore point is created, $ST_{TMR}(m)$ is the time from the beginning of the program to the time at which the $m$th save/restore point creation begins, $SDT_{TMR}$ is the save time difference between consecutive save points, and $W_{SI_{TMR}}$ is the index of the worst-case $SDT_{TMR}$. The value of $SDT_{TMR}$ is obtained by subtracting the 1st value of $ST_{TMR}$ from the second value, the second value from the third, and so on until the $(n_{SRP} - 1)^{th}$ value is subtracted from the $n_{SRP}^{th}$. The maximum value of $SDT_{TMR}$ is $T_{TMR\ retB\ Worst}$.

$$SDT_{TMR} = ST_{TMR} - [0, ST_{TMR}(1 \ to \ length(ST_{TMR}) - 1)]$$
$$[T_{TMR \ retB \ Worst}, W_{SI_{TMR}}] = max(SDT_{TMR}) \tag{A11}$$

TSR best-case and worst-case errors are similar to TMR Type B-Best and TMR Type B-Worst errors in that they occur immediately after save/restore point error and during save/restore point creation respectively. These errors are shown in Figure A6.

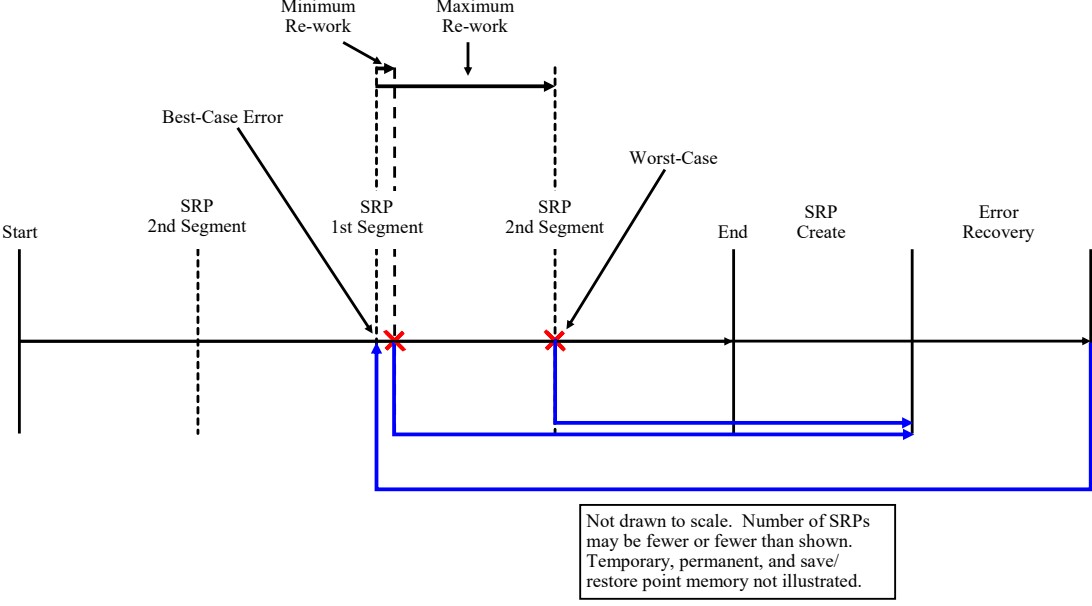

**Figure A6.** TSR MIPS Best- and Worst-Case error timing diagram.

The best-case scenario minimizes the number of instructions which must be executed after error recovery to return to the point in the program at which the error occurred. Therefore, the best-case error occurs immediately after creation of a save/restore point. The error is injected immediately prior to the branch comparison instruction before the first store word instruction after creating a save/restore point. The error is injected into one of the registers to be compared.

The TSR MIPS Best-case error is computed using Equation (A12) where $T_{TSR \ MIPS}$ is the time to complete a program in the absence of an error from the previous work [1], $T_{TSR \ Rec}$ is the time to perform error recovery operations and is determined from simulation results, and $T_{TSR \ Ret}$ is the time needed to return from the most recent save/restore point to the instruction at which the error occurred.

$$T_{TSR \ Best} = T_{TSR \ MIPS} + T_{TSR \ Rec} + T_{TSR \ Ret} \tag{A12}$$

The time to return to the instruction at which the error occurred is determined using Equation (A13) where $n_{TSR \ init}$ is the number of instructions needed to initialize a TSR program (4 instructions) and $SW_{TSR}$ is a vector containing the instruction indices of all store word instructions in a TSR MIPS program.

$$T_{TSR \ ret} = \sum_{n=N_{TSR}-3}^{N_{TSR}} t_{I_{TSR \ n}} + \sum_{n=n_{TSR \ init}+1}^{SW_{TSR}(1)} t_{I_{TSR \ n}} \tag{A13}$$

The TSR MIPS Worst-case scenario maximizes the number of instructions which must be executed after error recovery to return to the point in the program at which the error occurred. Therefore, the worst-case error occurs at the end of creating a save/restore point. This error would specifically target the loop counter, which is the last register to be written to the save/restore point during save/restore point creation. This error would force TSR MIPS to restore itself from the previous save/restore

point and then proceed past the end of the next save/restore point creation, which means completing 250 program loops all over again. Additionally, the worst-case error will occur when creating the save/restore point in the second segment of save/restore point memory rather than the first segment because the second segment takes longer to create.

The TSR MIPS Worst-case error is computed using Equation (A14) where $T_{TSR\ loop}$ is the time to complete a single TSR program loop defined in previous work [1] and $T_{TSR\ SRP1\ Err}$ is the time from the start of save/restore point creation to the time at which an error is detected in the difference between the loop counter and duplicate loop counter. The value of $T_{TSR\ SRP1\ Err}$ is determined from the simulation. The term $\sum_{n=N_{TSR}-3}^{N_{TSR}} t_{I_{TSR\ n}}$ is the time to complete the loop after performing error recovery and the term $250 \cdot T_{TSR\ loop}$ is the time to re-complete the 250 loops between save/restore points.

$$
\begin{aligned}
T_{TSR\ Worst} = T_{TSR\ MIPS} + T_{TSR\ Rec} + \sum_{n=N_{TSR}-3}^{N_{TSR}} t_{I_{TSR\ n}} + \cdots \\
250 \cdot T_{TSR\ loop} + T_{TSR\ SRP1\ Err}
\end{aligned}
\tag{A14}
$$

AHR may experience a TMR or TSR error depending on whether AHR is operating in TMR or TSR mode. TMR errors are further subdivided into early and late errors depending on whether they occur near the beginning of the program or near the TMR to TSR transition point respectively. Early errors have less impact on total program runtime and energy usage than late errors because early errors do not significantly affect the location of the TMR to TSR transition point. This is because the TMR to TSR transition point depends upon completing a predetermined number of instructions without error before transitioning AHR from TMR to TSR mode. Late errors have more impact on total program runtime and energy usage because they cause the TMR to TSR transition point to move towards the end of the program so that more of the program is executed in TMR mode. The result is that program runs faster, but uses more energy than if an early error or no error occurred. Errors encountered when AHR MIPS is operating in TSR mode are virtually identical to the errors encountered by TSR MIPS, but depend upon the point at which the TMR to TSR transition occurs within a program.

When AHR MIPS encounters a TMR Type A error, it handles the error the same way that TMR MIPS would. If the error occurs early in the program, such as at the first store word instruction in the program, the TMR to TSR transition point is only moved by a few instructions as shown in Figure A7. This is referred to as a Type A Early error and it has a minimal impact on the program runtime.

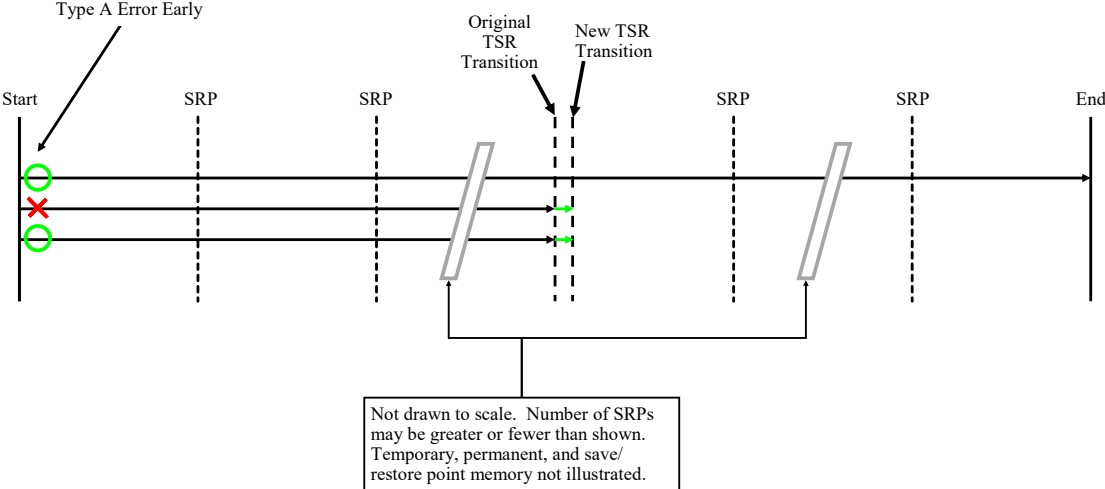

**Figure A7.** AHR MIPS TMR Type A Early error timing diagram.

Equation (A15) shows how to calculate the AHR MIPS Type A Early error timing where $P_{loops\ TMR\ A\ Early}$ is the new transition point loop count determined according to Equation (A16) and $n_{CSRP\ A\ Early}$ is the number of save/restore points to create prior to the transition determined by Equation (A17). The TMR to TSR transition point determines how many save/restore points are

created in TMR and TSR mode. The TMR mode save/restore points are determined by $n_{CSRP\ A\ Early}$, but the number of TSR mode save/restore points depends on where the transition occurs relative to the creation point for the TSR mode save/restore points which only occur at 250, 500, and 750 loops. This is the rationale for the if-else statements in these equations. There is also a possibility that the Type A error may push the TMR to TSR transition point out past the end of the program, in which case, AHR MIPS never enters TSR mode. Note also that $t_{CTMRAE\ TMR}$ and $t_{CTMRAE\ TSR}$ are the time AHR MIPS spends in TMR and TSR mode respectively when encountering a TMR Type A Early error. The time spent in TMR and TSR are separated to make the energy calculations simpler.

$$
\begin{aligned}
&t_{nom\ AE} = t_{TMR\ init} + P_{loops\ TMR\ A\ Early} \cdot T_{TMR\ loop} + \cdots \\
&T_{TMR\ SRP} \cdot n_{CSRP\ A\ Early} \\
&t_{err\ AE} = T_{TMR\ ttdA} + T_{TMR\ recA} + T_{TMR\ retA} + T_{TMR\ repA} \\
&if\ P_{loops\ TMR\ A\ Early}\ <\ 250 \\
&\quad t_{CTMRAE\ TMR} = t_{nom\ AE} + t_{err\ AE} + t_{TMR \to TSR} \\
&\quad t_{CTMRAE\ TSR} = (n_{loops} - P_{loops\ TMR\ A\ Early}) \cdot t_{TSR\ loop} + \cdots \\
&\quad T_{TSR\ SRP0} + 2 \cdot T_{TSR\ SRP1} + T_{TSR\ conc} - T_{TSR\ skip} \\
&elseif\ 250\ \leq\ P_{loops\ TMR\ A\ Early}\ <\ 500 \\
&\quad t_{CTMRAE\ TMR} = t_{nom\ AE} + t_{err\ AE} + t_{TMR \to TSR} \\
&\quad t_{CTMRAE\ TSR} = (n_{loops} - P_{loops\ TMR\ A\ Early}) \cdot t_{TSR\ loop} + \cdots \\
&\quad T_{TSR\ SRP0} + T_{TSR\ SRP1} + T_{TSR\ conc} - T_{TSR\ skip} \\
&elseif\ 500\ \leq\ P_{loops\ TMR\ A\ Early}\ <\ 750 \\
&\quad t_{CTMRAE\ TMR} = t_{nom\ AE} + t_{err\ AE} + t_{TMR \to TSR} \\
&\quad t_{CTMRAE\ TSR} = (n_{loops} - P_{loops\ TMR\ A\ Early}) \cdot t_{TSR\ loop} + \cdots \\
&\quad T_{TSR\ SRP1} + T_{TSR\ conc} - \frac{2}{3} T_{TSR\ skip} \\
&elseif\ 750\ \leq\ P_{loops\ TMR\ A\ Early}\ <\ n_{loops} \\
&\quad t_{CTMRAE\ TMR} = t_{nom\ AE} + t_{err\ AE} + t_{TMR \to TSR} \\
&\quad t_{CTMRAE\ TSR} = (n_{loops} - P_{loops\ TMR\ A\ Early}) \cdot t_{TSR\ loop} + \cdots \\
&\quad T_{TSR\ conc} \\
&elseif\ P_{loops\ TMR\ A\ Early}\ \geq\ n_{loops} \\
&\quad t_{CTMRAE\ TMR} = t_{TMR\ init} + n_{loops} \cdot T_{TMR\ loop} + \cdots \\
&\quad T_{TMR\ SRP} \cdot (n_{SRP} - 1) + t_{err\ AE} \\
&\quad t_{CTMRAE\ TSR} = 0 \\
&end \\
&T_{CTMR\ A\ Early} = t_{CTMRAE\ TMR} + t_{CTMRAE\ TSR}
\end{aligned}
\tag{A15}
$$

$$
P_{loops\ TMR\ A\ Early} = \left\lceil \frac{SW_{TMR}(1) + n_{transition} - n_{TMR\_init}}{N_{TMR}} \right\rceil
\tag{A16}
$$

$$
n_{CSRP\ A\ Early} = \left\lfloor \frac{P_{loops\ TMR\ A\ Early} \cdot N_{TMR} + n_{TMR\_init}}{n_{save}} \right\rfloor
\tag{A17}
$$

If the TMR Type A error occurs late in the program, such as at the last store word instruction before the TMR to TSR transition, the TMR to TSR transition is moved by nearly 15,000 instructions past the point at which it would have occurred if there were no error. This is shown in Figure A8. This is referred to as a Type A Late error and it causes the program to execute more instructions in TMR MIPS and fewer instructions in TSR MIPS than if no error had occurred. The expected effect is a significantly shorter runtime and increased energy usage.

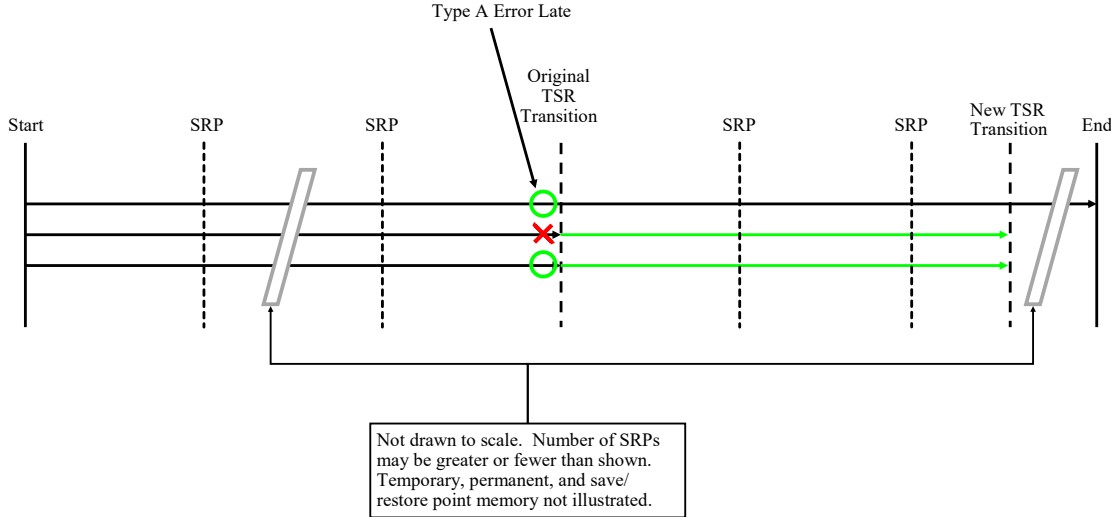

**Figure A8.** AHR MIPS TMR Type A Late error timing diagram.

Equation (A18) shows how to calculate the AHR MIPS Type A Late error timing where $P_{loops\ TMR\ A\ Late}$ is the new transition point loop count determined according to Equation (A19) and $n_{CSRP\ A\ Late}$ is the number of save/restore points to create prior to the transition determined by Equation (A20). Note also that $t_{CTMRAL\ TMR}$ and $t_{CTMRAL\ TSR}$ are the time AHR MIPS spends in TMR and TSR mode respectively when encountering a TMR Type A Late error.

$$
\begin{aligned}
&t_{nom\ AL} = t_{TMR\ init} + P_{loops\ TMR\ A\ Late} \cdot T_{TMR\ loop} + \cdots \\
&T_{TMR\ SRP} \cdot n_{CSRP\ A\ Late} \\
&t_{err\ AL} = T_{TMR\ ttdA} + T_{TMR\ recA} + T_{TMR\ retA} + T_{TMR\ repA} \\
&if\ P_{loops\ TMR\ A\ Late}\ <\ 250 \\
&\quad t_{CTMRAL\ TMR} = t_{nom\ AL} + t_{err\ AL} + t_{TMR \to TSR} \\
&\quad t_{CTMRAL\ TSR} = (n_{loops} - P_{loops\ TMR\ A\ Late}) \cdot t_{TSR\ loop} + \cdots \\
&\quad T_{TSR\ SRP0} + 2 \cdot T_{TSR\ SRP1} + T_{TSR\ conc} - T_{TSR\ skip} \\
&elseif\ 250\ \le\ P_{loops\ TMR\ A\ Late}\ <\ 500 \\
&\quad t_{CTMRAL\ TMR} = t_{nom\ AL} + t_{err\ AL} + t_{TMR \to TSR} \\
&\quad t_{CTMRAL\ TSR} = (n_{loops} - P_{loops\ TMR\ A\ Late}) \cdot t_{TSR\ loop} + \cdots \\
&\quad T_{TSR\ SRP0} + T_{TSR\ SRP1} + T_{TSR\ conc} - T_{TSR\ skip} \\
&elseif\ 500\ \le\ P_{loops\ TMR\ A\ Late}\ <\ 750 \\
&\quad t_{CTMRAL\ TMR} = t_{nom\ AL} + t_{err\ AL} + t_{TMR \to TSR} \\
&\quad t_{CTMRAL\ TSR} = (n_{loops} - P_{loops\ TMR\ A\ Late}) \cdot t_{TSR\ loop} + \cdots \\
&\quad T_{TSR\ SRP1} + T_{TSR\ conc} - \frac{2}{3} T_{TSR\ skip} \\
&elseif\ 750\ \le\ P_{loops\ TMR\ A\ Late}\ <\ n_{loops} \\
&\quad t_{CTMRAL\ TMR} = t_{nom\ AL} + t_{err\ AL} + t_{TMR \to TSR} \\
&\quad t_{CTMRAL\ TSR} = (n_{loops} - P_{loops\ TMR\ A\ Late}) \cdot t_{TSR\ loop} + \cdots \\
&\quad T_{TSR\ conc} \\
&elseif\ P_{loops\ TMR\ A\ Late}\ \ge\ n_{loops} \\
&\quad t_{CTMRAL\ TMR} = t_{TMR\ init} + n_{loops} \cdot T_{TMR\ loop} + \cdots \\
&\quad T_{TMR\ SRP} \cdot (n_{SRP} - 1) + t_{err\ AL} \\
&\quad t_{CTMRAL\ TSR} = 0 \\
&end \\
&T_{CTMR\ A\ Late} = t_{CTMRAL\ TMR} + t_{CTMRAL\ TSR}
\end{aligned}
\tag{A18}
$$

$$
P_{loops\ TMR\ A\ Late} = \left\lceil \frac{SW_{TMR}(length(SW_{TMR})) + n_{transition} - n_{TMR\_init}}{N_{TMR}} \right\rceil
\tag{A19}
$$

$$n_{CSRP\ A\ Late} = \left\lfloor \frac{P_{loops\ TMR\ A\ Late} \cdot N_{TMR} + n_{TMR\_init}}{n_{save}} \right\rfloor \tag{A20}$$

AHR MIPS may also encounter TMR MIPS Type B-Best errors early or late and these are referred to as TMR Type B-Best Early and TMR Type B-Best Late errors. As with the TMR Type A Early error, the TMR Type B-Best Early error has a minimal impact on runtime. As with the TMR Type A Late error, the TMR Type B-Best Late error is expected to significantly decrease runtime and increase energy usage. The TMR Type B-Best Early error is shown in Figure A9 while the TMR Type B-Best Late error is shown in Figure A10.

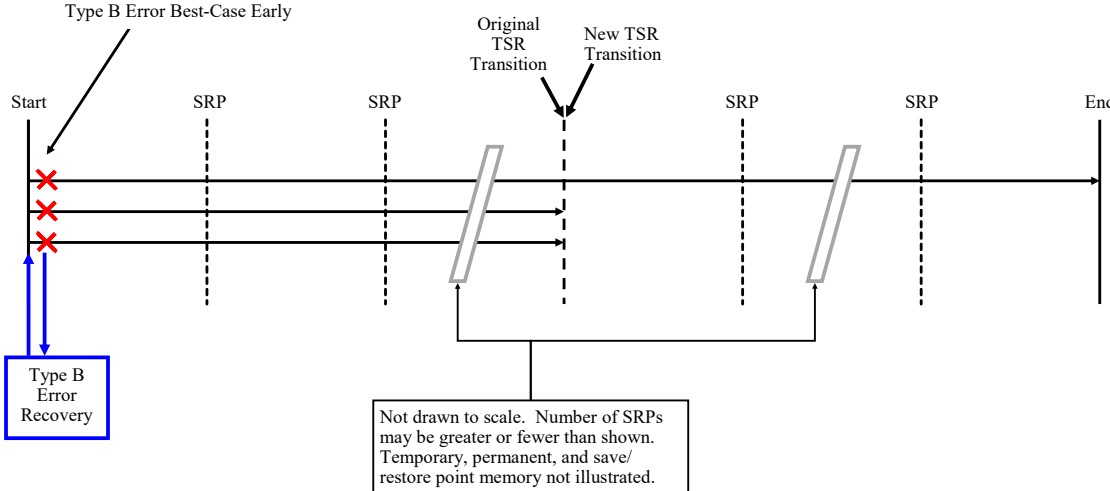

**Figure A9.** AHR MIPS TMR Type B Best-Case Early error timing diagram.

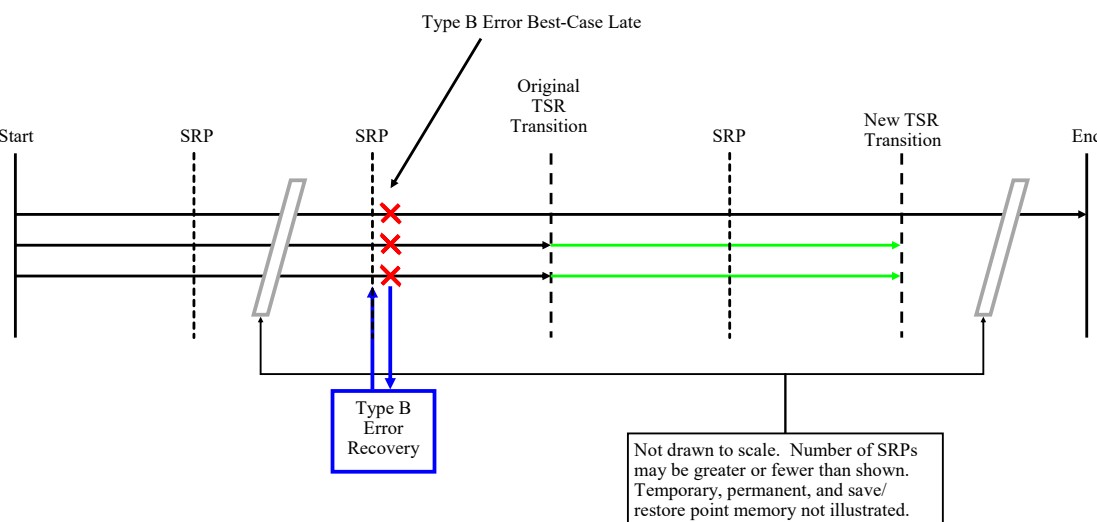

**Figure A10.** AHR MIPS TMR Type B Best-Case Late error timing diagram.

In order to determine the AHR MIPS runtime for programs with Type B-Best Early and Late errors, some of the variables used in computing TMR MIPS runtime for programs with Type B-Best errors need to be modified. The variable $SI_{TMR}$ needs to be modified so it only contains instruction indices of save/restore points that occur before the original TMR to TSR transition was expected to take place. The values of $SL_{TMR}$ and $ST_{TMR}$ must also be updated. These updates are illustrated in Equation (A21) where the $SL_{TMR}(SL_{TMR} < P_{loops})$ returns the vector of $SL_{TMR}$ where the values of $SL_{TMR}$ are less than the TMR to TSR transition point and all other values of the original $SL_{TMR}$ vector are excluded.

$$SL_{CTMR} = SL_{TMR}(SL_{TMR} < P_{loops})$$
$$SI_{CTMR} = SI_{TMR}(1 \text{ to } length(SL_{TMR}))$$
$$ST_{CTMR} = ST_{TMR}(1 \text{ to } length(SL_{TMR}))$$
(A21)

Next, all possible differences between save/restore point indices and store word indices are calculated according to Equation (A22). Note that this is different when compared with Equation (A4) because this formula must account for the fact that an error cannot be allowed to occur after the TMR to TSR transition or it would be a TSR error rather than a TMR Type B Error.

$$
\begin{aligned}
&for\ n = 1\ to\ length(SW_{TMR}) \\
&\quad SD_3(n,:) = SW_{TMR}(n) - SI_{TMR}^T \\
&\quad if\ SL_{CTMR}(length(SL_{CTMR})) = P_{loops} \\
&\quad\quad if\ SD_3(n, length(SL_{CTMR})) < 0 \\
&\quad\quad\quad SD_3(n, length(SL_{CTMR})) = 10^6 \\
&\quad\quad end \\
&\quad end \\
&end
\end{aligned}
$$
(A22)

Then, just as Equation (A5) made all values of $SD_1$ positive, Equation (A23) makes all values of $SD_3$ positive as well.

$$SD_4 = SD_3. \cdot (SD_3 > 0) + (SD_3. \cdot (SD_3 < 0) + N_{TMR} \cdot (SD_3 < 0))$$
(A23)

The next step is to determine which store word indices minimize the difference between each store word and save index. This is computed in Equation (A24). Note that this differs from Equation (A6) because there is no second step to determine the absolute minimum distance. This is because it is desirable to determine the early and late scenarios for a Type B-Best error. The absolute minimum of $SD_4$ might not minimize or maximize the number of instructions computed in TMR mode. Instead, each possible combination of minimum distance from a save index to a store word index is evaluated for total program completion time. The total program completion time for each scenario is then evaluated against the completion times to determine which is slowest (Early) and which is fastest (Late).

$$[a_2, b_2] = min(SD_4)$$
(A24)

Equations (A25)–(A27) show how to compute the time to complete each program for each possible combination of minimum distance from a save index to a store word index. (Equation (A27) is a continuation of Equation (A26) because the entire equation could not fit on one page.) Equation (A26) (and Equation (A27)) also shows that the Type B-Best Early solution is the maximum of these times and the Type B-Best Late solution is the minimum of these times. The *Flag* variable is used to keep track of whether a particular combination of save index and store word index is allowed. The flag is 1 if the combination is not allowed because the store word following the save index would occur after the TMR to TSR transition. The variable $P_{loops\ TMR\ B\ Best}(n)$ is the new TMR to TSR transition point based on the error location for the $n$th save index. The variable $n_{CSRP\ B\ Best}(n)$ is the new number of save/restore points to create for the $n$th save index. The variable $T_{add}(n)$ is the amount of time required to return from the save index to the store word index at which the error occurred for the $n$th save index. The time to complete the TMR portion of the program for the $n$th save index is $t_{CTMRBB\ TMR}(n)$. The time to complete the TSR portion of the program for the $n$th save index is $t_{CTMRBB\ TSR}(n)$. The value $NaN$ is assigned to $t_{CTMRBB\ TMR}(n)$ and $t_{CTMRBB\ TSR}(n)$ when $Flag = 1$ because the $max$ and $min$ functions ignore $NaN$ values and return only numerical values. Finally, $t_{CTMRBBE\ TMR}(n)$, $t_{CTMRBBE\ TSR}(n)$ are the time AHR MIPS spends in TMR and TSR mode when a TMR Type B-Best Early error is encountered.

Similarly, $t_{CTMRBBL\ TMR}(n)$, $t_{CTMRBBL\ TSR}(n)$ are the time AHR MIPS spends in TMR and TSR mode when a TMR Type B-Best Late error is encountered.

$$
\begin{aligned}
&for\ n = 1\ to\ length(b_2)\\
&\quad Flag = 0\\
&\quad if\ SI_{CTMR} = 1\\
&\quad\quad P_{loops\ TMR\ B\ Best}(n) = P_{loops}\\
&\quad else\\
&\quad\quad P_{loops\ TMR\ B\ Best}(n) = \cdots\\
&\quad\quad \left\lceil \frac{SI_{CTMR}(n) + SL_{CTMR} \cdot N_{TMR} + n_{transition} - n_{TMR\_init}}{N_{TMR}} \right\rceil\\
&\quad end\\
&\quad n_{CSRP\ B\ Best}(n) = \left\lfloor \frac{P_{loops\ TMR\ B\ Best}(n) \cdot N_{TMR} + n_{TMR\_init}}{n_{save}} \right\rfloor\\
&\quad if SI_{CTMR}(n) \leq SW_{CTMR}(b2(n)) - 1\\
&\quad\quad T_{add}(n) = \sum_{m=SI_{CTMR}(n)}^{SW_{CTMR}(b2(n))-1} t_{I_{TMR\ m}}\\
&\quad elseif\ SL_{CTMR}(n) < P_{loops\ TMR\ B\ Best}(n)\\
&\quad\quad T_{add}(n) = \sum_{m=SI_{CTMR}(n)}^{n_{TMR\_init}+N_{TMR}} t_{I_{TMR\ m}} + \sum_{n_{TMR\_init}+1}^{SW_{CTMR}(b2(n))-1} t_{I_{TMR\ m}}\\
&\quad else\\
&\quad\quad T_{add}(n) = 0\\
&\quad\quad Flag = 1\\
&\quad end\\
&end
\end{aligned}
\tag{A25}
$$

$$
\begin{aligned}
&for\ n = 1\ to\ length(b_2)\\
&\quad if\ Flag = 1\\
&\quad\quad t_{CTMRBB\ TMR}(n) = NaN\\
&\quad\quad t_{CTMRBB\ TSR}(n) = NaN\\
&\quad else\\
&\quad\quad t_{nom\ BB} = t_{TMR\ init} + P_{loops\ TMR\ B\ Best}(n) \cdot T_{TMR\ loop} + \cdots\\
&\quad\quad T_{TMR\ SRP} \cdot n_{CSRP\ B\ Best}(n)\\
&\quad\quad t_{err\ BB} = T_{TMR\ ttdB} + T_{TMR\ recB} + T_{add}(n)\\
&\quad\quad if\ P_{loops\ TMR\ B\ Best}(n) < 250\\
&\quad\quad\quad t_{CTMRBB\ TMR}(n) = t_{nom\ BB} + t_{err\ BB} + t_{TMR \to TSR}\\
&\quad\quad\quad t_{CTMRBB\ TSR}(n) = (n_{loops} - P_{loops\ TMR\ B\ Best}(n)) \cdot t_{TSR\ loop} + \cdots\\
&\quad\quad\quad T_{TSR\ SRP0} + 2 \cdot T_{TSR\ SRP1} + T_{TSR\ conc} - T_{TSR\ skip}\\
&\quad\quad elseif\ 250 \leq P_{loops\ TMR\ B\ Best}(n) < 500\\
&\quad\quad\quad t_{CTMRBB\ TMR}(n) = t_{nom\ BB} + t_{err\ BB} + t_{TMR \to TSR}\\
&\quad\quad\quad t_{CTMRBB\ TSR} = (n_{loops} - P_{loops\ TMR\ B\ Best}(n)) \cdot t_{TSR\ loop} + \cdots\\
&\quad\quad\quad T_{TSR\ SRP0} + T_{TSR\ SRP1} + T_{TSR\ conc} - T_{TSR\ skip}\\
&\quad\quad elseif\ 500 \leq P_{loops\ TMR\ B\ Best}(n) < 750\\
&\quad\quad\quad t_{CTMRBB\ TMR}(n) = t_{nom\ BB} + t_{err\ BB} + t_{TMR \to TSR}\\
&\quad\quad\quad t_{CTMRBB\ TSR} = (n_{loops} - P_{loops\ TMR\ B\ Best}(n)) \cdot t_{TSR\ loop} + \cdots\\
&\quad\quad\quad T_{TSR\ SRP1} + T_{TSR\ conc} - \frac{2}{3} T_{TSR\ skip}
\end{aligned}
\tag{A26}
$$

$$elseif \ 750 \ \leq \ P_{loops \ TMR \ B \ Best}(n) \ < \ n_{loops}$$
$$\quad t_{CTMRBB \ TMR}(n) = t_{nom \ BB} + t_{err \ BB} + t_{TMR \rightarrow TSR}$$
$$\quad t_{CTMRBB \ TSR} = (n_{loops} - P_{loops \ TMR \ B \ Best}(n)) \cdot t_{TSR \ loop} + T_{TSR \ conc}$$
$$elseif \ P_{loops \ TMR \ B \ Best}(n) \ \geq \ n_{loops}$$
$$\quad t_{CTMRBB \ TMR}(n) = t_{TMR \ init} + n_{loops} \cdot T_{TMR \ loop} + \cdots$$
$$\quad T_{TMR \ SRP} \cdot (n_{SRP} - 1) + t_{err \ BB}$$
$$\quad t_{CTMRBB \ TSR} = 0$$
$$\quad\quad end$$
$$\quad end$$
$$end$$
$$[T_{CTMR \ B \ Best \ Early}, b_3] = max(t_{CTMRBB \ TMR} + t_{CTMRBB \ TSR})$$
$$t_{CTMRBBE \ TMR} = t_{CTMRBB \ TMR}(b_3)$$
$$t_{CTMRBBE \ TSR} = t_{CTMRBB \ TSR}(b_3)$$
$$[T_{CTMR \ B \ Best \ Late}, b_4] = min(t_{CTMRBB \ TMR} + t_{CTMRBB \ TSR})$$
$$t_{CTMRBBL \ TMR} = t_{CTMRBB \ TMR}(b_4)$$
$$t_{CTMRBBL \ TSR} = t_{CTMRBB \ TSR}(b_4)$$

(A27)

Remembering that the *min* function used in Equation (A24) is defined and used in the same manner as in Equation (A6), the same problem with $SD_2$ possibly being a vector arises for $SD_4$ as well. This affects the indices used in Equations (A25) and (A26). If $SW_{TMR}$ is a scalar, Equation (A25) is rewritten in Equation (A28). If $SI_{CTMR}$ is a scalar, these equations are rewritten in Equations (A29)–(A31) where Equation (A31) is a continuation of Equation (A30).

$$for \ n = 1 \ to \ length(b_2)$$
$$\quad Flag = 0$$
$$\quad if \ SI_{CTMR} = 1$$
$$\quad\quad P_{loops \ TMR \ B \ Best}(n) = P_{loops}$$
$$\quad else$$
$$\quad\quad P_{loops \ TMR \ B \ Best}(n) = \cdots$$
$$\quad\quad \left\lceil \frac{SI_{CTMR}(n) + SL_{CTMR} \cdot N_{TMR} + n_{transition} - n_{TMR\_init}}{N_{TMR}} \right\rceil$$
$$\quad end$$
$$n_{CSRP \ B \ Best}(n) = \left\lfloor \frac{P_{loops \ TMR \ B \ Best}(n) \cdot N_{TMR} + n_{TMR\_init}}{n_{save}} \right\rfloor$$
$$if \ SI_{CTMR}(n) \leq SW_{CTMR} - 1$$
$$\quad T_{add}(n) = \sum_{m=SI_{CTMR}(n)}^{SW_{CTMR}-1} t_{I_{TMR \ m}}$$
$$elseif \ SL_{CTMR}(n) < P_{loops \ TMR \ B \ Best}(n)$$
$$\quad T_{add}(n) = \sum_{m=SI_{CTMR}(n)}^{n_{TMR\_init}+N_{TMR}} t_{I_{TMR \ m}} + \sum_{n_{TMR\_init}+1}^{SW_{CTMR}-1} t_{I_{TMR \ m}}$$
$$else$$
$$\quad T_{add}(n) = 0$$
$$\quad Flag = 1$$
$$end$$
$$end$$

(A28)

$$Flag = 0$$
$$if\ SI_{CTMR} = 1$$
$$\quad P_{loops\ TMR\ B\ Best} = P_{loops}$$
$$else$$
$$\quad P_{loops\ TMR\ B\ Best} = \cdots$$
$$\quad \left\lceil \frac{SI_{CTMR} + SL_{CTMR} \cdot N_{TMR} + n_{transition} - n_{TMR\_init}}{N_{TMR}} \right\rceil$$
$$end$$

$$n_{CSRP\ B\ Best} = \left\lfloor \frac{P_{loops\ TMR\ B\ Best} \cdot N_{TMR} + n_{TMR\_init}}{n_{save}} \right\rfloor \qquad \text{(A29)}$$
$$if SI_{CTMR} \le SW_{CTMR}(b2) - 1$$
$$\quad T_{add} = \sum_{m=SI_{CTMR}}^{SW_{CTMR}(b2)-1} t_{I_{TMR\ m}}$$
$$elseif\ SL_{CTMR}(n) < P_{loops\ TMR\ B\ Best}(n)$$
$$\quad T_{add} = \sum_{m=SI_{CTMR}}^{n_{TMR\_init}+N_{TMR}} t_{I_{TMR\ m}} + \sum_{n_{TMR\_init}+1}^{SW_{CTMR}(b2)-1} t_{I_{TMR\ m}}$$
$$else$$
$$\quad T_{add} = 0$$
$$\quad Flag = 1$$
$$end$$

$$if\ Flag = 1$$
$$\quad t_{CTRMBB\ TMR} = NaN$$
$$\quad t_{CTRMBB\ TSR} = NaN$$
$$else$$
$$\quad\quad t_{nom\ BB} = t_{TMR\ init} + P_{loops\ TMR\ B\ Best}(n) \cdot T_{TMR\ loop} + \cdots$$
$$\quad\quad T_{TMR\ SRP} \cdot n_{CSRP\ B\ Best}(n)$$
$$\quad\quad t_{err\ BB} = T_{TMR\ ttdB} + T_{TMR\ recB} + T_{add}(n)$$
$$\quad if\ P_{loops\ TMR\ B\ Best} < 250$$
$$\quad\quad t_{CTRMBB\ TMR} = t_{nom\ BB} + t_{err\ BB} + t_{TMR \to TSR}$$
$$\quad\quad t_{CTRMBB\ TSR} = (n_{loops} - P_{loops\ TMR\ B\ Best}) \cdot t_{TSR\ loop} + \cdots$$
$$\quad\quad T_{TSR\ SRP0} + 2 \cdot T_{TSR\ SRP1} + T_{TSR\ conc} - T_{TSR\ skip}$$
$$\quad elseif\ 250 \le P_{loops\ TMR\ B\ Best} < 500 \qquad \text{(A30)}$$
$$\quad\quad t_{CTRMBB\ TMR} = t_{nom\ BB} + t_{err\ BB} + t_{TMR \to TSR}$$
$$\quad\quad t_{CTRMBB\ TSR} = (n_{loops} - P_{loops\ TMR\ B\ Best}) \cdot t_{TSR\ loop} + \cdots$$
$$\quad\quad T_{TSR\ SRP0} + T_{TSR\ SRP1} + T_{TSR\ conc} - T_{TSR\ skip}$$
$$\quad elseif\ 500 \le P_{loops\ TMR\ B\ Best} < 750$$
$$\quad\quad t_{CTRMBB\ TMR} = t_{nom\ BB} + t_{err\ BB} + t_{TMR \to TSR}$$
$$\quad\quad t_{CTRMBB\ TSR} = (n_{loops} - P_{loops\ TMR\ B\ Best}) \cdot t_{TSR\ loop} + \cdots$$
$$\quad\quad T_{TSR\ SRP1} + T_{TSR\ conc} - \frac{2}{3} T_{TSR\ skip}$$
$$\quad elseif\ 750 \le P_{loops\ TMR\ B\ Best} < n_{loops}$$
$$\quad\quad t_{CTRMBB\ TMR} = t_{nom\ BB} + t_{err\ BB} + t_{TMR \to TSR}$$
$$\quad\quad t_{CTRMBB\ TSR} = (n_{loops} - P_{loops\ TMR\ B\ Best}) \cdot t_{TSR\ loop} + \cdots$$
$$\quad\quad T_{TSR\ conc}$$

$$elseif\ P_{loops\ TMR\ B\ Best}\ \geq\ n_{loops}$$
$$t_{CTRMBB\ TMR} = t_{TMR\ init} + n_{loops} \cdot T_{TMR\ loop} + \cdots$$
$$T_{TMR\ SRP} \cdot (n_{SRP} - 1) + t_{err\ BB}$$
$$t_{CTRMBB\ TSR} = 0$$
$$end$$
$$end$$
$$T_{CTMR\ B\ Best\ Early} = t_{CTRMBB\ TMR} + t_{CTRMBB\ TSR}$$
$$t_{CTRMBBE\ TMR} = t_{CTRMBB\ TMR}$$
$$t_{CTRMBBE\ TSR} = t_{CTRMBB\ TSR}$$
$$T_{CTMR\ B\ Best\ Late} = t_{CTRMBB\ TMR} + t_{CTRMBB\ TSR}$$
$$t_{CTRMBBL\ TMR} = t_{CTRMBB\ TMR}$$
$$t_{CTRMBBL\ TSR} = t_{CTRMBB\ TSR}$$

(A31)

AHR MIPS may also encounter TMR MIPS Type B-Worst errors early or late and these are referred to as TMR Type B-Worst Early and TMR Type B-Worst Late errors. As with the TMR Type A Early error, the TMR Type B-Worst Early error has a minimal impact on runtime. As with the TMR Type A Late error, the TMR Type B-Worst Late error is expected to significantly decrease runtime and increase energy usage. The TMR Type B-Worst Early error is shown in Figure A11 while the TMR Type B-Worst Late error is shown in Figure A12.

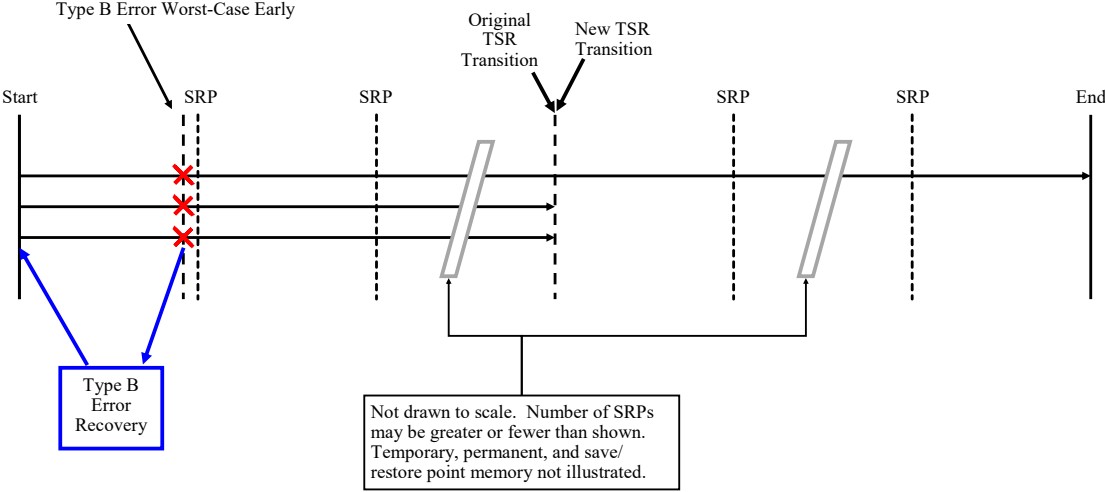

**Figure A11.** AHR MIPS TMR Type B Worst-Case Early error timing diagram.

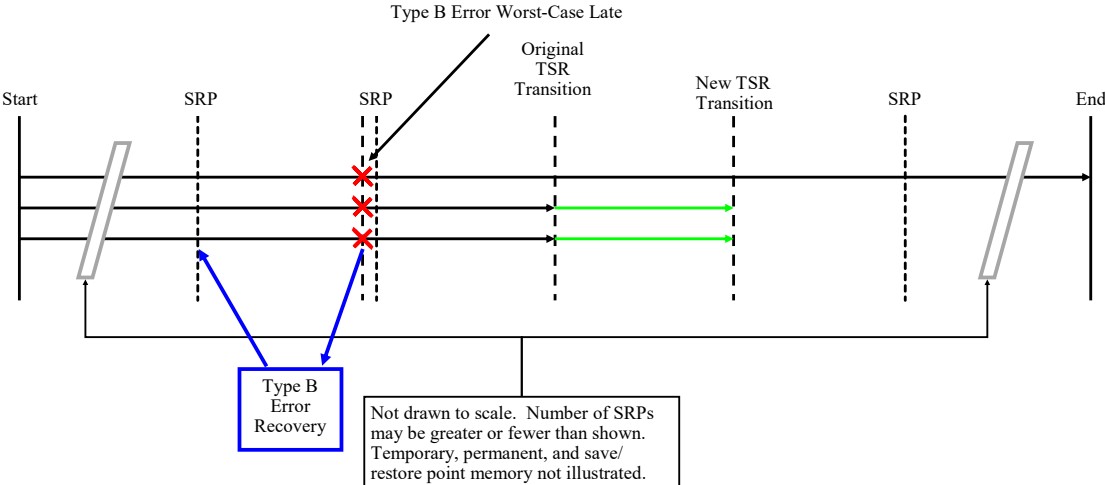

**Figure A12.** AHR MIPS TMR Type B Worst-Case Late error timing diagram.

Equation (A32) shows how to compute the time to complete a AHR MIPS program with a TMR Type B-Worst Early error where $P_{loops\ TMR\ B\ Worst\ Early}$ is the number of loops at which the transition point occurs when accounting for the error, $n_{CSRP\ B\ Worst\ Early}$ is the number of save/restore points to create in TMR MIPS when accounting for the error, and $T_{CTMR\ retB\ Worst\ Early}$ is the time needed to return to the point at which the error occurred after recovering from the error. Note that $t_{CTMRBWE\ TMR}$ and $t_{CTMRBWE\ TSR}$ are the time AHR MIPS spends in TMR and TSR mode respectively when encountering a TMR Type B-Worst Early error.

$$
\begin{aligned}
&t_{nom\ BWE} = t_{TMR\ init} + P_{loops\ TMR\ B\ Worst\ Early} \cdot T_{TMR\ loop} + \cdots \\
&T_{TMR\ SRP} \cdot n_{CSRP\ B\ Worst\ Early} \\
&t_{err\ BWE} = T_{TMR\ SRP\ Err} + T_{TMR\ recB} + T_{CTMR\ retB\ Worst\ Early} \\
&if\ P_{loops\ TMR\ B\ Worst\ Early}\ <\ 250 \\
&\quad t_{CTMRBWE\ TMR} = t_{nom\ BWE} + t_{err\ BWE} + t_{TMR \rightarrow TSR} \\
&\quad t_{CTMRBWE\ TSR} = (n_{loops} - P_{loops\ TMR\ B\ Worst\ Early}) \cdot t_{TSR\ loop} + \cdots \\
&\quad T_{TSR\ SRP0} + 2 \cdot T_{TSR\ SRP1} + T_{TSR\ conc} - T_{TSR\ skip} \\
&elseif\ 250\ \leq\ P_{loops\ TMR\ B\ Worst\ Early}\ <\ 500 \\
&\quad t_{CTMRBWE\ TMR} = t_{nom\ BWE} + t_{err\ BWE} + t_{TMR \rightarrow TSR} \\
&\quad t_{CTMRBWE\ TSR} = (n_{loops} - P_{loops\ TMR\ B\ Worst\ Early}) \cdot t_{TSR\ loop} + \cdots \\
&\quad T_{TSR\ SRP0} + T_{TSR\ SRP1} + T_{TSR\ conc} - T_{TSR\ skip} \\
&elseif\ 500\ \leq\ P_{loops\ TMR\ B\ Worst\ Early}\ <\ 750 \\
&\quad t_{CTMRBWE\ TMR} = t_{nom\ BWE} + t_{err\ BWE} + t_{TMR \rightarrow TSR} \\
&\quad t_{CTMRBWE\ TSR} = (n_{loops} - P_{loops\ TMR\ B\ Worst\ Early}) \cdot t_{TSR\ loop} + \cdots \\
&\quad T_{TSR\ SRP1} + T_{TSR\ conc} - \frac{2}{3}T_{TSR\ skip} \\
&elseif\ 750\ \leq\ P_{loops\ TMR\ B\ Worst\ Early}\ <\ n_{loops} \\
&\quad t_{CTMRBWE\ TMR} = t_{nom\ BWE} + t_{err\ BWE} + t_{TMR \rightarrow TSR} \\
&\quad t_{CTMRBWE\ TSR} = (n_{loops} - P_{loops\ TMR\ B\ Worst\ Early}) \cdot t_{TSR\ loop} + T_{TSR\ conc} \\
&elseif\ P_{loops\ TMR\ B\ Worst\ Early}\ \geq\ n_{loops} \\
&\quad t_{CTMRBWE\ TMR} = t_{TMR\ init} + n_{loops} \cdot T_{TMR\ loop} + \cdots \\
&\quad T_{TMR\ SRP} \cdot (n_{SRP} - 1) + t_{err\ BWE} \quad t_{CTMRBWE\ TSR} = 0 \\
&end \\
&T_{CTMR\ B\ Worst\ Early} = t_{CTMRBWE\ TMR} + t_{CTMRBWE\ TSR}
\end{aligned}
\tag{A32}
$$

The time $T_{CTMR\ retB\ Worst\ Early}$ is computed according to Equation (A33) where $SDT_{CTMR}$ is the save time difference between consecutive save points and $W_{SI_{CTMR}}$ is the index of the worst-case $SDT_{CTMR}$. This is nearly identical to Equation (A11).

$$
\begin{aligned}
SDT_{CTMR} &= ST_{CTMR} - [0, ST_{CTMR}(1\ to\ length(ST_{CTMR}) - 1)] \\
T_{CTMR\ retB\ Worst\ Early} &= SDT_{CTMR}(2)
\end{aligned}
\tag{A33}
$$

Next, the loop count at which the TMR to TSR transition will occur after encountering an error is determined using Equation (A34).

$$
\begin{aligned}
&if\ SL_{CTMR}(1) = 0 \\
&\quad P_{loops\ TMR\ B\ Worst\ Early} = P_{loops} \\
&else \\
&\quad P_{loops\ TMR\ B\ Worst\ Early} = \cdots \\
&\left\lceil \frac{SI_{CTMR}(1) + SL_{CTMR}(1) \cdot N_{TMR} + n_{transition} - n_{TMR\_init}}{N_{TMR}} \right\rceil \\
&end
\end{aligned}
\tag{A34}
$$

Finally, $n_{CSRP\ B\ Worst\ Early}$ is determined according to Equation (A35).

$$n_{CSRP\ B\ Worst\ Early} = \left\lfloor \frac{P_{loops\ TMR\ B\ Worst\ Early} \cdot N_{TMR} + n_{TMR\_init}}{n_{save}} \right\rfloor \tag{A35}$$

Equation (A36) shows how to compute the time to complete a AHR MIPS program with a TMR Type B-Worst Late error where $P_{loops\ TMR\ B\ Worst\ Late}$ is the number of loops at which the transition point occurs when accounting for the error, $n_{CSRP\ B\ Worst\ Late}$ is the number of save/restore points to create in TMR MIPS when accounting for the error, and $T_{CTMR\ retB\ Worst\ Late}$ is the time needed to return to the point at which the error occurred after recovering from the error. Note that $t_{CTMRBWL\ TMR}$ and $t_{CTMRBWL\ TSR}$ are the time AHR MIPS spends in TMR and TSR mode respectively when encountering a TMR Type B-Worst Late error.

$$
\begin{aligned}
&t_{nom\ BWL} = t_{TMR\ init} + P_{loops\ TMR\ B\ Worst\ Late} \cdot T_{TMR\ loop} + \cdots \\
&T_{TMR\ SRP} \cdot n_{CSRP\ B\ Worst\ Late} \\
&t_{err\ BWL} = T_{TMR\ SRP\ Err} + T_{TMR\ recB} + T_{CTMR\ retB\ Worst\ Early} \\
&if\ P_{loops\ TMR\ B\ Worst\ Late}\ <\ 250 \\
&\quad t_{CTMRBWL\ TMR} = t_{nom\ BWE} + t_{err\ BWE} + t_{TMR \rightarrow TSR} \\
&\quad t_{CTMRBWL\ TSR} = (n_{loops} - P_{loops\ TMR\ B\ Worst\ Late}) \cdot t_{TSR\ loop} + \cdots \\
&\quad T_{TSR\ SRP0} + 2 \cdot T_{TSR\ SRP1} + T_{TSR\ conc} - T_{TSR\ skip} \\
&elseif\ 250\ \leq\ P_{loops\ TMR\ B\ Worst\ Late}\ <\ 500 \\
&\quad t_{CTMRBWL\ TMR} = t_{nom\ BWE} + t_{err\ BWE} + t_{TMR \rightarrow TSR} \\
&\quad t_{CTMRBWL\ TSR} = (n_{loops} - P_{loops\ TMR\ B\ Worst\ Late}) \cdot t_{TSR\ loop} + \cdots \\
&\quad T_{TSR\ SRP0} + T_{TSR\ SRP1} + T_{TSR\ conc} - T_{TSR\ skip} \\
&elseif\ 500\ \leq\ P_{loops\ TMR\ B\ Worst\ Late}\ <\ 750 \\
&\quad t_{CTMRBWL\ TMR} = t_{nom\ BWE} + t_{err\ BWE} + t_{TMR \rightarrow TSR} \\
&\quad t_{CTMRBWL\ TSR} = (n_{loops} - P_{loops\ TMR\ B\ Worst\ Late}) \cdot t_{TSR\ loop} + \cdots \\
&\quad T_{TSR\ SRP1} + T_{TSR\ conc} - \frac{2}{3} T_{TSR\ skip} \\
&elseif\ 750\ \leq\ P_{loops\ TMR\ B\ Worst\ Late}\ <\ n_{loops} \\
&\quad t_{CTMRBWL\ TMR} = t_{nom\ BWE} + t_{err\ BWE} + t_{TMR \rightarrow TSR} \\
&\quad t_{CTMRBWL\ TSR} = (n_{loops} - P_{loops\ TMR\ B\ Worst\ Late}) \cdot t_{TSR\ loop} + T_{TSR\ conc} \\
&elseif\ P_{loops\ TMR\ B\ Worst\ Late}\ \geq\ n_{loops} \\
&\quad t_{CTMRBWL\ TMR} = t_{TMR\ init} + n_{loops} \cdot T_{TMR\ loop} + \cdots \\
&\quad T_{TMR\ SRP} \cdot (n_{SRP} - 1) + t_{err\ BWE} \\
&\quad t_{CTMRBWL\ TSR} = 0 \\
&end \\
&T_{CTMR\ B\ Worst\ Late} = t_{CTMRBWL\ TMR} + t_{CTMRBWL\ TSR}
\end{aligned}
\tag{A36}
$$

The time $T_{CTMR\ retB\ Worst\ Late}$ is computed according to Equation (A37). This is nearly identical to Equation (A11).

$$T_{CTMR\ retB\ Worst\ Late} = SDT_{CTMR}(length(SDT_{CTMR})) \tag{A37}$$

Next, the loop count at which the TMR to TSR transition will occur after encountering an error is determined using Equation (A38).

$$\begin{aligned}
&if\ SL_{CTMR}(length(SDT_{CTMR}) - 1) = 0\\
&\quad P_{loops\ TMR\ B\ Worst\ Late} = P_{loops}\\
&else\\
&\quad P_{loops\ TMR\ B\ Worst\ Late} = \cdots\\
&\left\lceil \left( SI_{CTMR}(length(SDT_{CTMR}) - 1) + \cdots \right.\right.\\
&SL_{CTMR}(length(SDT_{CTMR}) - 1) \cdot N_{TMR} + \cdots\\
&\left.\left. n_{transition} - n_{TMR\_init} \right) \middle/ N_{TMR} \right\rceil\\
&end
\end{aligned}$$
(A38)

Finally, $n_{CSRP\ B\ Worst\ Late}$ is determined according to Equation (A39).

$$n_{CSRP\ B\ Worst\ Late} = \left\lfloor \frac{P_{loops\ TMR\ B\ Worst\ Late} \cdot N_{TMR} + n_{TMR\_init}}{n_{save}} \right\rfloor$$
(A39)

In contrast to the TMR errors which can affect the TMR to TSR transition point, TSR errors do not affect the transition point; however, TSR worst-case errors may be affected by the transition point. The best-case errors are unaffected by the transition point.

When AHR MIPS encounters a TSR Best-case error, it encounters it immediately after the creation of a save/restore point. This could be the save/restore point created by the transition from TMR to TSR, or any of the save/restore points created by TSR MIPS after AHR MIPS enters TSR mode. Regardless of where the which save/restore point the TSR Best-case error occurs after, the recovery time is always the same. This is because of the way the TSR MIPS Best-case error was defined to be injected immediately prior to the branch comparison instruction before the first store word instruction after creating a save/restore point. A few examples of AHR MIPS TSR Best-case errors are shown in Figures A13–A16 where the transition occurs before the first, second, or third TSR save/restore creation point or after the third TSR save/restore creation point respectively.

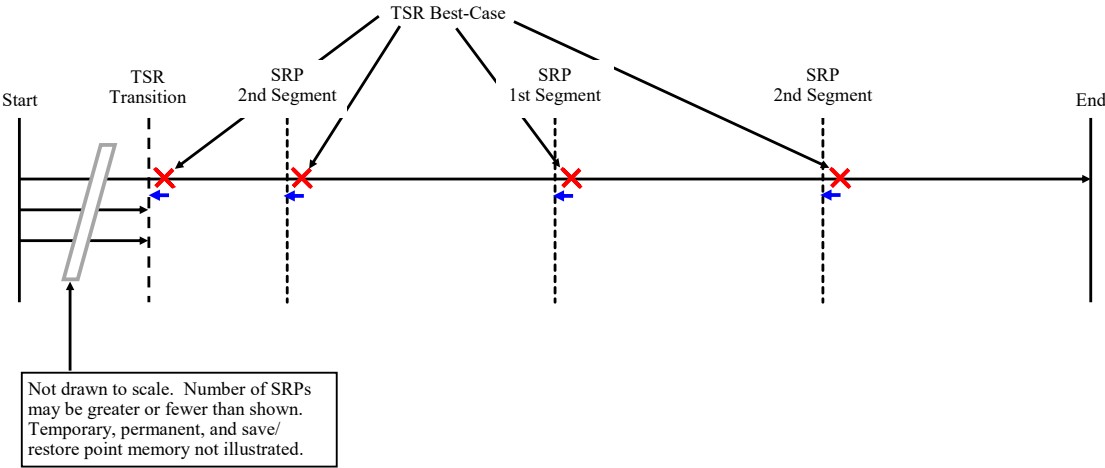

**Figure A13.** AHR MIPS TSR Best-Case Early Error Timing Diagram 1.

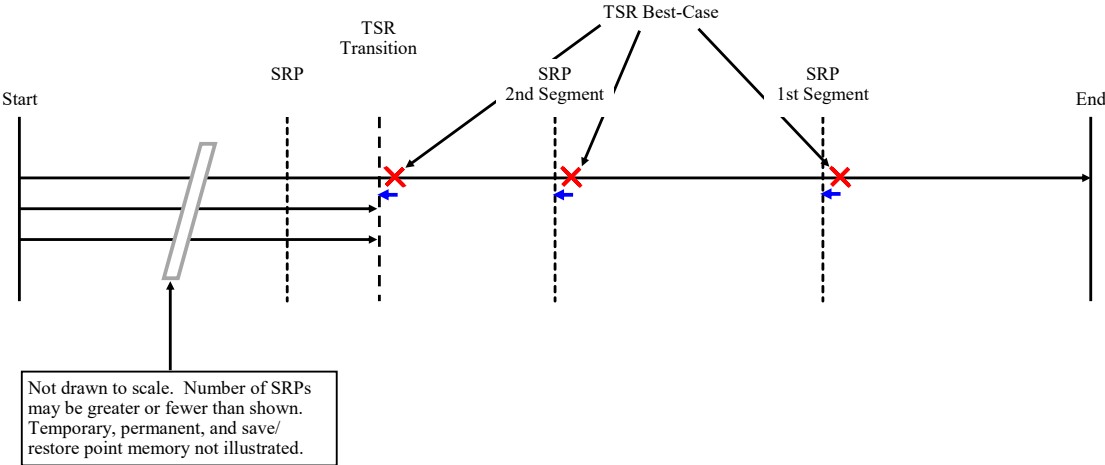

**Figure A14.** AHR MIPS TSR Best-Case Early Error Timing Diagram 2.

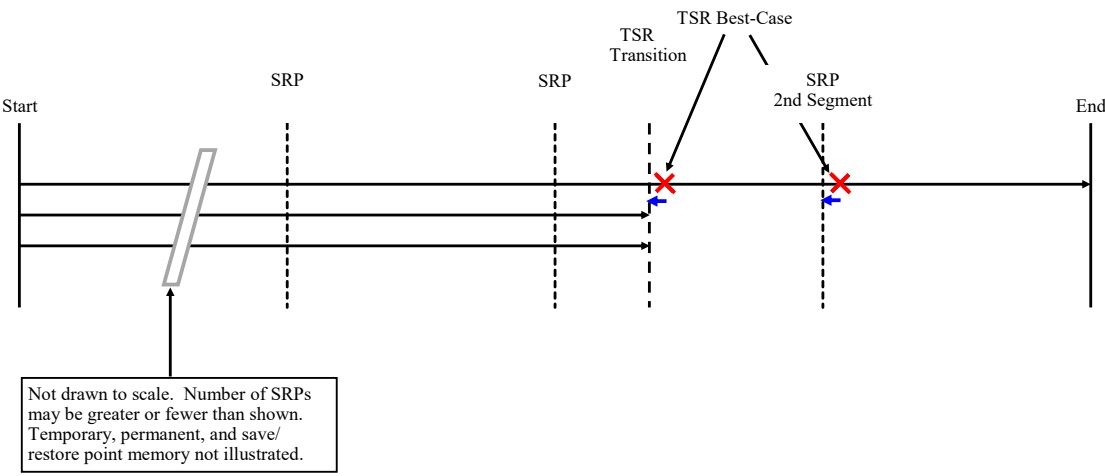

**Figure A15.** AHR MIPS TSR Best-Case Early Error Timing Diagram 3.

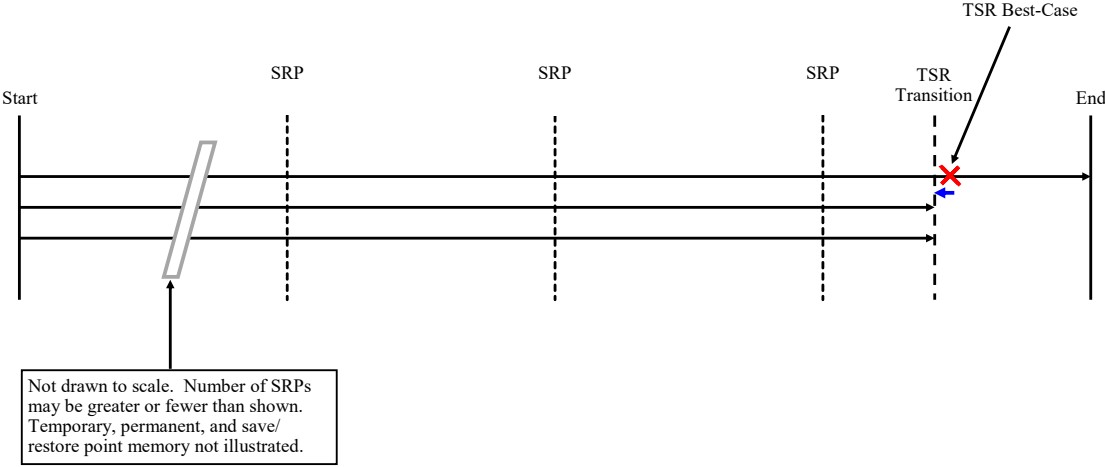

**Figure A16.** AHR MIPS TSR Best-Case Early Error Timing Diagram 4.

The time needed to complete a AHR MIPS program experiencing a TSR Best-case error is given in Equation (A40) where $T_{TSR\ Rec}$ and $T_{TSR\ ret}$ were previously defined in Equation (A12).

$$
\begin{aligned}
T_{CTSR\ Best} &= T_{AHR\ MIPS} + T_{TSR\ Rec} + T_{TSR\ ret} \\
T_{CTSR\ Best} &= t_{AHR\ TMR} + t_{AHR\ TSR} + T_{TSR\ Rec} + T_{TSR\ ret} \\
t_{CTSRB\ TMR} &= t_{AHR\ TMR} \\
t_{CTSRB\ TSR} &= t_{AHR\ TSR} + T_{TSR\ Rec} + T_{TSR\ ret} \\
T_{CTSR\ Best} &= t_{CTSRB\ TMR} + t_{CTSRB\ TSR}
\end{aligned}
\tag{A40}
$$

TSR Worst-case errors in AHR MIPS require special attention. While TSR Worst-case errors in TSR MIPS take place at the end of creating a save/restore point in the second save/restore point memory segment, that may not be possible in AHR MIPS depending on when the TMR to TSR transition takes place. If that transition occurs before the first TSR MIPS save/restore point is created, then the TSR worst-case error is still encountered at the end of creating a save/restore point in the second segment; in this case this would be the save/restore point created when the loop counter is at 250. This scenario is shown in Figure A17.

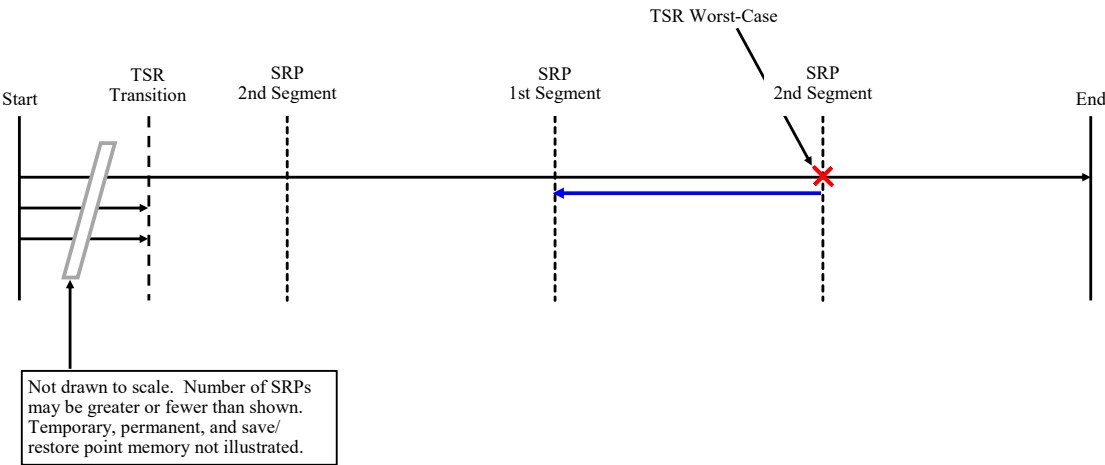

**Figure A17.** AHR MIPS TSR Worst-Case Early Error Timing Diagram 1.

When the TMR to TSR transition occurs after what would have been the first TSR MIPS save/restore point creation and before the second TSR MIPS save/restore point creation, there are two possibilities for a worst-case error. These possibilities are shown in Figure A18. Note that the first save/restore point created after the transition is always to the second save/restore point memory segment. This means that an error at the end of this save/restore point creation may not be the worst-case error. The worst-case error may be the one that occurs at the end of the next save/restore point creation which saves to the first save/restore point memory segment. The time to recover from the error and return to the point at which the error was encountered is calculated for both of these scenarios and the one that takes longer is the worst-case error.

If the TSR Worst-case error occurs after the second TSR MIPS save/restore point creation and before the third, then it is unclear what the worst-case error might be. According to the original definition of a TSR MIPS Worst-case error, it is an error that maximizes the number of instructions that TSR MIPS must re-execute. Therefore, the error may occur at the end of creating the third TSR MIPS save/restore point or at the last branch comparison at the end of the program. The amount of time to return to the point at which the error occurred is calculated for both scenarios, and the one that takes longer is the worst-case scenario. This is illustrated graphically in Figure A19.

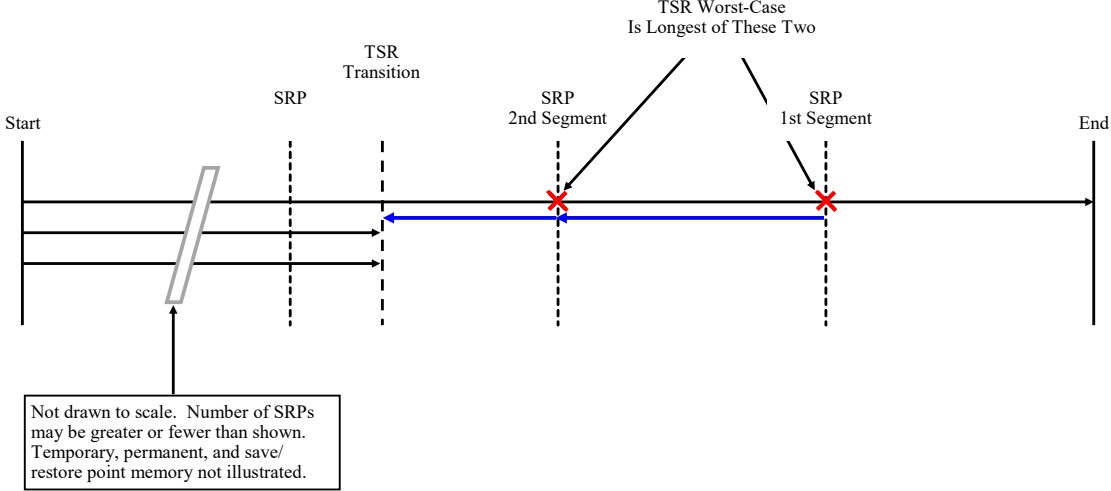

**Figure A18.** AHR MIPS TSR Worst-Case Early Error Timing Diagram 2.

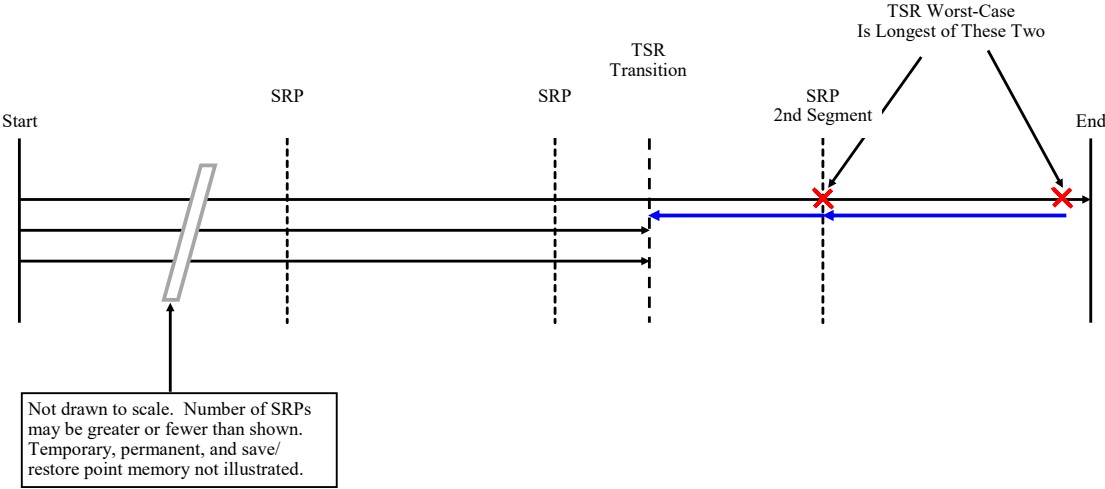

**Figure A19.** AHR MIPS TSR Worst-Case Early Error Timing Diagram 3.

Finally, if the TSR Worst-case error occurs after the last TSR MIPS save/restore point creation, the worst-case error occurs at the last branch comparison at the end of the program as shown in Figure A20.

No errors are injected to Basic MIPS because it has no way of detecting or correcting the errors. Any errors injected into a register to be stored to memory would not impact the runtime or energy usage of Basic MIPS. The only manifestation would be that the resulting computations would be incorrect.

Equations (A41) and (A42) show how to compute the time to complete a AHR MIPS program experiencing a TSR Worst-case error where Equation (A42) is a continuation of Equation (A41). If the transition point occurs before the completion of the first 250 loops, the AHR MIPS TSR worst-case error is identical to the TSR MIPS worst-case error in that the added time to complete the program is the same as in Equation (A14).

If the transition point occurs between the completion of 250 loops and 500 loops, there are two possibilities for the worst-case error. The first is that the error occurs at the end of creating the save/restore point upon completion of 500 loops, in which case all loops after the TMR to TSR transition must be re-completed and the save/restore point must be completed without error as well ($ctsrw_1$). The second is that the error occurs at the end of creating the save/restore point upon completion of 750 loops, in which case all loops after previous save/restore point creation must be

re-completed and the save/restore point at loop number 750 must be completed without error as well ($ctsrw_2$).

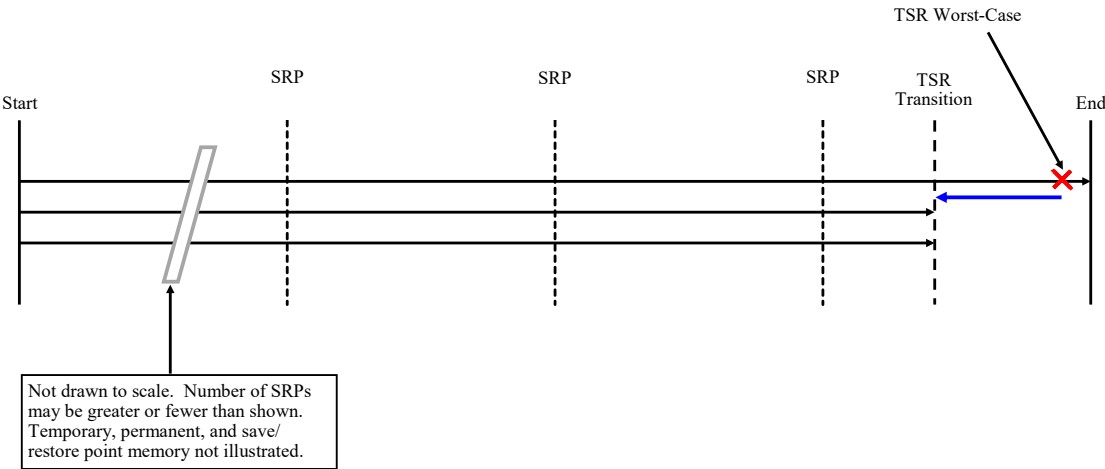

**Figure A20.** AHR MIPS TSR Worst-Case Early Error Timing Diagram 4.

If the transition point occurs between the completion of 500 loops and 750 loops, there are two possibilities for the worst-case error. The first is that the error occurs at the end of creating the save/restore point upon completion of 750 loops, in which case all loops after the TMR to TSR transition must be re-completed and the save/restore point must be completed without error as well ($ctsrw_3$). The second is that the error occurs at the last store word instruction in the program and the nearly 250 complete loops since the creation of the save/restore point at loop 750 must be re-completed ($ctsrw_4$). The only way to know which takes longer to complete is to calculate the values for both, compare the results, and select the larger of the two. If the transition point occurs after the completion of 750 loops, the worst-case error occurs at the last store word at the end of the program and all loops from the TMR to TSR transition to the end of the program must be re-completed.

Note that $t_{CTSRW\ TMR}$ and $t_{CTSRW\ TSR}$ are the time AHR MIPS spends in TMR and TSR mode respectively when encountering a TSR Worst-case error.

$$
\begin{aligned}
&if\ P_{loops}\ <\ 250 \\
&\quad t_{CTSRW\ TMR} = t_{AHR\ TMR} \\
&\quad t_{CTSRW\ TSR} = t_{AHR\ TSR} + T_{TSR\ Rec} + \sum_{n=N_{TSR}-3}^{N_{TSR}} t_{I_{TSR\ n}} + \cdots \\
&\quad 250 \cdot T_{TSR\ loop} + T_{TSR\ SRP1\ Err} \\
&elseif\ 250\ \leq\ P_{loops}\ <\ 500 \\
&\quad ctsrw_1 = T_{TSR\ Rec} + \sum_{n=N_{TSR}-3}^{N_{TSR}} t_{I_{TSR\ n}} + (500 - P_{loops}) \cdot T_{TSR\ loop} + \cdots \\
&\quad T_{TSR\ SRP1\ Err} \\
&\quad ctsrw_2 = T_{TSR\ Rec} + \sum_{n=N_{TSR}-3}^{N_{TSR}} t_{I_{TSR\ n}} + 250 \cdot T_{TSR\ loop} + T_{TSR\ SRP0\ Err} \\
&\quad t_{CTSRW\ TMR} = t_{AHR\ TMR} \\
&\quad if\ ctsrw_1 > ctsrw_2 \\
&\quad\quad t_{CTSRW\ TSR} = t_{AHR\ TSR} + ctsrw_1 \\
&\quad else \\
&\quad\quad t_{CTSRW\ TSR} = t_{AHR\ TSR} + ctsrw_2 \\
&\quad end
\end{aligned}
\tag{A41}
$$

$$
\begin{aligned}
&elseif\ 500\ \leq\ P_{loops}\ <\ 750 \\
&\quad ctsrw_3 = T_{TSR\ Rec} + \sum_{n=N_{TSR}-3}^{N_{TSR}} t_{I_{TSR\ n}} + \cdots \\
&\quad (750 - P_{loops}) \cdot T_{TSR\ loop} + T_{TSR\ SRP1\ Err} \\
&\quad ctsrw_4 = T_{TSR\ Rec} + \sum_{n=N_{TSR}-3}^{N_{TSR}} t_{I_{TSR\ n}} + 249 \cdot T_{TSR\ loop} + \cdots \\
&\quad \sum_{n_{TSR\ init}+1}^{SW_{TSR}(length(SW_{TSR})-1)} t_{I_{TSR\ n}} \\
&\qquad t_{CTSRW\ TMR} = t_{AHR\ TMR} \\
&\quad if\ ctsrw_3 > ctsrw_4 \\
&\qquad t_{CTSRW\ TSR} = t_{AHR\ TSR} + ctsrw_3 \\
&\quad else \\
&\qquad t_{CTSRW\ TSR} = t_{AHR\ TSR} + ctsrw_4 \\
&\quad end \\
&elseif\ P_{loops}\ \geq\ 750 \\
&\quad t_{CTSRW\ TMR} = t_{AHR\ TMR} \\
&\quad t_{CTSRW\ TSR} = t_{AHR\ TSR} + T_{TSR\ Rec} + \sum_{n=N_{TSR}-3}^{N_{TSR}} t_{I_{TSR\ n}} + \cdots \\
&\quad (n_{loops} - P_{loops} - 1) \cdot T_{TSR\ loop} + \sum_{n_{TSR\ init}+1}^{SW_{TSR}(length(SW_{TSR})-1)} t_{I_{TSR\ n}} \\
&end \\
&T_{CTSR\ Worst} = t_{CTSRW\ TMR} + t_{CTSRW\ TSR}
\end{aligned}
\tag{A42}
$$

## Appendix C. Error Injection Energy Calculations

The energy computations are straightforward for TMR MIPS and TSR MIPS programs even when errors are injected. The time to complete these programs is multiplied by the dynamic power used by the appropriate architecture. The TMR Type A, Type B-Best, and Type B-Worst error energy calculations are shown in Equations (A43)–(A45) respectively. The TSR MIPS Best-case and Worst-case error energy calculations are shown in Equations (A46) and (A47) respectively.

$$
E_{TMR\ ErrA} = P_{TMR\ MIPS} \cdot T_{TMR\ ErrA}
\tag{A43}
$$

$$
E_{TMR\ ErrB\ Best} = P_{TMR\ MIPS} \cdot T_{TMR\ ErrB\ Best}
\tag{A44}
$$

$$
E_{TMR\ ErrB\ Worst} = P_{TMR\ MIPS} \cdot T_{TMR\ ErrB\ Worst}
\tag{A45}
$$

$$
E_{TSR\ Best} = P_{TSR\ MIPS} \cdot T_{TSR\ Best}
\tag{A46}
$$

$$
E_{TSR\ Worst} = P_{TSR\ MIPS} \cdot T_{TSR\ Worst}
\tag{A47}
$$

While it was trivial to calculate the energy used by TMR MIPS and TSR MIPS programs experiencing errors, it is more complicated to calculate the energy used by programs running in AHR MIPS. It is more difficult because of the time divided between TMR and TSR modes of operation. Fortunately, the times to complete the TMR and TSR portions were recorded separately to make these calculations simpler.

Equations (A48) and (A49) show how to calculate the energy used by AHR MIPS when encountering TMR Type A Early and Late errors respectively.

$$
E_{CTMR\ A\ Early} = P_{CTMR\ MIPS} \cdot t_{CTMRAE\ TMR} + P_{CTSR\_MIPS} \cdot t_{CTMRAE\ TSR}
\tag{A48}
$$

$$
E_{CTMR\ A\ Late} = P_{CTMR\ MIPS} \cdot t_{CTMRAL\ TMR} + P_{CTSR\_MIPS} \cdot t_{CTMRAL\ TSR}
\tag{A49}
$$

Equations (A50) and (A51) show how to calculate the energy used by AHR MIPS when encountering a TMR Type B-Best Early and Late error respectively.

$$
\begin{aligned}
E_{CTMR\ B\ Best\ Early} &= P_{CTMR\ MIPS} \cdot t_{CTMRBBE\ TMR} + \cdots \\
&P_{CTSR\_MIPS} \cdot t_{CTMRBBE\ TSR}
\end{aligned}
\tag{A50}
$$

$$
\begin{aligned}
E_{CTMR\ B\ Best\ Late} = P_{CTMR\ MIPS} \cdot t_{CTMRBBL\ TMR} + \cdots \\
P_{CTSR\_MIPS} \cdot t_{CTMRBBL\ TSR}
\end{aligned}
\tag{A51}
$$

Equations (A52) and (A53) show how to calculate the energy used by AHR MIPS when encountering a TMR Type B-Worst Early and Late error respectively.

$$
\begin{aligned}
E_{CTMR\ B\ Worst\ Early} = P_{CTMR\ MIPS} \cdot t_{CTMRBWE\ TMR} + \cdots \\
P_{CTSR\_MIPS} \cdot t_{CTMRBWE\ TSR}
\end{aligned}
\tag{A52}
$$

$$
\begin{aligned}
E_{CTMR\ B\ Worst\ Late} = P_{CTMR\ MIPS} \cdot t_{CTMRBWL\ TMR} + \cdots \\
P_{CTSR\_MIPS} \cdot t_{CTMRBWL\ TSR}
\end{aligned}
\tag{A53}
$$

Equations (A54) and (A55) show how to calculate the energy used by AHR MIPS when encountering a TSR Type Best-Case and Worst-Case error respectively.

$$
E_{CTSR\ Best} = P_{CTMR\ MIPS} \cdot t_{CTSRB\ TMR} + P_{CTSR\_MIPS} \cdot t_{CTSRB\ TSR}
\tag{A54}
$$

$$
E_{CTSR\ Worst} = P_{CTMR\ MIPS} \cdot t_{CTSRW\ TMR} + P_{CTSR\_MIPS} \cdot t_{CTSRW\ TSR}
\tag{A55}
$$

**Appendix D. VHDL Code to Reproduce Basic MIPS, TMR MIPS, TSR MIPS, and AHR MIPS**

The VHDL code used to implement Basic MIPS, TMR MIPS, TSR MIPS, AHR MIPS and perform error injection for simulations is available on GitHub at: https://github.com/nicolas-hamilton/Adaptive-Hybrid-Redundancy-VHDL.

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
