# Peer review of "Adaptive-Hybrid Redundancy with Error Injection"

_electronics, doi:10.3390/electronics8111266_

Round 1
Reviewer 1 Report
The structure of the paper is not adequate. It lacks a clear objective. The words 'aim', 'goal' or 'objective' do not even appear in the 60 pages of the paper. You write, “This work will examine how AHR performs when the TMR to TSR switch point is varied as well as how it performs when errors are injected into the simulation.” However, it is not clear why you are doing it. Albeit the unusual extension of the introduction, it is not clearly defined which novel ideas/methods authors are proposing. TMR and TSR and well-known methods and their comparative performance in terms of speed and energy consumption known.
Fault-tolerant microprocessor architectures have been studied by many authors for a long time now. Several different fault injection techniques were proposed and used to test the effectiveness of those fault-tolerant infrastructures. Extensive literature exists about these subjects. However, no account of all this previous work is given in this paper. Therefore it is difficult to understand which are the innovative aspects of the work that justify its publication.
Another problem with this paper is its extension. Authors are unable to synthesize the information making it understandable to readers. On the contrary, they excessively detail all the steps of the work done, as if they were writing a Ph.D. thesis instead of a paper, while missing to explain certain aspects that would be important to understand the work. This is evident right in the first pages of the paper.
For example:
Page 2 – “If they are not equal, an error has been detected and error recovery occurs.” It is not clear how this recovery mechanism works.
Page 4 – “AHR was shown to bridge the gap between these two methods by switching between TMR and TSR so that it runs faster than TSR and uses less energy than TMR at the expense of running slower than TMR and using more energy than TMR.” This is obvious but which is not clear is when and why one or the other will be used. If it is only to recover from errors that the system will pass from TSR to TMR, returning to TSR after n instructions, this statement is less clear, as running using TSR or TMR is dependent on the error rate. If the error rate is small (“An appropriate error rate to be used for analysis was determined to be approximately one single event upset (SEU) per hour”), the system will almost exclusively run in TSR mode. Therefore, it will always run slower than non-redundant and TMR architectures. But running slower or faster will depend on the requirements of specific applications and the constraints of the system.
More generically, apart from the occurrence of two consecutive errors, when TSR mode changes to TMR mode, which seems to be a rare event due to the low error rate, it is not clear what determines the transitions between TMR to TSR? It is mentioned that it is after n number of instructions, but it is not specified how n is determined. Why 10,000 and not 5,000 or 20,000?
In page 35 authors write, “Figures 31 to 38 show what happens to the bounding boxes as the TMR to TSR transition point increases from 11,000, to 20,000, 30,000, 40,000, 50,000, 60,000, 70,000, and 80,000 instructions.”. But why must/should the TMR to TSR transition point increase or decrease? What determines the optimal number of instructions before the transition to occur?
Page 5 – “Injecting errors into GPRs was selected as the best method because it would allow an apples-to-apples comparison between TMR, TSR, and AHR performance in the presence of errors because all three architectures are able to detect and correct these errors. TSR may not detect and correct as many instruction, FSM, and program counter register errors as TMR is able to detect and correct.” If the system runs mainly in TSR mode and errors are not detected in this mode, what are the consequences for the system, if any, when those errors go undetected? Can you just ignore them as you tell?
Page 7 – What are the criteria used to determine when to create save/restore points?
Page 12 – “TSR errors encountered when AHR is operating in TSR mode are almost identical to TSR errors” I do not understand this statement.
To test the approach, the authors use 1,000 programs. However, nothing is said about these 1,000 programs. We have no idea which resources these programs use and how it affects (or not) the obtained results?
These are examples of statements produced and parameters/numbers assumed that are not justified in the text.
In comparison to the extension and detail of the description of the work done, the discussion of the results occupies just five lines highlighting the lack of a clear goal and the inability of the authors to synthesize the results and provide a meaningful analysis based on the proposed objectives. “AHR still uses less energy than TMR and takes less time than TSR even when errors are injected. Additionally, changing the TMR to TSR transition point allows space vehicle designers, mission planners, and operators the flexibility to select operating points that meet mission processing speed and energy usage requirements not only under optimal error free conditions, but also in the worst-case error scenarios.” This is a conclusion already presented in the Introduction and that is obvious due to the nature of the two techniques used, TSR and TMR. In space applications, to use one or the other depends on the specific requirements and constraints of each one of the applications, and may vary along the journey. With approximately one single event upset (SEU) per hour, the analysis that was done albeit rigorous, lack significance as switching from TSR to TMR and back for 10,000 or 80,000 will have a minimal impact in speed and energy consumption when compared with the billions of instructions that will run in one hour.
In sum, it is very difficult to review this paper because it lacks a clear objective and a clear indication of the innovative aspects of the work done by the authors, which prevents a good understanding of the novelties and contributions of this work if any. It also lacks a state-of-art section able to show the innovative aspects of this work that justify its publication. It also needs to be substantially reduced, making it more objective and less descriptive. The discussion of the results needs to be much more in-depth, clearly showing that the initial objectives were achieved and the novelty of the work proved.
The correction of these aspects is essential to clearly show the relevance of the work to other researchers and its merits, making it worthy of publication.
Author Response
Dear Reviewers,
The authors are grateful for your quick response and in depth review of our paper. You encouraged us to reexamine the clarity of our paper’s purpose and sourcing of the research. You have also challenged us to improve the quality of our paper by providing crucial feedback on sections and statements that were opaque and difficult to follow for the reader. We have endeavored to improve on these shortcomings by addressing each of your comments to the best of our ability and substantially revising our paper. We have addressed your comments in bold, italicized text below each comment. We look forward to further comments to help us make this paper even better. Thank you.
Reviewer 1
The structure of the paper is not adequate. It lacks a clear objective. The words 'aim', 'goal' or 'objective' do not even appear in the 60 pages of the paper. You write, “This work will examine how AHR performs when the TMR to TSR switch point is varied as well as how it performs when errors are injected into the simulation.” However, it is not clear why you are doing it. Albeit the unusual extension of the introduction, it is not clearly defined which novel ideas/methods authors are proposing. TMR and TSR and well-known methods and their comparative performance in terms of speed and energy consumption known.
The authors are grateful to Reviewer 1 for pointing out the lack of a clear objective statement. To remedy this, we have added an objective statement in the introduction on page 1 to make it clearer that one of the novel aspects of this paper is the variation in the TMR to TSR transition point when compared to the previous work which kept the TMR to TSR transition point fixed at 15,000 instructions. We also point out that another novel aspect of this paper is that this flexibility is still present when errors are present.
Fault-tolerant microprocessor architectures have been studied by many authors for a long time now. Several different fault injection techniques were proposed and used to test the effectiveness of those fault-tolerant infrastructures. Extensive literature exists about these subjects. However, no account of all this previous work is given in this paper. Therefore it is difficult to understand which are the innovative aspects of the work that justify its publication.
Thank you for noting the absence of background information. We have added background information that addresses various redundancy methods and mentions a few different error injection techniques on pages 2, 3, and 4.
Another problem with this paper is its extension. Authors are unable to synthesize the information making it understandable to readers. On the contrary, they excessively detail all the steps of the work done, as if they were writing a Ph.D. thesis instead of a paper, while missing to explain certain aspects that would be important to understand the work. This is evident right in the first pages of the paper.
We originally included the detailed steps of the work because the Journal Submission Guidelines request that authors include as much detail as possible so that others can reproduce their work.
We have moved the more detailed steps of the work to the appendices because we feel it is still necessary to include them to address the Journal Submission Guidelines. We have also addressed the missing aspects you specifically mention below.
For example:
Page 2 – “If they are not equal, an error has been detected and error recovery occurs.” It is not clear how this recovery mechanism works.
We have described how the recovery mechanism functions in more detail as it pertains to DMR on page 2. This same, or similar, recovery mechanism is used by TMR, TSR, and AHR. TMR does this using hardware just like DMR. TSR does this by implementing the same recovery mechanism as DMR in software. AHR does this using the TMR recovery method when in TMR mode and the TSR recovery method when in TSR mode.
Page 4 – “AHR was shown to bridge the gap between these two methods by switching between TMR and TSR so that it runs faster than TSR and uses less energy than TMR at the expense of running slower than TMR and using more energy than TMR.” This is obvious but which is not clear is when and why one or the other will be used. If it is only to recover from errors that the system will pass from TSR to TMR, returning to TSR after n instructions, this statement is less clear, as running using TSR or TMR is dependent on the error rate. If the error rate is small (“An appropriate error rate to be used for analysis was determined to be approximately one single event upset (SEU) per hour”), the system will almost exclusively run in TSR mode. Therefore, it will always run slower than non-redundant and TMR architectures. But running slower or faster will depend on the requirements of specific applications and the constraints of the system.
More generically, apart from the occurrence of two consecutive errors, when TSR mode changes to TMR mode, which seems to be a rare event due to the low error rate, it is not clear what determines the transitions between TMR to TSR? It is mentioned that it is after n number of instructions, but it is not specified how n is determined. Why 10,000 and not 5,000 or 20,000?
In page 35 authors write, “Figures 31 to 38 show what happens to the bounding boxes as the TMR to TSR transition point increases from 11,000, to 20,000, 30,000, 40,000, 50,000, 60,000, 70,000, and 80,000 instructions.”. But why must/should the TMR to TSR transition point increase or decrease? What determines the optimal number of instructions before the transition to occur?
Reviewer 1 perfectly understands the operation of AHR and when AHR transitions from TMR to TSR and TSR to TMR. Reviewer 1 is also correct in stating that the system will almost exclusively run in TSR mode and that running slower or faster will depend on the requirements of specific applications and constraints on the system.
We have attempted to clarify when and why AHR would operate in either TMR or TSR. We have added an explanation concerning how the TMR to TSR transition point can be controlled in such a way that a space vehicle designer, mission planner, or operator can vary the amount of time AHR operates in TMR or TSR modes as desired by changing the transition point at will. Please see the Discussion on page 22 of the revised paper. We have also added example scenarios for why one might prefer to use AHR in TMR mode or TSR mode in the last full paragraph of page 11. The TMR to TSR transition point could be varied from 0 to any arbitrarily large number, what impact that would have on AHR, and why a space vehicle designer, mission planner, or operator might choose to change that transition point at any particular time.
Optimality is determined by the space vehicle designer, mission planner, or operator based upon mission needs (i.e. radiation environment, required processing speed, required energy usage, and etc.).
Page 5 – “Injecting errors into GPRs was selected as the best method because it would allow an apples-to-apples comparison between TMR, TSR, and AHR performance in the presence of errors because all three architectures are able to detect and correct these errors. TSR may not detect and correct as many instruction, FSM, and program counter register errors as TMR is able to detect and correct.” If the system runs mainly in TSR mode and errors are not detected in this mode, what are the consequences for the system, if any, when those errors go undetected? Can you just ignore them as you tell?
We have added a discussion about what could occur in TSR if an error occurred in a register other than a general purpose register. We have also discussed that the likelihood of an error occurring in one of these other registers is small. Please see Section 2.1 on pages 7 and 8 for this discussion
Page 7 – What are the criteria used to determine when to create save/restore points?
The save/restore point creation criteria would be mission specific, but was arbitrarily chosen for this research based on the size of the programs to ensure that every program created at least one save/restore point. Additionally, this criteria would be mission specific in real world applications. This has been updated in the paper. Please see the discussion of DMR Save/Restore point creation in the second paragraph of Section 1.1.1 on page 2 and the first paragraph of Section 1.2 on page 4
Page 12 – “TSR errors encountered when AHR is operating in TSR mode are almost identical to TSR errors” I do not understand this statement.
We have reworded this statement to clarify. “Errors encountered when AHR MIPS is operating in TSR mode are virtually identical to the errors encountered by TSR MIPS, but depend upon the point at which the TMR to TSR transition occurs within a program.” This is now located in Appendix B at the bottom of page 29
To test the approach, the authors use 1,000 programs. However, nothing is said about these 1,000 programs. We have no idea which resources these programs use and how it affects (or not) the obtained results?
We have added a discussion about the nature of the 1,000 programs in the last paragraph of Section 2.2 on page 9
These are examples of statements produced and parameters/numbers assumed that are not justified in the text.
In comparison to the extension and detail of the description of the work done, the discussion of the results occupies just five lines highlighting the lack of a clear goal and the inability of the authors to synthesize the results and provide a meaningful analysis based on the proposed objectives. “AHR still uses less energy than TMR and takes less time than TSR even when errors are injected. Additionally, changing the TMR to TSR transition point allows space vehicle designers, mission planners, and operators the flexibility to select operating points that meet mission processing speed and energy usage requirements not only under optimal error free conditions, but also in the worst-case error scenarios.” This is a conclusion already presented in the Introduction and that is obvious due to the nature of the two techniques used, TSR and TMR. In space applications, to use one or the other depends on the specific requirements and constraints of each one of the applications, and may vary along the journey. With approximately one single event upset (SEU) per hour, the analysis that was done albeit rigorous, lack significance as switching from TSR to TMR and back for 10,000 or 80,000 will have a minimal impact in speed and energy consumption when compared with the billions of instructions that will run in one hour.
In sum, it is very difficult to review this paper because it lacks a clear objective and a clear indication of the innovative aspects of the work done by the authors, which prevents a good understanding of the novelties and contributions of this work if any. It also lacks a state-of-art section able to show the innovative aspects of this work that justify its publication. It also needs to be substantially reduced, making it more objective and less descriptive. The discussion of the results needs to be much more in-depth, clearly showing that the initial objectives were achieved and the novelty of the work proved.
The correction of these aspects is essential to clearly show the relevance of the work to other researchers and its merits, making it worthy of publication.
Extensive revisions have been made to address the aforementioned comments. The discussion of results has also been updated to better capture the importance and impact of the research.

Reviewer 2 Report
The paper tackles the important problem of redundancy in electronic components. In the paper Authors proposes a hybrid of software/hardware redundancy scheme. The paper itself is extensive and well addresses the given topic. However it is extremely long (60 pages) which makes it difficult to follow, especially in the experimental section. I would advice to remove all the code, leaving only equations and some pseudo-code. I also advice to make some table of references instead enabling some shortenings, e.g. instead }\P_{CTRM MIPS}} it may be {\P_{CM}} so also spaces are excluded. This would make all the equations and pseudo-code easier to read and follow. Also readability od the figures might be improved. Figures 6 and 7 can be replaced by a more clear and simplified block diagrams. Furthermore Figs. 26 and 27 can be fixed, so the particular elements are more visible. It goes without saying that in case of chosen models of MIPS the relation between power consumption and calculation time is linear for either TMR or TSM. Thus I would remove the figures in favour a table comparing either the energy or time. and leave only the figure that compares them with hybrid approach. I would also recommend changing a type of figure. Instead of Average Performance Bounds maybe a centroids may represent the bounds, thus making he figure 39 clearer. Also the references are a bit doubtful as 50% of it are self-citations, 3 out of 10 are only technical reports. Position 1 and 10 cannot be found in any of the following: IEEE Xplore, Google Scholar. Web of Science. Position 10 in references is marked as submitted, thus it should not be referred to as it is unpublished. Probably position 1 is also not published yet. I strongly advice to make an extra effort for additional research in order to extend the references (or prove that none else makes a research in this area, as it seems improbable).
While the topic seems to be very attractive and conclusions are supported by the results it is hard to say whether it is conclusive and meaningful due to a poor choice of references.
Author Response
Reviewer 2
The paper tackles the important problem of redundancy in electronic components. In the paper Authors proposes a hybrid of software/hardware redundancy scheme. The paper itself is extensive and well addresses the given topic. However it is extremely long (60 pages) which makes it difficult to follow, especially in the experimental section. I would advice to remove all the code, leaving only equations and some pseudo-code. I also advice to make some table of references instead enabling some shortenings, e.g. instead }\P_{CTRM MIPS}} it may be {\P_{CM}} so also spaces are excluded. This would make all the equations and pseudo-code easier to read and follow.
The authors are grateful to Reviewer 2 for pointing out the extreme length of the paper which makes it difficult to follow. We have moved the detailed discussion of the error types, runtime calculations, and energy calculations to the appendices to reduce the length of the paper and focus more on the big picture. We have chosen to still include these discussions, as well as the code, because the Journal Submission Guidelines request that authors include as much detail as possible so that others can reproduce their work
Also readability od the figures might be improved. Figures 6 and 7 can be replaced by a more clear and simplified block diagrams. Furthermore Figs. 26 and 27 can be fixed, so the particular elements are more visible. It goes without saying that in case of chosen models of MIPS the relation between power consumption and calculation time is linear for either TMR or TSM. Thus I would remove the figures in favour a table comparing either the energy or time. and leave only the figure that compares them with hybrid approach.
We decided to move Figures 6 and 7 to the Appendix while updating the description of error injection in the text in an effort to make the error injection mechanism easy to understand.
I would also recommend changing a type of figure. Instead of Average Performance Bounds maybe a centroids may represent the bounds, thus making he figure 39 clearer.
We decided to remove this plot because it is redundant. There are already bounding box plots for various TMR to TSR transition points and repeating those by overlaying them one on top of the other is redundant.
Also the references are a bit doubtful as 50% of it are self-citations, 3 out of 10 are only technical reports. Position 1 and 10 cannot be found in any of the following: IEEE Xplore, Google Scholar. Web of Science.
We have provided a much more extensive background section with references within the introduction.
Position 10 in references is marked as submitted, thus it should not be referred to as it is unpublished.
I have removed this reference since it is not yet published.
We have removed reference 10 because it has not been published.
Probably position 1 is also not published yet. I strongly advice to make an extra effort for additional research in order to extend the references (or prove that none else makes a research in this area, as it seems improbable).
The NAECON conference proceedings have not yet been published, but likely will be before this paper is published in the Fault-Tolerant Digital Circuits: Protection Techniques, CAD Tools, and Emerging Applications special issue of the Electronics Journal which we are targeting. In the past, NAECON was held in July and the proceedings were published in December. The submission deadline for this special issue is 31 March 2020, well after when the NAECON proceedings will be published.
While the topic seems to be very attractive and conclusions are supported by the results it is hard to say whether it is conclusive and meaningful due to a poor choice of references.
We have provided a much more extensive background section with references within the introduction to help Reviewer 2 determine that this work is in fact conclusive and meaningful.

Reviewer 3 Report
This paper proposes hardware and software redundancy architectures to make the processors robust to radiation events, present in some fields such as aeronautics and energy production.
The TMR architecture is a modular redundancy (hardware redundancy) associated with a voting device that selects the non-faulty module.
The TSR architecture is a hybrid software and hardware redundancy, software redundancy by the duplication of instructions and hardware by the duplication of registers.
The AHR architecture is a hybrid redundancy that adds a controller to the TMR architecture.
The proposed methods are implemented using the Matlab software.
Injecting errors into GPRs was selected as the best method because it would allow an apples-to-apples comparison between TMR, TSR, and AHR performance in the presence of errors because all three architectures are able to detect and correct these errors. Errors are injected into Basic MIPS GPRs by adding hardware to the Basic MIPS Datapath. 

The authors presented in detail the error injection procedure which is indeed an important step in the validation of this work. Errors are injected as programs that represent single or multiple errors. Formal (mathematical) arguments have been given by the authors to confirm the validity of error injection processes. I am not expert enough to comment on the relevance of this part, but the practical value of this work is undeniable.
The paper can be improved on the following points :
1) It would be good to compare the proposed approach to existing systems diagnostic approaches, an overview of these approaches is proposed in
A survey of fault diagnosis and fault-tolerant techniques-part II: Fault diagnosis with knowledge- based and hybrid/active approaches. IEEE Transactions on Industrial Electronics, 62(6), 3768–3774.
Architectures for online error detection and recovery in multicore processors. 2011 Design, Automation & Test in Europe, (c), 1–6. doi: 10.1109/DATE.2011.5763096.
Testing and built-in self-test – A survey. Journal of Systems Architecture, 46(9), 721–747. doi:10.1016/S1383-7621(99)00041-7.
2) In some risky applications, the stability of the switch from one configuration to another must be studied, in this work, can the transition from one configuration to the other after occurrence of a fault cause instability of the system?
3) If this is the case, can setting up an early detection or fault prognosis tool in embedded systems be a solution? as for example the method proposed in "Data-driven approach augmented in simulation for robust fault prognosis . " Engineering Applications of Artificial Intelligence 86 (2019) 154–164, that it is good to address in the introduction.
4) Before the reconfiguration, there is the diagnosis, the introduction can be improved with papers dealing with the diagnosis of the drifts of the characteristics of the embedded systems, because the diagnosis is a preliminary step to the reconfiguration by redundancy. As for example the method proposed in: Data-Driven Approach for Feature Drift Detection in Embedded Electronic Devices IFAC-PapersOnLine, Volume 51, Issue 24, 2018, Pages 1024-1029.
Author Response
Reviewer 3
This paper proposes hardware and software redundancy architectures to make the processors robust to radiation events, present in some fields such as aeronautics and energy production.
The TMR architecture is a modular redundancy (hardware redundancy) associated with a voting device that selects the non-faulty module.
The TSR architecture is a hybrid software and hardware redundancy, software redundancy by the duplication of instructions and hardware by the duplication of registers.
The authors are thankful to Reviewer 3 for an accurate summary of TMR and TSR architecture as we presented them; however, your comments on TMR and TSR have caused us to recognize our failure to accurately portray TMR and TSR. We have added additional background information on pages 2, 3, and 4 to clarify the operation of TMR and TSR.
The AHR architecture is a hybrid redundancy that adds a controller to the TMR architecture.
This statement is accurate
The proposed methods are implemented using the Matlab software.
We have added clarifying statements that AHR, TMR, and TSR were implemented in VHDL. This stands in contrast to the statement concerning Matlab which was used to perform the timing and energy analyses. Please see the opening paragraph of Section 2 on page 7.
Injecting errors into GPRs was selected as the best method because it would allow an apples-to-apples comparison between TMR, TSR, and AHR performance in the presence of errors because all three architectures are able to detect and correct these errors. Errors are injected into Basic MIPS GPRs by adding hardware to the Basic MIPS Datapath. 

We have updated the text addressing why injecting errors into GPRs was best to address comments by Reviewer 1. While it is still accurate to say that GPR error injection allows an apples-to-apples comparison between TMR, TSR, and AHR, we have added text to discuss the possible outcomes if errors occurred in registers other than GPRs for TSR. Please see Section 2.1 on pages 7 and 8 for this discussion
The authors presented in detail the error injection procedure which is indeed an important step in the validation of this work. Errors are injected as programs that represent single or multiple errors. Formal (mathematical) arguments have been given by the authors to confirm the validity of error injection processes. I am not expert enough to comment on the relevance of this part, but the practical value of this work is undeniable.
The paper can be improved on the following points :
1) It would be good to compare the proposed approach to existing systems diagnostic approaches, an overview of these approaches is proposed in
A survey of fault diagnosis and fault-tolerant techniques-part II: Fault diagnosis with knowledge- based and hybrid/active approaches. IEEE Transactions on Industrial Electronics, 62(6), 3768–3774.
Architectures for online error detection and recovery in multicore processors. 2011 Design, Automation & Test in Europe, (c), 1–6. doi: 10.1109/DATE.2011.5763096.
Testing and built-in self-test – A survey. Journal of Systems Architecture, 46(9), 721–747. doi:10.1016/S1383-7621(99)00041-7.
We have added more background materials to the paper with references. Many of these references were also cited by the “Architectures for online error detection and recovery in multicore processors” paper. We found that “A survey of fault diagnosis and fault-tolerant techniques-part II: Fault diagnosis with knowledge- based and hybrid/active approaches” and “Testing and built-in self-test – A survey. Journal of Systems Architecture” were not directly applicable to this research because they do not examine the unique aspect of single event upsets in a radiation environment. A comparison of AHR to TMR and TSR redundancy approaches is included in the discussion of results.
2) In some risky applications, the stability of the switch from one configuration to another must be studied, in this work, can the transition from one configuration to the other after occurrence of a fault cause instability of the system?
The transition itself would not cause system instability, however, a frequent concern for hardware redundancy systems is the immunity of the voter from errors. If the voter can suffer an error, then an unrecoverable system fault may occur. This logic would extend to the controller module in AHR. In this work, it was assumed that the voter and AHR controller are immune from errors. This immunity could be created by implementing the voter and AHR controller in a rad-hardened by design chip (i.e. different transistor technology, shielding, redundancy, or any combination of these). These error immunity assumptions have been explicitly stated in Section 1.1.1 on page 2 (for the TMR voter) and in the last paragraph on page 5 (for the AHR controller).
3) If this is the case, can setting up an early detection or fault prognosis tool in embedded systems be a solution? as for example the method proposed in "Data-driven approach augmented in simulation for robust fault prognosis . " Engineering Applications of Artificial Intelligence 86 (2019) 154–164, that it is good to address in the introduction.
The voter and AHR controller are assumed to be immune from errors so there is no need to set up an early detection or fault prognosis tool because there is no way that the stability of the switch could be impacted.
4) Before the reconfiguration, there is the diagnosis, the introduction can be improved with papers dealing with the diagnosis of the drifts of the characteristics of the embedded systems, because the diagnosis is a preliminary step to the reconfiguration by redundancy. As for example the method proposed in: Data-Driven Approach for Feature Drift Detection in Embedded Electronic Devices IFAC-PapersOnLine, Volume 51, Issue 24, 2018, Pages 1024-1029.
More background information was added to address how other methods detect that an error has occurred as well as information on how this paper detects errors

Round 2
Reviewer 1 Report
The authors thoroughly and satisfactorily approached all my concerns. Only two notes that should be addressed in the final paper:
Line 93 – “Oh et al.” – cross-reference citations should use the same format in the whole text. Furthermore, several papers may be referred to as “Oh et al.”. Authors should write the cross-reference they are using in the References’ list that enables the identification of which paper they are considering – [17], [18] or [20]
Line 104 – “The first redundancy method discussed is only applicable to Field Programmable Gate Arrays (FPGAs)”. Which first redundancy method is authors referring to? TMR? TMR theory is older than FPGAs themselves. The first paper that mentions TMR, #Probabilistic Logics and the Synthesis of Reliable Organisms from Unreliable Components#, was written by J. von Neumann and published in Automata Studies, a journal published by Princeton University Press, in 1956, almost 30 years before the introduction of the first SRAM-based FPGA. This statement seems not to be correct. Maybe some context able to justify the "is only applicable" is missing.
Author Response
Reviewer 1, Round 2 comments:
The authors thoroughly and satisfactorily approached all my concerns. Only two notes that should be addressed in the final paper:
Line 93 – “Oh et al.” – cross-reference citations should use the same format in the whole text. Furthermore, several papers may be referred to as “Oh et al.”. Authors should write the cross-reference they are using in the References’ list that enables the identification of which paper they are considering – [17], [18] or [20]
We have attempted to address this comment by more clearly linking authors to references. Please see lines 91, 92, and 95.
Line 104 – “The first redundancy method discussed is only applicable to Field Programmable Gate Arrays (FPGAs)”. Which first redundancy method is authors referring to? TMR? TMR theory is older than FPGAs themselves. The first paper that mentions TMR, #Probabilistic Logics and the Synthesis of Reliable Organisms from Unreliable Components#, was written by J. von Neumann and published in Automata Studies, a journal published by Princeton University Press, in 1956, almost 30 years before the introduction of the first SRAM-based FPGA. This statement seems not to be correct. Maybe some context able to justify the "is only applicable" is missing.
We have attempted to address this comment by clarifying that this is the first redundancy method discussed in the section on hybrid redundancy. It has been rephrased to state that “The first hybrid redundancy method this paper discusses is only…”. We also noted a similar statement at the beginning of the software redundancy section. This line previously stated “The first software redundancy method, Error Detection by Duplicated Instructions (EDDI), is…” has been changed to “The first software redundancy method discussed in this paper, Error Detection by Duplicated Instructions (EDDI), is …”.

Reviewer 2 Report
After considering the reply from the Authors I hereby recommend to accept the paper. However I am still concerned with reference 1. Also I remain with an opinion that placing all the codes somewhere for download, e.g. Github would not diminish the fullness of a paper in scope of Journal Guidelines but improve the ease to repeat the experiments carried by the Authors.
Author Response
Reviewer 2, Round 2 comments:
After considering the reply from the Authors I hereby recommend to accept the paper. However I am still concerned with reference 1. Also I remain with an opinion that placing all the codes somewhere for download, e.g. Github would not diminish the fullness of a paper in scope of Journal Guidelines but improve the ease to repeat the experiments carried by the Authors.
We have revised reference 1 to point to the first author’s PhD dissertation available on AFIT Scholar. We have also created a github repository containing all VHDL code for Basic MIPS, TMR MIPS, TSR MIPS, and AHR MIPS and referenced this repository in the text.
